# Omissions of threat trigger subjective relief and prediction error-like signaling in the human reward and salience systems

Anne L Willems[1,2]*, Lukas Van Oudenhove[2,3], Bram Vervliet[1,2]

[1]Laboratory of Biological Psychology, Department of Brain & Cognition, KU Leuven, Leuven, Belgium; [2]Leuven Brain Institute, KU Leuven, Leuven, Belgium; [3]Laboratory for Brain-Gut Axis Studies (LaBGAS), Translational Research in GastroIntestinal Disorders (TARGID), Department of chronic diseases and metabolism, KU Leuven, Leuven, Belgium

## eLife assessment

This study presents **valuable** findings on the relationship between prediction errors and brain activation in response to unexpected omissions of painful electric shocks. The strengths are the research question posed, as it has remained unresolved if prediction errors in the context of biologically aversive outcomes resemble reward-based prediction errors. The evidence is **solid** but there are weaknesses in the experimental design, where verbal instructions do not align with experienced outcome probabilities. It is further unclear how to interpret neural prediction error signaling in the assumed absence of learning. The work will be of interest to cognitive neuroscientists and psychologists studying appetitive and aversive learning.

**Abstract** The unexpected absence of danger constitutes a pleasurable event that is critical for the learning of safety. Accumulating evidence points to similarities between the processing of absent threat and the well-established reward prediction error (PE). However, clear-cut evidence for this analogy in humans is scarce. In line with recent animal data, we showed that the unexpected omission of (painful) electrical stimulation triggers activations within key regions of the reward and salience pathways and that these activations correlate with the pleasantness of the reported relief. Furthermore, by parametrically violating participants' probability and intensity related expectations of the upcoming stimulation, we showed for the first time in humans that omission-related activations in the VTA/SN were stronger following omissions of more probable and intense stimulations, like a positive reward PE signal. Together, our findings provide additional support for an overlap in the neural processing of absent danger and rewards in humans.

## Introduction

We experience pleasurable relief when an expected threat stays away (***Deutsch et al., 2015***). This relief indicates that the outcome we experienced ('nothing') was better than we expected it to be ('threat'). Such a mismatch between expectation and outcome is generally regarded as the trigger for new learning, and is typically formalized as the prediction error (PE) that determines how much there can be learned in any given situation (***Rescorla and Wagner, 1972***). Over the last two decades, the PE elicited by the absence of expected threat (threat omission PE) has received increasing scientific interest, because it is thought to play a central role in learning of safety. Impaired safety learning is one of the core features of clinical anxiety (***Beckers et al., 2023***). A better understanding of how the

*For correspondence: anne.willems@kuleuven.be

Competing interest: The authors declare that no competing interests exist.

Sent for Review 28 July 2023

Preprint posted 17 August 2023

Reviewed preprint posted 16 October 2023

Reviewed preprint revised 26 April 2024

Reviewed preprint revised 23 August 2024

Version of Record published 26 February 2025

threat omission PE is processed in the brain may therefore be key to optimizing therapeutic efforts to boost safety learning. Yet, despite its theoretical and clinical importance, research on how the threat omission PE is computed in the brain is only emerging.

To date, the threat omission PE has mainly been studied using fear extinction paradigms that mimic safety learning by repeatedly confronting a human or animal with a threat predicting cue (conditional stimulus, CS; e.g. a tone) in the absence of a previously associated aversive event (unconditional stimulus, US; e.g., an electrical stimulation). These (primarily non-human) studies have revealed that there are striking similarities between the PE elicited by unexpected threat omission and the PE elicited by unexpected reward. In the context of reward, it is well-established that dopamine neurons in the ventral tegmental area (VTA) and substantia nigra (SN) of the midbrain increase their firing rate to unexpected rewards (positive PE), suppress their firing for unexpected reward omissions (negative PE) and show no change in firing in response to completely predicted rewards, in line with a formalized PE (*Schultz, 2016*; *Watabe-Uchida et al., 2017*). Likewise, in fear extinction, dopaminergic neurons in the VTA phasically increase their firing rates to early (unexpected), but not late (expected) US omissions (*Luo et al., 2018*; *Salinas-Hernández et al., 2018*; *Cai et al., 2020*; *de Jong et al., 2019*; *Badrinarayan et al., 2012*), which consequently triggers downstream dopamine release in the nucleus accumbens (NAc) shell (*Luo et al., 2018*; *de Jong et al., 2019*; *Badrinarayan et al., 2012*). Furthermore, optogenetically blocking (or enhancing) the firing rate of these dopaminergic VTA neurons during US omissions impairs (or facilitates) subsequent fear extinction learning (*Luo et al., 2018*; *Salinas-Hernández et al., 2018*; *Cai et al., 2020*). Notably, such dopaminergic VTA/NAc responses to threat omissions have also been observed in other experimental tasks, such as conditioned inhibition *Yau and McNally, 2018* and avoidance (*Oleson et al., 2012*; *Wenzel et al., 2018*), confirming that these neural activations match a more general threat omission PE-signal.

In humans, reward-like PE responses to threat omissions during extinction and avoidance have mainly been reported in projection regions of dopaminergic midbrain neurons, such as the ventral striatum (more specifically, NAc and ventral putamen) and prefrontal areas (the ventromedial prefrontal cortex, vmPFC; *Raczka et al., 2011*; *Thiele et al., 2021*; *Lange et al., 2020*; *Esser et al., 2021*; *Leknes et al., 2011*; *Boeke et al., 2017*). Activations in these regions correlate with computationally modeled PE signals *Raczka et al., 2011*; *Thiele et al., 2021*; *Esser et al., 2021* and are modulated by pharmacological manipulation of dopamine receptors *Esser et al., 2021* and by genetic mutations that are known to enhance striatal phasic dopamine release (*Raczka et al., 2011*). Furthermore, connectivity analyses revealed that NAc activations during US omissions in fear extinction were functionally coupled to VTA/SN activations, and that this connectivity was enhanced by the administration of the dopamine precursor L-dopa prior to extinction (*Esser et al., 2021*). The emerging picture is that cortico-striatal activations to unexpected omissions of threat are triggered by dopaminergic inputs from the VTA/SN, just like a positive reward PE, and that these activations play a central role in different types of threat omission-induced learning such as fear extinction and avoidance learning (*Papalini et al., 2020*; *Kalisch et al., 2019*). Still, direct observations of threat omission-related VTA/SN responses are currently lacking in humans.

As mentioned above, unexpected omissions of threat not only trigger neural activations that resemble a reward PE, they are also accompanied by a pleasurable emotional experience: relief (*Deutsch et al., 2015*). Because these feelings of relief coincide with the PE at threat omission, relief has been proposed to be an emotional correlate of the threat omission PE (*Leknes et al., 2011*; *Vervliet et al., 2017*). Indeed, emerging evidence has shown that subjective experiences of relief follow the same time-course as theoretical PE during fear extinction. Participants in fear extinction experiments report high levels of relief pleasantness during early US omissions (when the omission was unexpected and the theoretical PE was high) and decreasing relief pleasantness over later omissions (when the omission was expected and the theoretical PE was low; *Vervliet et al., 2017*; *Papalini et al., 2021*). Accordingly, preliminary fMRI evidence has shown that the pleasantness of this relief is correlated to activations in the NAc at the time of threat omission (*Leknes et al., 2011*). In that sense, studying relief may offer important insights in the mechanism driving safety learning.

However, is a correlation with the theoretical PE over time sufficient for neural activations/relief to be classified as a PE-signal? In the context of reward, Caplin and colleagues proposed three necessary and sufficient criteria all PE-signals should comply to, independent of the exact operationalizations of expectancy and reward the so-called axiomatic approach (*Caplin and Dean, 2008*; *Rutledge et al.,*

*2010*); which has also been applied to aversive PE (*Jepma et al., 2022*; *Roy et al., 2014*; *Ojala et al., 2022*). Specifically, the magnitude of a PE signal should: (1) be positively related to the magnitude of the reward (larger rewards trigger larger PEs); (2) be negatively related to likelihood of the reward (more probable rewards trigger smaller PEs); and (3) not differentiate between fully predicted outcomes of different magnitudes (if there is no error in prediction, there should be no difference in the PE signal).

The previously discussed fear conditioning and extinction studies have been invaluable for clarifying the role of the threat omission PE within a learning context (*Raczka et al., 2011*; *Thiele et al., 2021*; *Lange et al., 2020*; *Esser et al., 2021*; *Vervliet et al., 2017*; *Papalini et al., 2021*). However, these studies were not tailored to create the varying intensity and probability-related conditions that are required to systematically evaluate the threat omission PE in the light of the PE axioms. First, these only included one level of aversive outcome: the electrical stimulation was either delivered or omitted; but the intensity of the stimulation was never experimentally manipulated within the same task. As a result, the magnitude-related axiom could not be tested. Second, as safety learning progressively developed over the course of extinction learning, the most informative trials to evaluate the probability axiom (i.e. the trials with the largest PE) were restricted to the first few CS+ offsets of the extinction phase, and the exact number of these informative trials likely differed across participants as a result of individually varying learning rates. This limited the experimental control and necessary variability to systematically evaluate the probability axiom. Third, because CS-US contingencies changed over the course of the task (e.g. from acquisition to extinction), there was never complete certainty about whether the US would (not) follow. This precluded a direct comparison of fully predicted outcomes. Finally, within a learning context, it remains unclear whether brain responses to the threat omission are in fact responses to the violation of expectancy itself, or whether they are the result of subsequent expectancy updating.

Based on these reasons, we recently developed the Expectancy Violation Assessment (EVA) task (*Willems and Vervliet, 2021*) in order to study threat omission responses outside of a learning context. By providing verbal instructions on the probability and intensity of an upcoming electrical stimulation, which are then violated by not delivering the stimulation, we showed that the experienced pleasantness of the omission-relief reflects the degree of fearful expectation violation, with omissions of more intense and more probable stimulations eliciting more pleasurable feelings of relief, much like a PE-signal.

Here, we applied the EVA-task in the MRI scanner to investigate brain responses to unexpected omissions of threat in greater detail, examine their similarity to reward PE-signals, and explore the link with subjective relief. Specifically, participants received trial-by-trial instructions about the probability (0%, 25%, 50%, 75%, and 100%) and intensity (weak, moderate, strong) of a potentially painful upcoming electrical stimulation, time-locked by a countdown clock (see *Figure 1A*). While stimulations were always delivered on 100% trials and never on 0% trials, most of the other trials (25–75%) did not contain the expected stimulation and hence provoked an omission PE. We expected that (1) expected-but-omitted stimulation would trigger increased activity within key areas of the reward circuit (such as the VTA/SN, NAc, left ventral Putamen and vmPFC); that (2) this omission-related activity would fit the three criteria of a positive reward PE, and that (3) this activity would be related to self-reported relief. These hypotheses and the analysis approach were preregistered on Open Science Framework (OSF, https://osf.io/ugkzf). Small deviations related to the approach are reported in Appendix 1.

## Results
### Self-reported relief and omission SCR track omissions of threat in a PE-like manner

The verbal instructions were effective at raising the expectation of receiving the electrical stimulation in line with the provided probability and intensity levels. Anticipatory SCR, which we used as a proxy of fearful expectation, increased as a function of the probability and intensity instructions (see Appendix 4). Accordingly, post-experimental questions revealed that by the end of the experiment participants recollected having received more stimulations after higher probability instructions, and were willing to exert more effort to prevent stronger hypothetical stimulations (see Appendix 5).

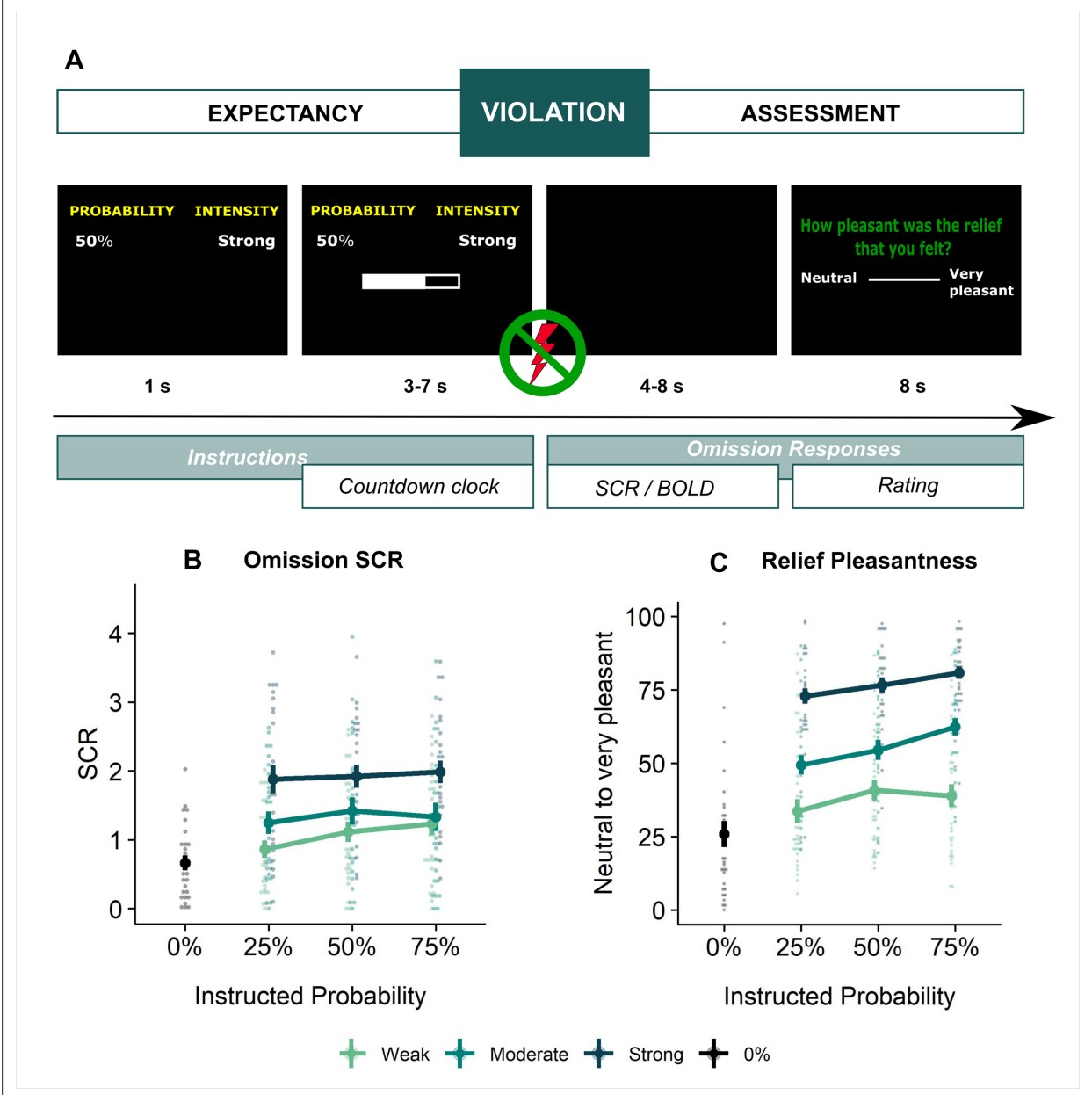

**Figure 1.** Experimental design and behavioral results. (**A**) All trials started with instructions on the probability (ranging from 0% to 100%) and intensity (weak, moderate, strong) of a potentially painful electrical stimulation (1 s), followed by the addition of a countdown bar that indicated the exact moment of stimulation or omission. The duration of the countdown clock was jittered between 3 and 7 s. Then, the screen cleared and the electrical stimulation was either delivered or omitted. Most of the trials (48 trials) did not contain the anticipated electrical stimulation (omission trials). Following a delay of 4–8 s, a rating scale appeared, probing stimulation-unpleasantness on stimulation trials, and relief-pleasantness on omission trials. After 8 s, an ITI between 4 and 7 s started, during which a fixation cross was presented on the screen. The task consisted in total of 72 trials, divided equally over 4 runs (18 trials/run). Each run contained all probability (25, 50, 75) x intensity (weak, moderate, strong) combinations exactly once not followed by the stimulation (9 omission trials), three 0%-omission trials (without any intensity information), three 100%-stimulation trials (followed by the stimulation of the given intensity, one per intensity level per run), and three additional trials from the probability (25,50,75) x Intensity (weak, moderate, strong) matrix that were followed by the stimulation (per run each level of intensity and probability was once paired with the stimulation). For a detailed overview of the trials see ***Supplementary file 3*** – Trial types and numbers. (**B**) SCR were scored as the time integral of the deconvoluted phasic activity (using CDA in Ledalab) within a response window of 1–4 s post omission. Responses were square root transformed. SCR were larger following omissions of more probable and more intense US omissions (N = 26; main effect Probability: $F = 5.15$, $p < .01$; main effect Intensity, $F = 107.47$, $p < .001$) . (**C**) The pleasantness of the relief elicited by US omissions was rated on a VAS-scale ranging from 0 (neutral) to 100 (very pleasant). Omission-relief was rated as being more pleasant following omissions of more intense and more probable US (N = 31; main effect Probability: $F = 30.64$, $p < .001$; main effect Intensity: $F = 623.79$, $p < .001$). In both graphs, individual data points are presented, with the group averages plotted on top. The error bars represent standard error of the mean.

Replicating our previous findings (*Willems and Vervliet, 2021*), self-reported relief-pleasantness and omission SCR tracked the PE signal during threat omission (see *Figure 1B/C*). Overall, unexpected (non-0%) omissions of threat elicited higher levels of relief-pleasantness and omission SCR than fully expected omissions (0%), evidenced by a main effect of Probability in the 4 (Probability: 0%, 25%, 50%, 75%) x 4 (Run: 1, 2, 3, 4) LMM (For relief pleasantness (N=31): $F(3,1417)=188.34$, p<0.001, $\omega_p^2 = 0.28$; and for omission SCR (N=26): $F(3,1190)=72.90$, p<0.001, $\omega_p^2 = 0.15$, with responses to all non-0% probability levels being significantly higher than responses to 0%, p's<0.001, Bonferroni-Holm corrected). Furthermore, relief-pleasantness and omission SCR to unexpected omissions (non-0% omissions) increased as a function of instructed Probability and Intensity, in line with the first two PE axioms, evidenced by main effects of Probability (for relief-pleasantness (N=31): $F(2,1031)=30.64$, p<0.001, $\omega_p^2 = 0.05$, all corrected pairwise comparisons, p's<.005; for omission SCR (N=26): $F(2,862) = 5.15$, p<0.01, $\omega_p^2 = 0.01$, with corrected 75% to 25% comparison, p<0.01), and Intensity (for relief-pleasantness (N=31): $F(2, 1031)=623.79$, p<0.001, $\omega_p^2 = 0.55$, all corrected pairwise comparisons, p's<0.001; for omission SCR (N=26): $F(2,862.01)=107.47$, p<0.001, $\omega_p^2 = 0.20$, all corrected pairwise comparisons, p's<0.001) in a 3 (Probability: 25%, 50%, 75%) x 3 (Intensity: weak, moderate, strong) x 4 (Run: 1, 2, 3, 4) LMM. Relief-pleasantness also showed a significant Probability x Intensity interaction ($F(4,1031)=3.76$, p<0.005, $\omega_p^2 = 0.01$), indicating that the effect of probability was most pronounced for omissions of moderate stimulation (all p's<0.05). Note that while there was a general drop in reported relief pleasantness and omission SCR over time, the effects of Probability and Intensity remained present until the last run (see Appendix 5). This further confirms that probability and intensity instructions were effective until the end of the task.

## Unexpected omissions of threat trigger activations in the VTA/SN and ventral putamen, but deactivations in the vmPFC

In line with our hypothesis and similar to relief and omission SCR, unexpected (non-0%) omissions of threat elicited on average stronger fMRI activations than fully expected (0%) omissions in the VTA/SN ($t(30) = 4.48$, p<0.001, d=0.81) and left ventral putamen ($t(30) = 3.50$, p<0.005, d=0.63) ROIs (see *Figure 2A/B*). Surprisingly, NAc showed no significant change in activation ($t(30) = -0.59$, p=0.56) (*Figure 2D*), and vmPFC showed a significant deactivation ($t(30) = -4.71$, p<0.001, d=-0.85; *Figure 2C*). This apparent deactivation could indicate that omission-related responses were lower for unexpected omissions compared to expected omissions. However, it could also have resulted from lingering safety-related vmPFC activations to the 0%-instructions (corresponding to certainty that no stimulation will follow). Such safety-related vmPFC activations are indeed commonly observed during the presentation of CS- in Pavlovian fear conditioning (*Fullana et al., 2016*). To exclude this alternative hypothesis, we examined the non0%>0% contrast during the instruction window. We found no significant difference in vmPFC activation between 0% and non-0% trials, either as ROI average ($t(30) = -1.69$, $p_{unc} = 0.1$) or voxel-wise, SVC within the vmPFC mask (see Appendix 4 for full description of the anticipatory fMRI activations). This follow-up analysis suggests that the deactivation to unexpected omissions only emerged after the instruction window, and could therefore not be explained by safety-related activation that were obtained during 0% trials.

## Omission-related VTA/SN, but not striatal or vmPFC activations increased in a PE-like manner

We next assessed if the omission-related (de)activations could represent reward-like PE-signals by testing the PE axioms for each ROI separately. For axiom 1 and 2, we contrasted omissions following all intensity x probability combination with 0%-omissions and extracted ROI-specific beta averages. These beta-estimates were then entered into linear mixed models that included instructed intensity and probability as regressors of interest, and averaged US-unpleasantness as regressor of no-interest, in addition to a subject-specific intercept. Axiom 3 was tested via a one-sample (two-sided) t-test over the 100%-stimulation versus 0%-omission contrast.

We found that only omission-related **VTA/SN** activations partially fit the profile of a positive reward PE-signal (*Figure 2A*). Activations were stronger following omissions of more intense (Axiom 1, $F(2,240) = 6.14$, p<0.005, $\omega_p^2 = 0.04$), and at trend level of more probable threat (Axiom 2, $F(2,240) = 2.94$, p=0.055, $\omega_p^2 = 0.02$). However, fully predicted stimulations (100%-trials) elicited stronger

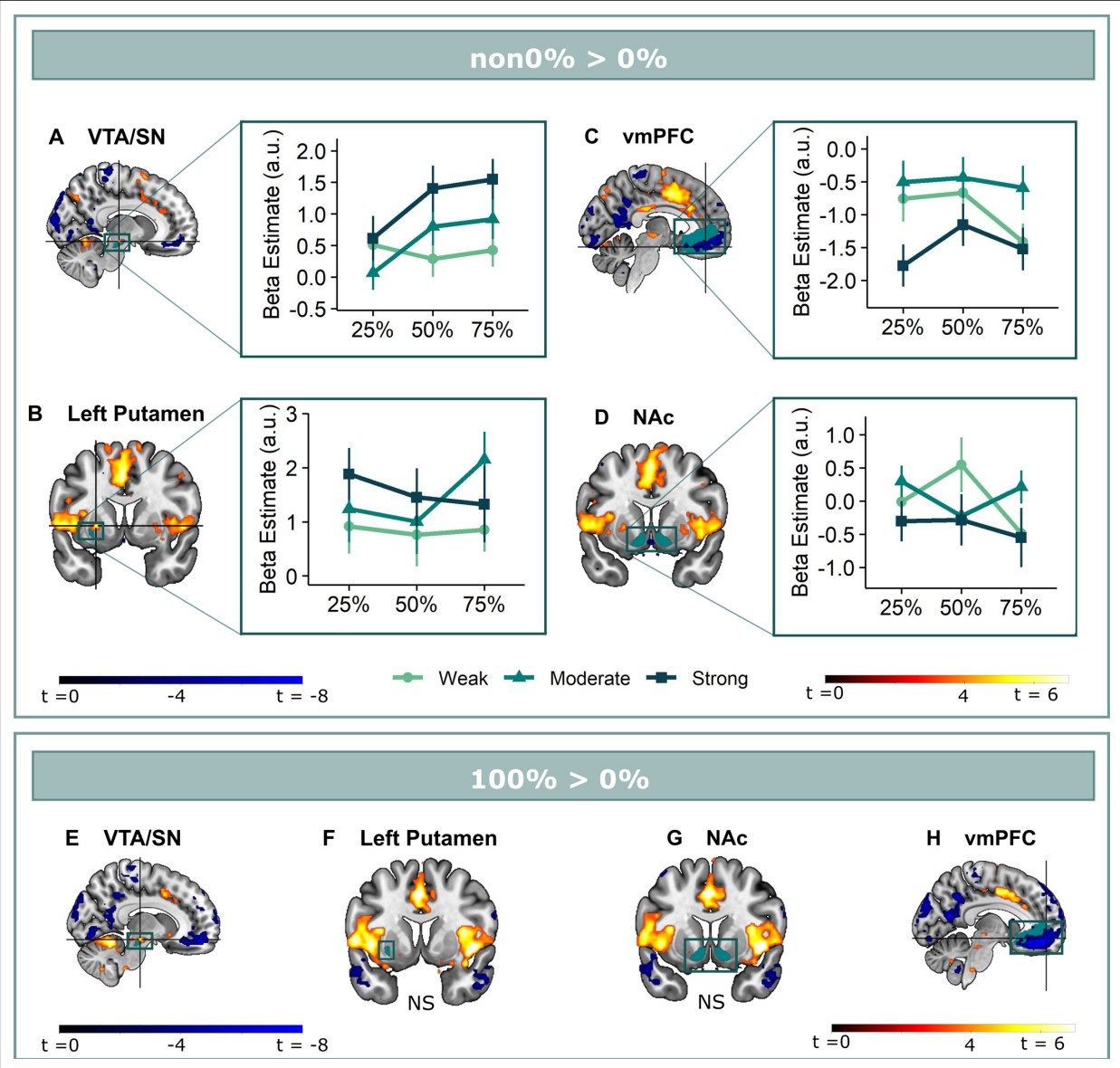

**Figure 2.** Omission-related activations in the a priori ROIs. Unexpected omissions of stimulation (non0%>0%) triggered significant fMRI responses in (**A**) the VTA/SN, and (**B**) left ventral putamen, but deactivations in (**C**) the vmPFC and no change in activation in (**D**) the NAc. Only for the VTA/SN did the activations increase with increasing Probability and Intensity of omitted stimulation (N = 31; Probability: $F = 2.94$, $p = .055$; Intensity: $F = 6.14$, $p < .005$). vmPFC responses decreased with increasing intensity of the omitted stimulation (N = 31; $F = 9.29$, $p < .001$). Fully predicted stimulations (100%) elicited stronger activations than fully predicted omission (0%) in (**E**) the VTA/SN, no difference in activation for (**F**) the left Putamen and (**G**) the NAc, and stronger deactivations for fully predicted stimulation versus omission in (**H**) the vmPFC. In all figures, the unexpected omission maps were overlayed with the a priori ROI masks (in teal) and were displayed at threshold p<0.001 (unc) for visualization purposes. The crosshairs represent the peak activation within each a priori ROI. The extracted beta-estimates in figures A-D represent the ROI averages from each non-0%>0% contrast (i.e. 25%>0%; 50%>0%; and 75%>0% for the weak, moderate, and strong intensity levels). Any positive beta therefore indicates a stronger activation in the given region compared to a fully predicted omission. Any negative beta indicates a weaker activation. The dots and error bars represent the mean and standard error of the mean.

activations than fully predicted omissions (0%-trials) ($V_{wilcoxon} = 452$, p<0.001, $r=0.72$), contradicting axiom 3 (*Figure 2E*).

Unlike previous findings *Raczka et al., 2011*; *Thiele et al., 2021*; *Esser et al., 2021*; *Leknes et al., 2011*, we found no evidence for striatal reward PE-like processing of threat omissions. While **ventral putamen** responses were stronger following unexpected omissions compared to expected omissions, these activations did not increase with increasing intensity (axiom 1, $F(2,240) = 2.29$, p=0.10) or

probability (axiom 2, $F_{(2,240)}$ = 0.57, p=0.57) (*Figure 2B*). Notably, we did find anecdotal evidence that fully predicted stimulations and fully predicted omissions triggered similar activations in the ventral putamen (axiom 3, $t_{(30)}$ = 1.23, p=0.46, BF of 2.62 in favor of the null-hypothesis, *Figure 2F*). This indicates that ventral putamen activations were exclusively triggered by unexpected threat omissions, and not by fully predicted outcomes, which is similar to a PE-signal.

In general, there was no evidence for omission-related **NAc** activations. Activations were not affected by intensity (axiom 1, $F_{(2,240)}$ = 1.88, p = 0.16) or probability instructions (axiom 2, $F_{(2,240)}$ = 0.75, p=0.48) (*Figure 2D*), and while there were no differences in activation between fully predicted stimulation and fully predicted omission ($t_{(30)}$ = 0.67, p=0.51, BF in favor of null hypothesis = 4.24), this equivalence was most likely caused by an overall absence of activation (*Figure 2G*).

Finally, omission-related **vmPFC** deactivations were stronger for omissions of more intense threat (axiom 1, $F_{(2,240)}$ = 9.29, p<0.001, $\omega_p^2$ = 0.06), but were unaffected by probability instructions (axiom 2, $F_{(2,240)}$ = 1.78, $p$ = 0.17) (*Figure 2C*). Furthermore, responses were smaller for completely predicted stimulation compared to completely predicted omission (axiom 3, $t_{(30)}$ = –8.65, p<0.001, $d$=–1.55, *Figure 2H*). Taken together, we found no evidence that vmPFC deactivations reflected a positive reward PE-like signal.

A potential explanation for the absent probability effects in the putamen and vmPFC might be that the effects were obscured by including participants who did not believe the probability instructions. Indeed, the provided instructions did not map exactly onto the actually experienced probabilities, but were all followed by stimulation in 25% on the trials (except for the 0% trials and the 100% trials). We therefore reran our analyses on a subset of participants who showed probability-related increases in their anticipatory SCR during the countdown clock (N=21, larger SCR to 75% compared to 25% instructions, see Appendix 4), which we used as a post-hoc index of actual probability-related expectancy. This subgroup analysis revealed no additional effect of Probability for the ventral putamen or the vmPFC, but it rendered the effect of Intensity for the ventral Putamen significant ($F_{(2,160)}$ = 3.10 p<0.05, $\omega_p^2$ = 0.03). In addition, it increased the effect of probability for the VTA/SN activation ($\omega_p^2$ = 0.02 to 0.05). Likewise, a post-hoc trial-by-trial analysis of the omission-related fMRI activations confirmed that the Probability effect for the VTA/SN activations was stable over the course of the experiment (no Probability x Run interaction) and remained present when accounting for the Gambler's fallacy (i.e. the possibility that participants start to expect a stimulation more when more time has passed since the last stimulation was experienced; see Appendix 5). Overall, these post-hoc analyses further confirm the PE-profile of omission-related VTA/SN responses.

## Anterior insula and dmPFC/aMCC clusters show increased activation for unexpected omissions of threat in a PE-like fashion

We then explored neural threat omission processing within a wider secondary mask that combined our primary ROIs with additional regions that have previously been associated to reward, pain and PE processing (such as the wider striatum, including the caudate nucleus, and putamen; midbrain nuclei, including the periaqueductal gray [PAG], and red nucleus; medial temporal structures, including the amygdala and hippocampus; midline thalamus, habenula and cortical regions, including the anterior insula [aINS], orbitofrontal [OFC], dorsomedial prefrontal [dmPFC], and anterior cingulate [ACC] cortices). Significant omission-processing clusters (contrast non0%>0% omissions) were extracted from the mask using a cluster-level threshold (p<0.05, FWE-corrected), following a primary voxel-threshold (p<.001) and included the bilateral anterior insula, bilateral putamen, and a medial cortical cluster encompassing parts of the (dmPFC and the anterior medial cingulate cortex aMCC; *Figure 3A–D*). The left putamen cluster bordered and minimally overlapped with our predefined ventral putamen ROI (4 out of 82 voxels) which was based on the peak PE activity in previous studies (*Raczka et al., 2011*; *Thiele et al., 2021*). Exploratory analyses within a wider whole-brain grey-matter mask identified several other omission-processing clusters (see Appendix 3). Probability and Intensity related activity modulations of these clusters can likewise be found in Appendix 3.

Follow-up analysis of the bilateral **putamen** clusters confirmed that putamen activations did not fit the profile of a positive reward PE. Activations in neither cluster increased with increasing intensity (axiom 1, left: $F_{(2,240)}$ = 0.18, p=0.83; right: $F_{(2,240)}$ = 0.06, p=0.94) nor probability of threat (axiom 2, left: $F_{(2,240)}$ = 1.41, p=0.25; right: $F_{(2,240)}$ = 0.87, p=0.42), but like for the a priori ventral Putamen

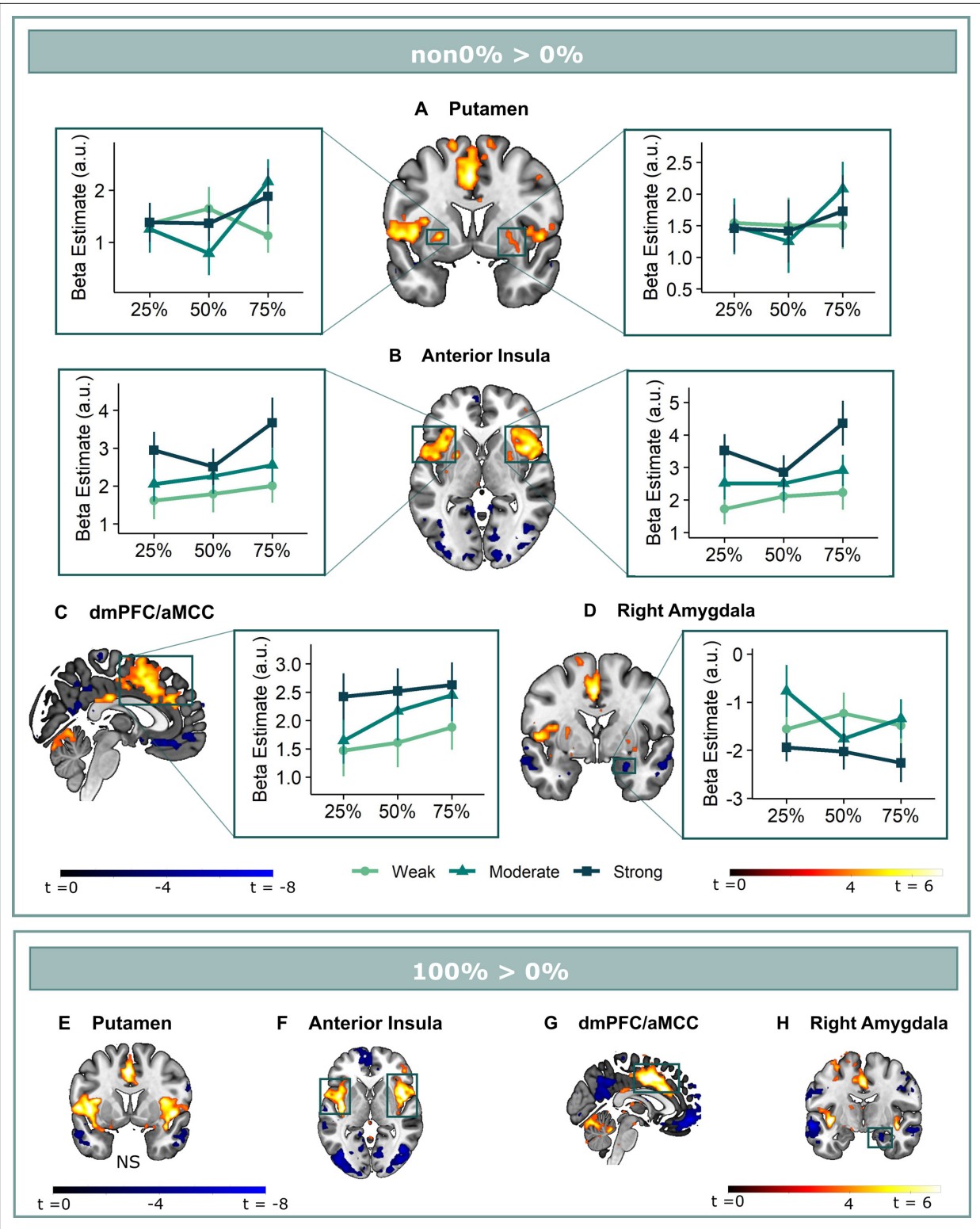

**Figure 3.** Omission-related activations in the secondary mask. We extracted unexpected omission (non0%>0%) processing clusters within our secondary mask, using a voxel threshold, p<0.001, followed by a cluster-threshold (FWE-corrected) of p<0.05. We found significant positive omission processing clusters in (**A**) the bilateral putamen, (**B**) bilateral aINS, (**C**) dmPFC/aMCC, and a trend-level negative omission processing cluster in (**D**) the right amygdala. Omission related activations in the bilateral aINS and dmPFC/aMCC increased with increasing probability (at trend-level) and intensity of omitted threat (N = 31; main effect Intensity: left aINS: $F = 8.95$, $p < 0.001$; right aINS: $F = 13.49$, $p < 0.001$; dmPFC/aMCC: $F = 6.59$, $p < 0.005$; main effect Probability: right aINS: $F_{(2,240)} = 2.78$, $p = 0.06$, dmPFC/aMCC: $F = 2.48$, $p = 0.09$), whereas amygdala activations decreased with omissions

*Figure 3 continued on next page*

*Figure 3 continued*

of increasingly intense threat (N = 31, main effect Intensity: F = 3.26, p<0.05). Nevertheless, fully predicted stimulations (100%) elicited stronger activations than fully predicted omission (0%) in (**F**) the bilateral aIns, and (**G**) the dmPFC/aMCC, and stronger deactivations in the (**H**) right amygdala, but no difference in activation in the (**E**) bilateral putamen. In all figures, the unexpected omission maps are displayed at threshold p<0.001 (unc) for visualization purposes. The extracted beta-estimates in figures A-D represent the ROI averages from each non-0%>0% contrast (i.e. 25%>0%; 50%>0%; and 75%>0% for the weak, moderate, and strong intensity levels). Any positive beta therefore indicates a stronger activation in the given region compared to a fully predicted omission. Any negative beta indicates a weaker activation. The dots and error bars represent the mean and standard error of the mean.

ROI, activations were comparable for fully predicted outcomes, especially in the left cluster (axiom 3, left: $V$=339, p=0.46, BF = 2.41, right: $t(30)$ = 1.83, p=0.46, BF = 1.20).

Positive reward PE-like responses were found in the bilateral **aINS** and **dmPFC/aMCC**, where omission-related activations were stronger following omissions of more intense (left aINS: $F(2,240)$ = 8.95, p<0.001, $\omega_p^2$ = 0.06; right aINS: $F(2,240)$ = 13.49, p<0.001, $\omega_p^2$ = 0.09; dmPFC/aMCC: $F(2,240)$ = 6.59, p<0.005, $\omega_p^2$ = 0.04), and at trend level of more probable threat (left aINS: $F(2,240)$ = 1.87, p=0.16, right aINS: $F(2,240)$ = 2.78, p=0.06, $\omega_p^2$ = 0.01; dmPFC/aMCC: $F(2,240)$ = 2.48, p=0.09, $\omega_p^2$ = 0.01). Notably, aINS and dmPFC/aMCC clusters extended beyond our predefined secondary mask, and including all adjacent above-threshold voxels (p<0.001) rendered the effects of probability significant (see Appendix 3). However, fully predicted stimulations also elicited stronger activations than fully predicted omissions (left aINS: $t(30)$ = 3.21, p<0.05, $d$=0.58, right aINS: $t(30)$ = 5.26, p<0.001, $d$=0.95, mdPFC/aMCC: $t(30)$ = 5.57, p<0.001, $d$=1.00).

Finally, in addition to the vmPFC deactivations (which fell entirely within our vmPFC mask), we found trend-level deactivations for unexpected omission in the *right amygdala* (p=0.053). These deactivations were stronger for omissions of more intense ($F(2,240)$ = 3.26, p<0.05, $\omega_p^2$ = 0.02), but not more probable threat ($F(2,240)$ = 0.43, p=0.65). Furthermore, fully predicted stimulations triggered larger deactivation than fully predicted omissions ($t$=–2.77, p=0.07, BF = 0.21).

## Omission-related activations are related to self-reported relief-pleasantness

We then examined whether omission-related fMRI activations were related to self-reported relief-pleasantness on a trial-by-trial basis. In a pre-registered analysis, we entered z-scored relief-pleasantness ratings as a parametric modulator to the omission regressor in a separate GLM that did not distinguish between the different probability x intensity levels. We found that the VTA/SN ($t(30)$ = 3.26, p<0.01, $d$=0.59) and ventral putamen ROI (at trend level, $t(30)$ = 2.22, p=0.068, $d$=0.40) were positively modulated by relief-pleasantness ratings, whereas the vmPFC ROI was negatively modulated by relief-pleasantness ratings ($V$=46, p<0.001, $r$=0.71) (**Figure 4A–D**). The positive and negative modulations indicate that stronger omission-related activations in the VTA/SN and ventral putamen, and stronger deactivations in the vmPFC were associated with more pleasant relief-reports. Likewise, the bilateral aINS ($t$>6.70, p<0.001, $d$>1.20), dmPFC/aMCC ($t(30)$ = 6.13, p<0.001, $d$=1.10), and right putamen ($t(30)$ = 4.90, p<0.001, $d$=0.88), and at trend level left putamen ($t(30)$ = 2.37, p=0.097, $d$=0.43) clusters we identified from the secondary mask were positively modulated, and right amygdala was negatively modulated by relief-pleasantness ($V$=59, p<0.001, $r$=0.67) (**Figure 4E–H**). Omission-related NAc activation was unrelated to self-reported relief-pleasantness ($V$=214, p=0.52).

## A neural signature for relief-pleasantness

The (mass univariate) parametric modulation analysis showed that omission-related fMRI activity in our primary and secondary ROIs correlated with the pleasantness of the relief. However, given that each voxel/ROI is treated independently in this analysis, it remains unclear how the activations were embedded in a wider network of activation across the brain, and which regions contributed most to the prediction of relief. To overcome these limitations, we trained a (multivariate) LASSO-PCR model (Least Absolute Shrinkage and Selection Operator-Regularized Principle Component Regression) in order to identify whether a spatially distributed pattern of brain responses can predict the perceived pleasantness of the relief (or "neural signature" of relief) (*Wager et al., 2013*). Because we used the whole-brain pattern (and not only our a priori ROIs), this analysis is completely data driven and can thus identify which clusters contribute most to the relief prediction. We trained the model using

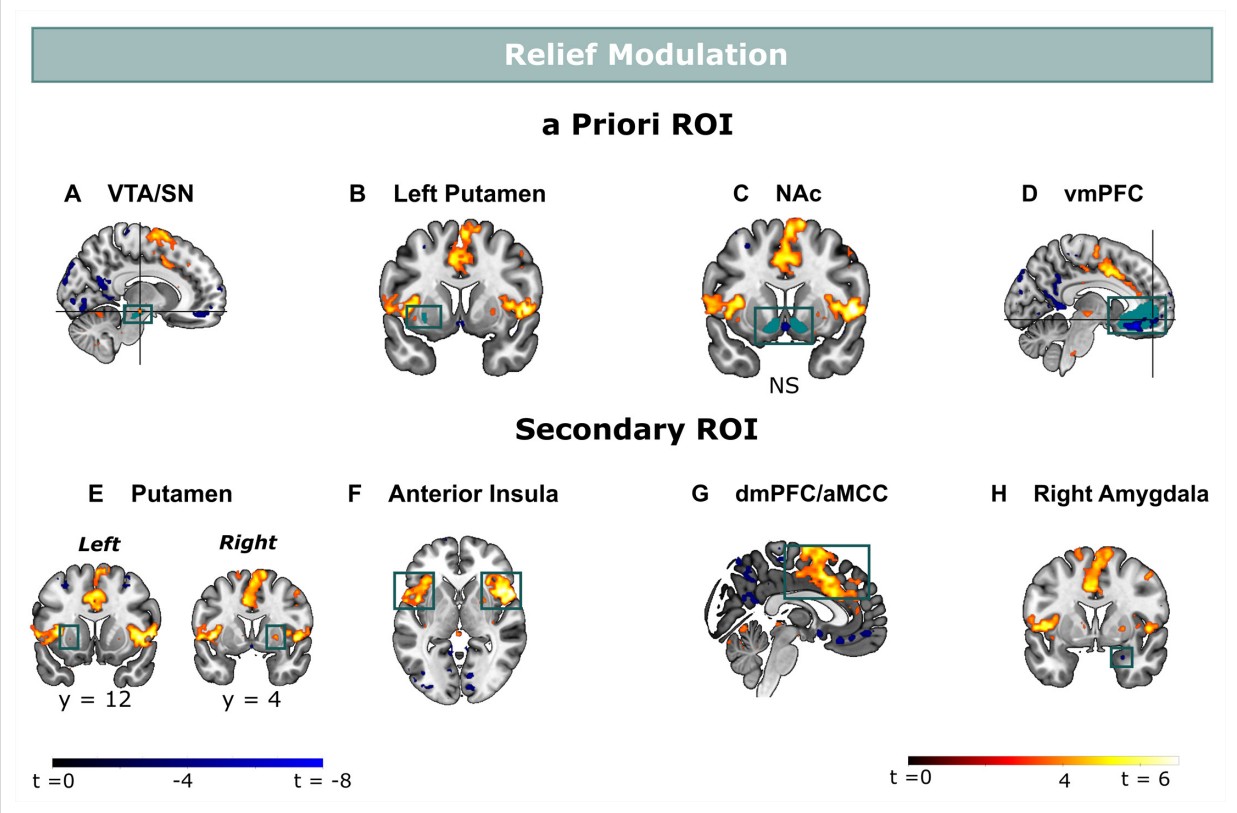

**Figure 4.** Relief modulation in the a priori and secondary ROIs. Omission-related activations were modulated by trial-by-trial levels of relief-pleasantness in (**A**) the VTA/SN, (**B**) left ventral Putamen, and (**D**) vmPFC, but not in (**C**) the NAc. Omission-related activations in the secondary ROIs were modulated by trial-by-trial levels of relief-pleasantness in (**A**) the right putamen, (**B**) bilateral aINS, (**C**) dmPFC/aMCC, and the (**D**) right amygdala. In all figures, the relief modulation maps are displayed at threshold p<0.001 (unc) for visualization purposes.

fivefold cross-validation with trial-by-trial whole-brain omission-related activation-maps as predictors, and trial-by-trial relief-pleasantness ratings as outcome.

Predicted and reported relief correlated significantly (*r*=0.28, p<0.001) (*Figure 5C*), indicating that part of the variance in reported relief-pleasantness could be explained by the neural relief signature response (*Figure 5A*). Follow-up bootstrap tests (5000 samples) identified a distributed pattern of positive and negative predictive clusters across the brain (*Figure 5B*, *Table 1*). Increased responses in these clusters predicted increased/decreased relief-pleasantness, respectively. Notably, bootstrap tests indicated that none of our a priori regions of interest significantly contributed to the signature. This was further supported by a pre-registered virtual lesion analysis where we compared the predictive performance of our LASSO-PCR model based on whole-brain data to separate models excluding voxels from our main ROIs in each iteration (see *Figure 5D*).

## Discussion

We examined whether brain reactions to unexpected omissions of threat qualify as positive reward PE signals, and explored their link with subjective relief. We showed that, similar to an unexpected reward, unexpected omissions of stimulation triggered fMRI activations within key regions of the reward and salience pathways (such as the VTA/SN, putamen, dmPFC/aMCC and aINS), and that the magnitude of these activations correlated with the pleasantness of the reported relief. Moreover, omission-related activations in the VTA/SN, the primary reward PE-encoding region in animals *Schultz, 2016*; *Watabe-Uchida et al., 2017* and humans (*D'Ardenne et al., 2008*; *Zaghloul et al., 2009*), also tracked the probability and intensity of omitted stimulation, in line with the first two criteria of a positive reward-PE signal. In contrast, the NAc and the vmPFC, two other regions that have previously been implicated in reward PE, threat omission processing, and the valuation of rewards and

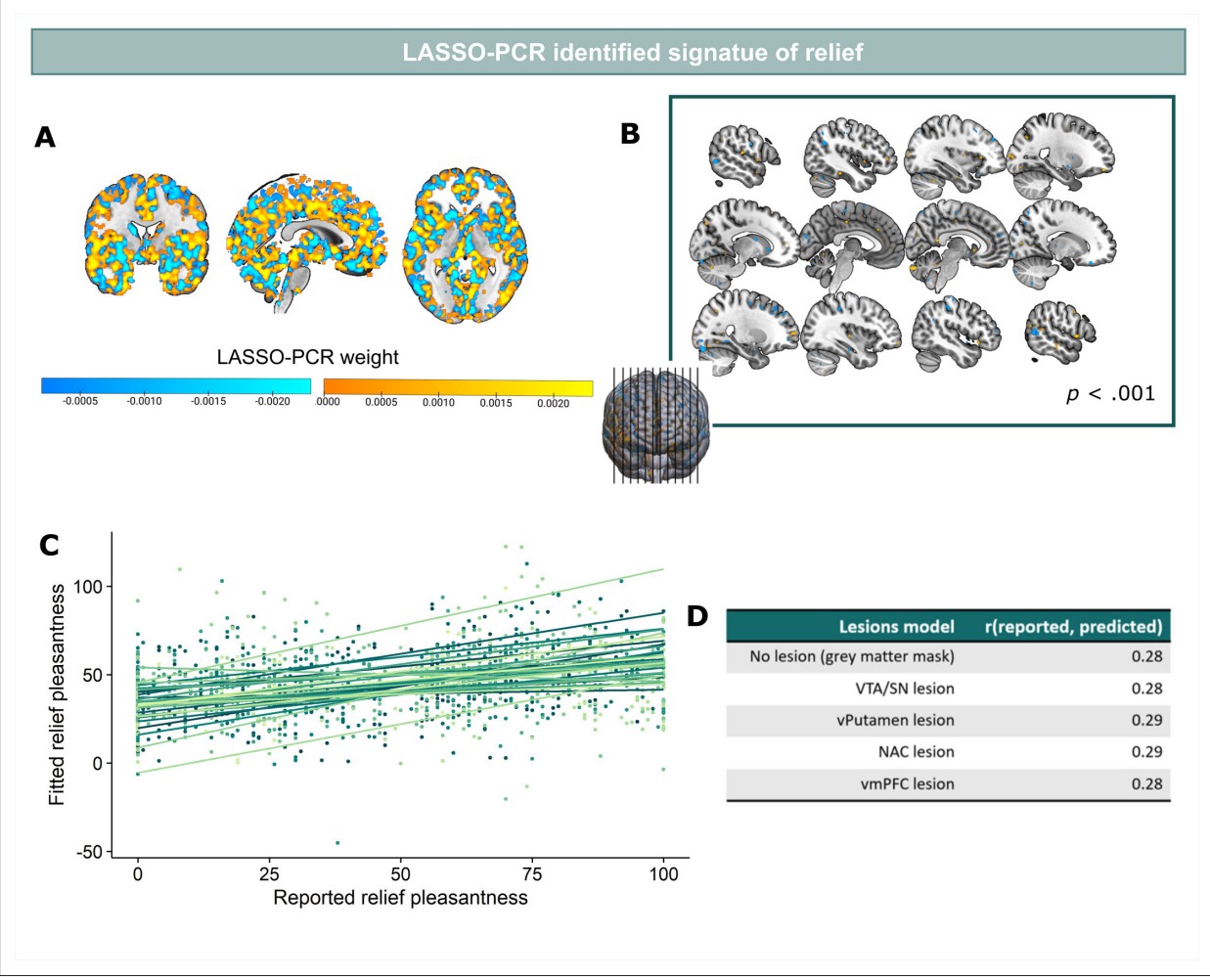

**Figure 5.** LASSO-PCR based neural signature of relief-pleasantness. (**A**) Relief predictive signature map consisting of all voxels within a grey matter mask. All weights were used for prediction. (**B**) Thresholded signature map (p<0.001), consisting of clusters that contribute significantly to the relief prediction (all clusters < 65 voxels). (**C**) Predicted and reported relief-pleasantness correlated significantly, *r*=0.28, p<0.001. Each line/color represents data of a single participant (N=31). (**D**) Correlations between reported and predicted relief in all lesion models (in each model one of the ROIs was removed from the grey matter mask). The stable correlation across models confirmed that none of our ROIs contributed significantly to the relief-predictive signature model.

relief (*Lange et al., 2020*; *Esser et al., 2021*; *Leknes et al., 2011*; *Gerlicher et al., 2018*; *Kim et al., 2006*; *Berridge and Kringelbach, 2015*), showed no (NAc) or even decreased activations (vmPFC) in response to omitted threat; and no correlation (NAc) and a negative correlation (vmPFC) with subjective relief.

Overall, the observed activity pattern of the VTA/SN supports the hypothesis that unexpected omissions of threat are processed as reward PE-like signals in the human brain. However, there are two caveats to this interpretation. First, it remains unclear whether these activations reflect the activity of dopamine cells in this region. The dopamine basis of the reward PE is well established (*Schultz, 2016*; *Watabe-Uchida et al., 2017*), and similar VTA dopaminergic responses to threat omissions have been found during fear extinction in rodents (*Luo et al., 2018*; *Salinas-Hernández et al., 2018*; *Cai et al., 2020*; *Yau and McNally, 2018*). Yet, the nature of fMRI measurements does not allow us to directly trace back the observed BOLD responses to the phasic firing of dopamine cells at the time of threat omission. Still, the general location of the omission-related VTA/SN activation is consistent with a dopaminergic basis (*Düzel et al., 2009*; *Zhang et al., 2017*), as the peak activation falls within a more medial subregion of the SN, which is predominantly composed of dopamine cells (>80% of all cells; *Root et al., 2016*). In addition, a recent human fear extinction study found that ingestion of the

**Table 1.** Main clusters contributing to the relief signature identified via bootstrapping.

**Positive weight clusters**

| L/R | Region | K | MNI Coordinates (xyz) | | | Z (peak) |
|-----|--------|---|---|---|---|---|
| R | Cerebellum Crus 2 | 17 | 6 | −85 | −30 | 0.006 |
| L | Cerebellum Crus 1 | 18 | −41 | −75 | −28 | 0.005 |
| L | Inferior orbitofrontal gyrus | 10 | −21 | 14 | −23 | 0.005 |
| R | Inferior temporal gyrus | 10 | 56 | −23 | −15 | 0.006 |
| L | Middle orbitofrontal gyrus | 15 | −25 | 54 | −15 | 0.007 |
| R | Middle orbitofrontal gyrus | 13 | 38 | 58 | −10 | 0.007 |
| R | Caudate gyrus | 12 | 4 | 14 | -1 | 0.007 |
| L | Superior temporal gyrus | 11 | −55 | 2 | 1 | 0.005 |
| R | Rolandic operculum | 19 | 58 | 10 | 3 | 0.006 |
| R | Superior occipital gyrus | 13 | 20 | −103 | 7 | 0.006 |
| L | Middle occipital gyrus | 14 | −23 | −97 | 7 | 0.006 |
| R | Calcarine gyrus | 18 | 12 | −79 | 10 | 0.007 |
| L | Middle occipital gyrus | 12 | −31 | −95 | 14 | 0.005 |
| R | Middle temporal gyrus | 19 | 54 | −47 | 14 | 0.005 |
| R | Cuneus | 12 | 4 | −79 | 25 | 0.006 |
| R | Supramarginal gyrus | 12 | 52 | −37 | 29 | 0.005 |
| L | Superior occipital gyrus | 17 | −25 | −71 | 38 | 0.004 |
| R | Precentral gyrus | 15 | 50 | 6 | 43 | 0.005 |
| L | Superior occipital gyrus | 14 | −13 | −79 | 43 | 0.004 |
| L | Mid cingulate gyrus | 27 | -9 | −23 | 45 | 0.005 |
| R | Mid cingulate gyrus | 12 | 2 | −13 | 45 | 0.005 |

**Negative weight clusters**

| L/R | Region | K | | | | |
|-----|--------|---|---|---|---|---|
| R | Cerebellum Crus 1 | 12 | 30 | −69 | −28 | −0.005 |
| L | Cerebellum Crus 1 | 11 | −17 | −89 | −19 | −0.007 |
| R | Cerebellum Crus 1 | 41 | 30 | −85 | −19 | −0.014 |
| R | Fusiform gyrus | 10 | 28 | −49 | -8 | −0.006 |
| R | Superior temporal gyrus | 10 | 52 | -2 | -6 | −0.004 |
| R | Inferior occipital gyrus | 11 | 28 | −89 | -4 | −0.006 |
| R | Superior temporal gyrus | 10 | 66 | −17 | -4 | −0.006 |
| L | Superior temporal gyrus | 16 | −63 | -5 | -1 | −0.005 |
| L | Caudate | 10 | −17 | 18 | 1 | −0.006 |
| L | Middle temporal gyrus | 14 | −57 | −65 | -1 | −0.004 |
| R | Middle occipital gyrus | 12 | 34 | −93 | 5 | −0.005 |
| L | Middle occipital gyrus | 11 | −39 | −89 | 3 | −0.004 |
| R | Middle temporal gyrus | 40 | 58 | −57 | 10 | −0.005 |
| L | Calcarine gyrus | 18 | −11 | −81 | 12 | −0.007 |
| L | Cuneus | 20 | -3 | −89 | 21 | −0.006 |
| L | Angular gyrus | 16 | −45 | −59 | 29 | −0.005 |
| L | Middle frontal gyrus | 12 | −33 | 46 | 34 | −0.005 |
| R | Postcentral gyrus | 64 | 46 | −31 | 49 | −0.005 |

*Table 1 continued on next page*

*Table 1 continued*

**Positive weight clusters**

| | | | | | | |
|---|---|---|---|---|---|---|
| R | Middle frontal gyrus | 19 | 26 | 32 | 45 | −0.004 |
| L | Postcentral gyrus | 10 | −49 | −17 | 45 | −0.005 |
| R | Middle frontal gyrus | 12 | 26 | 10 | 47 | −0.004 |

Note. Contributing clusters are defined based on voxels-wise FDR-correction with *q*<0.05, *k*>10; *L/R* indicates if the cluster (or peak of the cluster) is part of the left or right hemisphere; *Region* name is identified using the AAL atlas; *K* is the number of voxels in the cluster; *coordinates* are the MNI coordinates of cluster peak; *Z* is the weight of the cluster peak.

dopamine precursor L-Dopa increased functional coupling between NAc and VTA/SN at the time of threat omission (*Esser et al., 2021*).

A second caveat to the PE-interpretation of VTA/SN activations in the light of the present results is that fully predicted stimulations (100%) triggered stronger activations than fully predicted omissions (0%), which violates the third PE axiom and therefore opposes a PE interpretation. Theoretically, the third axiom states that a pure PE-signal would not differentiate between these fully predicted outcomes, whatever the outcomes are. As such, the violation implies that the VTA/SN responses could not represent PE-signals. Yet, we argue that this axiom should not be a decisive criterion when comparing fully predicted threat to its fully predicted omission. Specifically, a wealth of studies has reported VTA/SN responses to both salient events (even aversive) and PE (*Schultz, 2016*; *Diederen and Fletcher, 2021*), and showed that these responses might be functionally and anatomically distinct (*Zhang et al., 2017*; *Matsumoto and Hikosaka, 2009*; *Pauli et al., 2015*). Moreover, on a smaller scale, recent electrophysiological studies suggest that dopaminergic PE-signals themselves consist of two components: an initial unselective and short-lasting component that detects any event of sufficient intensity, followed by a subsequent component that codes the prediction error (*Schultz, 2016*). Thus, given that we could not control for the delivery of the stimulation in the 100%>0% contrast (the delivery of the stimulation completely overlapped with the contrast of interest), it is impossible to disentangle responses to the salience of the stimulation from those to the predictability of the outcome. A fairer evaluation of the third axiom would require outcomes that are roughly similar in terms of salience. When evaluating threat omission PE, this implies comparing fully expected threat omissions following 0% instructions to fully expected absence of stimulation at another point in the task (e.g. during a safe intertrial interval).

Beyond the midbrain, unexpected omissions of threat also elicited striatal activations that spread across the bilateral putamen (as in *Raczka et al., 2011*; *Thiele et al., 2021*). Although these activations did not increase with increasing probability and intensity of the omitted stimulation, they were correlated with the reported relief-pleasantness and only occurred when the omission was unexpected (not when the outcome was fully expected, 100%-stimulation or 0% omission). Yet, in contrast to our predictions, the activations did not extend to the NAc. This was surprising, because the NAc is the main striatal projection area of the VTA, and numerous rodent and human studies have attributed reward PE signaling to this region (*Rutledge et al., 2010*; *Hart et al., 2014*). Likewise, two human studies found that self-reported relief and modeled PE-estimates at the time of threat omission covaried with NAc activations (*Esser et al., 2021*; *Leknes et al., 2011*).

Nevertheless, a growing body of research now indicates that human PE encoding is not confined to the NAc, but instead spreads across the striatum. Meta-analyses on reinforcement learning in humans have identified the (left) putamen, and not the NAc as the most consistent reward PE-encoding region across studies (*Garrison et al., 2013*; *Chase et al., 2015*). Arguably, this emerging functional divergence between rodent and human striatal responses may be linked to neuroanatomical differences at the level of the midbrain. Specifically, where the (medial) VTA and its NAc projections have revealed marked omission-processing in rodents, we found the strongest omission-related activations in the (medial) SN and the putamen, which is a SN projection target. In addition, task-related differences might have contributed to the absent NAc activations. One of the meta-analyses found that the NAc was especially involved in PE-responses when a response is required to obtain the reward/threat omission (e.g. in instrumental tasks; *Garrison et al., 2013*). Arguably, as the EVA task does not allow active

control over the omission, it might not have engaged the NAc. Together, our findings call for caution when directly extrapolating rodent findings to human fMRI results.

We found omission-related *deactivations* in the vmPFC that were stronger following omissions of more intense threat, and that correlated with the reported relief. This is again in contrast with our predictions, and with previous studies that showed vmPFC activations during early US omissions in the context of fear extinction *Lange et al., 2020* and positive associations between vmPFC activity and subjective pleasure (*Kim et al., 2006*; *Berridge and Kringelbach, 2015*). Instead, we found vmPFC deactivations for both omissions and stimulations. Interestingly, these deactivations were not limited to the vmPFC, but spanned across key regions of the default mode network (*Raichle, 2015*), such as the PCC and precuneus (see Appendix 3). Furthermore, they were accompanied by widespread activations in key regions of the salience network (*Seeley, 2019*; such as the VTA/SN, striatum, aINS, dmPFC/aMCC). One potential explanation is therefore that the deactivation resulted from a switch from default mode to salience network, triggered by the salience of the unexpected threat omission or by the salience of the experienced stimulation.

In addition to examining the PE-properties of neural omission responses in our a priori ROIs, we trained a LASSO-PCR model to establish a signature pattern of relief. One interesting finding that only became evident when we compared the univariate and multivariate approach was that none of our a priori ROIs appeared to be an important contributor to the multivariate neural signature, even though all of them (except NAc) were significantly modulated by relief in the univariate analysis. Instead, we identified a spatially distributed pattern of brain responses that consisted of several small clusters (all <65 voxels) across the brain. Some of these clusters fell within other important valuation and error-processing regions in the brain (e.g. OFC, MCC, caudate nucleus). However, all were small (all <28 voxels) and require further validation in out of sample participants. Still, these data-driven maps suggest that other regions than our ROIs might have been especially important for the emotional experience of relief and that examining these multivariate patterns can aid our understanding of emotional relief.

Finally, two limitations of the study need to be addressed. First, by aiming to provide a fine-grained analysis of the reward PE-properties of human fMRI responses to threat omission, we focused exclusively on the necessary and sufficient requirements of reward PE signaling, and thereby disregarded another core aspect of PE: its teaching property. In the EVA task expectancies are instructed and all learning is explicitly discouraged. As a result, this task assesses PE completely outside of a learning context. It therefore remains unclear how the PE-signals we observed relate to actual learning. It could for instance be that the observed responses mainly reflected the surprisingness of the outcome, independent of subsequent learning. It therefore remains important to study how the activity patterns of PE-encoding regions are related to expectancy updating in learning paradigms. Furthermore, it would be interesting to see how the neural omission-PE signals are affected in clinical populations. Second, although single unit recordings in rodents have revealed clear PE-like phasic increases in the firing of dopamine cells at the time of threat omission (*Luo et al., 2018*; *Salinas-Hernández et al., 2018*; *Cai et al., 2020*), one could argue that fMRI measurements cannot capture the same sub-second response. Indeed, it is generally difficult to disentangle prediction *error* responses from the immediately preceding *prediction responses* in fMRI paradigms (*Linnman et al., 2011*). Nevertheless, there was no multicollinearity between anticipation and omission regressors in the first-level GLMs (Variance Inflation Factor, VIF <4), making it unlikely that the omission responses purely represented anticipation. Still, because of the slower timescale of fMRI measurements, we cannot conclusively dismiss the alternative interpretation that we assessed (part of) expectancy instead.

In conclusion, by violating instructions about the probability and intensity of a potentially painful stimulation, we found widespread activations in the reward and salience pathways that were furthermore related to PE-like feelings of pleasurable relief. But, more importantly, we showed for the first time in humans that unexpected threat omission triggered VTA/SN activations that partially met the necessary and sufficient criteria of a positive reward PE signal. In doing so, we provided an important missing link for the human translation of the reward-like threat omission processing in rodents.

## Methods

### Participants

To ensure sufficient statistical power for our fMRI analyses, we decided to recruit n=30 healthy volunteers, a commonly used sample size for fMRI research. Notably, power calculations confirmed that this sample size exceeded the required sample size for replicating the relief and SCR effects we observed in the behavioral validation study (*Willems and Vervliet, 2021*). Specifically, effect sizes for the main effects of Probability (within the Probability x Block RM-ANOVA) and Intensity (within Probability x Intensity x Block RM-ANOVA) in this validation study were $\eta_p^2=0.62$, $\varepsilon=0.52$ and $\eta_p^2=0.81$, $\varepsilon=0.62$ for relief-pleasantness ratings, and $\eta_p^2=0.72$ and $\eta_p^2=0.48$, $\varepsilon=0.67$ for omission-induced SCR, respectively. Our calculations (G*power software, selecting F tests, ANOVA: repeated measures, within factors, for 4 levels of probability and 3 levels of intensity) revealed that for relief-pleasantness, a sample size of n=9, and for omission SCR a sample size of n=14 was sufficient to replicate the probability and intensity effects we observed in the subjective and physiological data with a power of 0.95 and an alpha of 0.05.

In the end, 31 healthy volunteers between the ages of 18 and 25 (mean = 20.65, 19 females) and with a body-mass index (BMI) of 18.5–25 kg/m² were recruited to participate in our study. All were right-handed and non-smoking, and declared to be free of any serious medical disorder (including neurological, cardiovascular or respiratory disorders, hypertension, migraine, head trauma with loss of consciousness, chronic (duration of >3 months) or acute pain) or psychiatric disorder (including clinical depression, anxiety, psychotic disorders, mood disorders, eating disorders, somatoform disorders, substance-related disorders or any other psychiatric disorder). Furthermore, participants were excluded when they had to regularly take medication (e.g. treatment in the last 6 months with antidepressants, antipsychotics, sedative hypnotics, psychostimulants, glucocorticoids, appetite suppressants, estrogens, opiates such as pain medication and coughing syrup with codeine, or dopaminergic medications); when they regularly consumed alcohol (>10 units/week), caffeine (>1000 ml coffee daily or equivalent), or energy drinks (>1 drink/day); when their medical doctor requested them to stay away from stressful situations; when they had used cannabis or any other drug of abuse in the last year; or when there was any other contraindication for an MRI scan (e.g. cochlear implant, cardiac pacemaker, neural stimulator, metallic body inclusion or any other metal (implanted) in the body which may interfere with MRI scanning; claustrophobia or severe back problems that may interfere with complying to scanning procedures). Female participants were furthermore required to take hormonal contraceptives or to be tested in the follicular phase of their menstrual cycle (and not to be pregnant or lactating). An overview of the demographics and questionnaire scores of the included participants is provided in *Supplementary file 1* – Demographics of included participants, and *Supplementary file 2* – Descriptives of questionnaire scores. Upon their enrollment, participants were furthermore asked to refrain from consuming any alcohol and/or caffeine and exerting any recreational physical exercise in the 24 hr before the scan session. The study was approved by the Ethical committee UZ/KU Leuven (S63852). All participants provided written informed consent and received either partial course credits or a monetary compensation for their participation.

### Stimuli

#### Expectancy Violation Assessment (EVA) task

The task was an fMRI adaptation of the previously validated EVA task (*Willems and Vervliet, 2021*) and was programmed in affect5 software (*Spruyt et al., 2009*). In this task, probability (0%, 25%, 50%, 75%, 100%) and intensity (weak, moderate, strong) information of an upcoming electrical stimulation to the wrist was presented on each trial in the upper left and right corner of the screen respectively. A countdown clock, visualized as a receding bar, indicated the exact moment of stimulation or omission. Responses to the omissions/stimulations were measured.

#### Electrical stimulation

The electrical stimulation consisted of a single 2ms electro-cutaneous pulse, generated by a Digitimer DS8R Bipolar Constant Current Stimulator (Digitimer Ltd, Welwyn Garden City, UK), and delivered via two MR-compatible EL509 electrodes (Biopac Systems, Goleta, CA, USA). Electrodes were filled with Isotonic recording gel (Gel 101; Biopac Systems, Goleta, CA, USA), and were attached to the right

wrist. To match the instructions, a total of three intensities were individually selected at the start of the scanning session. The weak stimulus was calibrated to be 'mildly uncomfortable', the moderate stimulus to be 'very uncomfortable, but not painful', and the strong stimulus to be 'significantly painful, but tolerable'. Selected Weak ($M$=9.35, $SD$ = 4.65), moderate ($M$=16.26 mA, SD = 8.13), and strong ($M$=34.26, SD = 19.53) differed significantly (Friedman test: $\chi 2(2)$=62, p<0.001, W=1, all Bonferroni-Holm corrected pairwise comparisons, p<0.001).

## Experimental procedure

Participants were invited to the lab for an intake session, during which exclusion criteria were extensively checked, experimental procedures were explained, and informed consent was obtained. All included participants then filled out the questionnaire battery (see *Supplementary file 2* – Descriptives of questionnaire scores) and were familiarized with the task and rating scales.

At the start of the scanning session, participants were fitted with skin conductance and stimulation electrodes and stimulation intensities were calibrated using a standard workup procedure. Specifically, participants were presented with a range of increasingly intense stimulations and were asked to rate their unpleasantness on a scale from 0 (no sensation) to 10 (extreme, intolerable pain), with a rating of 1 corresponding to 'clear sensation', a rating of 3 to a 'mildly uncomfortable sensation', a rating of 5 to a 'very uncomfortable, but not painful sensation', a rating of 6 to 'faint pain', a rating of 7 to 'pain' and a rating of 8 to 'significant, but tolerable pain'. While participants were asked to select a stimulus that was significantly painful, but tolerable (rated as an 8), the researcher additionally selected two other intensities corresponding to a rating of 3 and 5. Participants were then prepared for the scanner and task instructions were repeated. It was emphasized that all trials in the EVA task were independent of one another, meaning that the presence/absence of stimulations on previous trials could not predict the presence/absence of stimulation on future trials. Finally, the selected intensities were presented again and recalibrated if necessary.

In total, the EVA task comprised 72 trials (48 omission trials), divided equally over four runs of 18 trials/run (for a schematic overview, see *Supplementary file 3* – Trial types and numbers). Since we were mainly interested in how omissions of threat are processed, we wanted to maximize and balance the number of omission trials across the different probability and intensity levels, while also keeping the total number of presentations per probability and intensity instruction constant. Therefore, we crossed all non-0% probability levels (25, 50, 75, 100) with all intensity levels (weak, moderate, strong; 12 trials). The three 100% trials were always followed by the stimulation of the instructed intensity, while stimulations were omitted in the remaining nine trials. Six additional trials were intermixed in each run: Three 0% omission trials with the information that no electrical stimulation would follow (akin to 0% Probability information, but without any Intensity information as it does not apply); and three trials from the Probability x Intensity matrix that were followed by electrical stimulation (across the four runs, each Probability x Intensity combination was paired at least once, and at most twice with the electrical stimulation). Note that, based on previous research, we did not expect the inconsistency between the instructed and perceived reinforcement rate to have a negative effect on the Probability manipulation (see Appendix 5). Within each run trials were presented in a pseudo-random order, with at most two trials with the same intensity or probability as the previous trial.

All trials started with the Probability and Intensity instructions for 1 s, followed by the addition of the countdown bar to the middle of the screen, counting down for 3–7 s (see *Figure 1A*). Then, the screen cleared and the electrical stimulation was either delivered or omitted. Following a delay of 4–8 s (during which skin conductance and BOLD responses to the omission were measured), the rating scale appeared (probing shock-unpleasantness on shock trials and relief-pleasantness on omission trials). The scale remained on the screen for 8 s or until the participant responded, followed by an intertrial interval between 4 and 7 s during which only a fixation cross was shown. Note that all phases in the trial were jittered (duration countdown clock, duration outcome window, duration intertrial interval). After the last run, participants were asked some control questions regarding the intensity and probability instructions. Specifically, they were asked how much effort they would exert to prevent future weak/moderate/strong stimulation (from 0 'no effort' to 100 'a lot of effort'); and to estimate how many stimulations they thought they received following each probability instruction.

## Subjective ratings

Relief-pleasantness and shock-unpleasantness were probed on omission-and shock-trials using Visual Analogue Scales (VAS) ranging from 0 (neutral) to 100 (very pleasant/unpleasant).

## Skin conductance responses (SCR)

Fluctuations in skin conductance were measured between two disposable, EL509 electrodes filled with Isotonic recording gel (Gel 101; Biopac Systems, Goleta, CA, USA) that were attached to the hypothenar palm of the left hand. Data were recorded continuously at a 1000 Hz sampling rate via a Biopac MP160 System (Biopac Systems, Goleta, CA, USA), and Acqknowledge software (version 5.0). Raw data were low-passed filtered at 5 Hz (Butterworth, zerophased) and downsampled to 100 Hz in Matlab (version R2020b), after which they were entered into a continuous decomposition analysis (CDA) with two optimization runs (Ledalab, version 3.4.9 *Benedek and Kaernbach, 2010*). Skin conductance responses (SCRs) were scored as the time integral of the deconvoluted phasic activity (integrated SCR) within response windows of 1–4 s after the onset (anticipatory SCR) and the offset (omission/stimulation SCR) of the countdown clock. Above-threshold responses (>0.01 μS) were square root transformed to reduce the skewness of the distribution. N=3 participants had incomplete datasets because of a missing run (N=2) or delayed recording (N=1), and data of N=5 were completely excluded for SCR analyses as a result of data loss due to technical difficulties (N=1) or because they were identified as anticipation non-responder (i.e. participant with smaller average SCR to the clock on 100% than on 0% trials; N=4). This resulted in a total sample of N=26 for all SCR analyses. For all other analyses, the full data set (N=31) was used (see *Supplementary file 1* – Demographics of included participants).

## Statistical analyses of ratings and SCR

Rating and SCR data were analyzed in R 4.2.1 (R Core Team, 2022; https://www.R-project.org) via linear mixed models that were fit using the lme4 package (v1.1.29 *Bates et al., 2015*). All models included within-subject factors of Intensity and/or Probability and their interaction as fixed effect and a subject-specific intercept as random effect. The Intensity factor always contained 3 levels (weak, moderate, strong). The Probability factor depended on the outcome variable. For models of relief and omission SCR that looked at Probability as well as Intensity, Probability had 3 levels (25%, 50%, 75%); but for models of relief and omission SCR that only assessed Probability, Probability had 4 levels (0%, 25%, 50%, 75%). To account for potential changes in the effects of Probability (and Intensity) over time, models included Run (4 levels: 1, 2, 3, 4) and their interaction with Probability (and Intensity) as regressor of no-interest. In addition, we controlled for individual differences in the perceived unpleasantness of the stimulation, by calculating the average of the reported stimulation unpleasantness across the entire task, and entering the resulting (mean-centered) scores as a between-subjects covariate. Inclusion of both regressors of no-interest indeed increased model fit (lower AIC). Model assumptions were checked and influential outliers were identified. Influential outliers were defined as data points above $q_{0.75}+1.5 * IQR$ or below $q_{0.25}–1.5*IQR$, with IQR the inter quartile range and $q_{0.25}$ and $q_{0.75}$ corresponding to the first and third quartile, respectively; and with a cook's distance of greater than 4/number of data points (calculated via influence.ME package in R, v0.9–9 *Nieuwenhuis et al., 2012*). To reduce the influence of these data points, they were rescored to twice the standard deviation from the mean of all data points (corresponding approximately to either 0.05 and 0.95 percentile). If results did not change, we report the model including the original data points. If results changed, we report the model with adjusted data points.

Main and interaction effects were evaluated using *F*-tests and p-values that were computed via type III analysis of Variance using Kenward-Rogers degrees of freedom method of the lmerTest package (v3.1.3 *Kuznetsova et al., 2017*), and omega squared were reported as an unbiased estimate of effect size (calculated via effectsize package, v0.7.0 *Ben shachar et al., 2020*). All significant effects were followed up with Bonferroni-Holm corrected pairwise comparisons of the estimated marginal means in order to assess the direction of the effect (emmeans package, v1.7.5, Length, 2022). Results related to the regressors of no-interest, the anticipatory SCR, the post-experimental questions, and the stimulation responses are reported in the Appendix 4 and 5 for completeness.

## fMRI analyses

### MRI acquisition

MRI data were acquired on a 3 Tesla Philips Achieva scanner, using a 32-channel head coil, at the Department of Radiology of the University Hospitals Leuven. The four functional runs (226 volumes each) were recorded using an T2*-weighted echo planar imaging sequence (60 axial slices; FOV = 224 x 224 mm; in-plane resolution = 2 × 2 mm; interslice gap = 0.2 mm; TR = 2000ms; TE = 30ms; MB = 2; flip angle = 90°). In addition, a high-resolution T1-weighted anatomical image was acquired for each subject for co-registration and normalization of the EPI data using a MP-RAGE sequence (182 axial slices; FOV = 256 x 240 mm; in-plane resolution = 1 x 1 mm; TR = shortest, TE = 4.6ms, flip angle = 8°); and a short reverse phase functional run (10 volumes) was acquired using the exact same imaging parameters as the functional runs, but with opposite phase encoding direction. This reverse phase run was used to estimate the B0-nonuniformity map (or fieldmap) to correct for susceptibility distortion. Functional data of one run were missing for N=4 participants as a result of technical difficulties during scanning. Whenever available, rating and SCR data was still included in the analyses.

### fMRI Preprocessing

Prior to preprocessing, image quality was visually checked via quality reports of the anatomical and functional images generated through MRIQC (*Esteban et al., 2017*). MRI data were then preprocessed using a standard preprocessing pipeline in fMRIPrep 20.2.3 (*Esteban et al., 2019*). A detailed overview of the preprocessing steps in fMRPrep can be found in Appendix 2. In line with our preregistration, spikes were identified and defined as volumes having a framewise displacement (FD) exceeding a threshold of 0.9 mm or DVARS exceeding a threshold of 2. No functional run had more than 15% spikes, and hence none of the runs had to be excluded from our analyses based on our preregistered criterium. Afterwards, the functional data were spatially smoothed with a 4 mm FWHM isotropic Gausian kernel within the 'Statistical parametric Mapping' software (SPM12; https://www.fil.ion.ucl.ac.uk/spm/).

### Subject level analysis

Three subject-level general linear models (GLM) were specified. In all models, we concatenated the functional runs and added run-specific intercepts to account for changes over time. The first model investigated the effects of instructed Probability and Intensity on neural omission processing and therefore included separate stick regressors (duration = 0) for omissions of each Probability x Intensity combination (10 regressors), and stimulations (2 regressors: one for non-100% stimulations, one for 100% stimulations), in addition to boxcar regressors (duration = total duration of event) for the instruction (1 regressor) and rating (1 regressor) windows. The second model assessed how omission fMRI data was related to trial-by-trial relief-pleasantness ratings, by only including a single stick regressor for all omissions (1 regressor) and shocks (1 regressor), and boxcar regressors for instructions (1 regressor) and ratings (1 regressor). Z-scored relief-pleasantness ratings were added as parametric modulator for the omission regressor. A final model estimated single-trial omission responses (48 stick regressors), in addition to a single stick regressor for stimulation and boxcar regressors for instructions (1 regressor) and ratings (1 regressor) and was used for the LASSO-PCR analyses. Regressors in all models were convolved with a canonical hemodynamic response function and a high pass filter of 180 s was applied to remove low-frequency drift. Additional non-task related noise was modeled by including nuisance regressors of no-interest for global CSF signal, 24 head motion regressors (consisting of 6 translation and rotation motion parameters and their derivatives, z-scored; and their quadratic terms), and dummy spike regressors.

## Group level univariate analysis

### Omission processing

To test whether our regions of interest (ROI) were activated by the unexpected omission of threat, we contrasted all non-0% omissions (unexpected omissions) with 0% omissions (expected omissions) at subject-level. Mean activity of each ROI was extracted from the resulting contrast map through marsbar (v0.45 *Brett et al., 2025*), and was entered into group-level (two-sided) one-sample *t*-tests

(per ROI) that were Bonferroni-Holm corrected for the total number of ROIs (4 in the main analyses, 10 in the secondary analyses) in R.

We then evaluated whether the observed activity qualified as a 'Prediction Error' by applying an axiomatic testing approach. Specifically, we tested for each ROI if the omission-related activity increased with increasing expected Intensity (axiom 1) and Probability (axiom 2) of threat, and whether completely predicted outcomes (0% omission and 100% stimulations) elicited equivalent activation (axiom 3). For axioms 1 and 2, we extracted mean activity of each ROI from separate omission contrasts that contrasted each omission type (i.e. all possible Probability x Intensity combinations) separately with 0% omissions. These were then entered into a LMM in R including Probability (3 levels: 25%, 50%, 75%) and Intensity (3 levels: weak, moderate, strong) as within-subject factors, and a between-subject covariate of average stimulation unpleasantness (mean-centered) as fixed effects, in addition to a subject-specific intercept. Note that we did not include Run as regressor of no-interest, as Run effects were already accounted for by adding run-specific intercepts to the first level models. Main and inter-action effects within each model were followed up with Bonferroni-Holm corrected pairwise comparisons of the estimated marginal means in order to assess the direction of the effect. Finally, to fulfill all necessary and sufficient requirements of a prediction error signal, we contrasted completely predicted omissions (0%) with completely predicted stimulations (100%), as these should trigger equivalent activation in PE-encoding regions. Given that we would expect equivalent activation, Bayes Factors were computed using the BayesFactor package in R (v0.9.12–4.4, Morey & Rouder, 2022) to compare the evidence in favor of alternative and null hypotheses. Larger Bayes Factors indicated more evidence in favor of the null hypotheses.

## Parametric modulation of relief

We then tested if the omission-related activity was correlated to self-reported relief-pleasantness on a trial-by-trial basis. To this end, we extracted the mean activity from the modulation contrast for each ROI, and entered these averages in separate one-sample (two-sided) t-test in R, again correcting for the number of ROIs (4 for main analysis, 10 for secondary analyses). In a subsequent exploratory analysis, we entered z-scored omission SCR-responses as parametric modulator to the omission regressor. Results related to this analysis are reported in Appendix 6.

## Neural signature of relief

A LASSO-PCR model (Least Absolute Shrinkage and Selection Operator-Regularized Principal Component Regression) as implemented in CANlab neuroimaging analysis tools (see https://canlab.github.io/) was trained using trial-by-trial whole-brain omission-related activation-maps as predictors, and trial-by-trial relief-pleasantness ratings as outcome (for other applications of this approach, see e.g. *Wager et al., 2013*; *Wager et al., 2011*; *Speer et al., 2023*; *Zhou et al., 2021*). The added value of this machine-learning technique is that relief is predicted across a set of voxels instead of being predicted for each voxel separately (as in standard univariate regression). Specifically, while each voxel in the activation maps is considered a predictor of relief, the LASSO-PCR technique uses a combination of Principle Component Analyses (PCA) and LASSO-regression to (1) group predictive information across individual voxels into larger components (PCA), and (2) maximize the predictive weight of the most informative components by shrinking the regression weight of the least informative components to zero (LASSO regression). Here, we embedded the LASSO-PCR technique within a fivefold cross-validation loop that iteratively trained a LASSO-PCR model in each loop on a different training and validation dataset, which we then averaged across loops to obtain the final model. Model performance was then assessed by calculating the Pearson correlation between reported and model predicted relief. Important features to the signature pattern were identified using bootstrap tests (5000 samples). Furthermore, the contribution of our ROIs to the model's performance was assessed using virtual lesion analysis that consisted of repeating the model training, but excluding voxels within the ROIs and assessing model performance. Note that we also estimated a neural signature of omission SCR, by applying a LASSO-PCR model to predict omission-related SCR responses. Results related to these analyses are reported in Appendix 6.

## Regions of interest (ROI)

Our main ROI consisted of key regions of the reward and (relief) valuation pathways such as the Ventral Tegmental Area (VTA) /Substantia Nigra (SN), Nucleus Accumbens (NAC), ventral Putamen, and ventromedial prefrontal cortex (vmPFC). The VTA/SN mask was obtained from Esser and colleagues (*Esser et al., 2021*), but was originally defined by *Bunzeck and Düzel, 2006*. The ventral Putamen ROI was defined as a 6 mm sphere centered around the peak voxel (MNI coordinates: –32,8–6) in the left ventral putamen identified by *Raczka et al., 2011*, as in *Thiele et al., 2021*. However, we overlayed this sphere with a Putamen mask obtained from a high-resolution anatomical atlas for subcortical nuclei defined by *Pauli et al., 2018* to assure the mask was restricted to voxels of the putamen and did not extend to the adjacent anterior Insula. Likewise, a bilateral NAc mask was obtained from the *Pauli et al., 2018* atlas. The vmPFC mask was obtained by selecting specific parcels from the atlas vmPFC cortex of AFNI (area 14, 32, 24, bilateral). Since we consider the pleasantness of relief from omission of a threat a type of reward, the selection of parcels was made on the base of an activation likelihood estimation (ALE) meta-analysis of 87 studies (1452 subjects) comparing the brain responses to monetary, erotic and food reward outcomes (*Sescousse et al., 2013*).

In addition to our main ROIs, we specified a wider secondary mask that extended our main ROIs with additional regions that have previously been associated to reward, pain and PE processing (such as the wider striatum, including the nucleus caudatus, and putamen; midbrain nuclei, including the periaqueductal gray [PAG], habenula, and red nucleus; limbic structures, including the amygdala and hippocampus; midline thalamus and cortical regions, including the anterior insula [aINS], medial orbitofrontal [OFC], dorsomedial prefrontal [dmPFC] and anterior cingulate [ACC] cortices). Masks for these regions were obtained from CANlab combined atlas (2018) (see https://canlab.github.io/), and Pauli atlas (*Pauli et al., 2018*). Whole brain voxel-wise analyses were restricted to the grey matter mask sparse (from CANlab tools), extended with midbrain voxels (including VTA/SN, RN, habenula). All masks were registered to functional space before analyses and are freely available online at OSF (https://osf.io/ywpks/).

## Code availability

Analyses code and anatomical masks are freely available at OSF (https://osf.io/ywpks/).

## Acknowledgements

This research was supported by FWO project grant G078929N (National Research Fund Flanders, Belgium) and C1 project grant C16/19/002 (KU Leuven, Belgium) awarded to BV. ALW was supported by the Internal Funds KU Leuven. LVO is a research professor funded by the KU Leuven Special Research Fund. We would like to thank Mathijs Franssen and Ronald Peeters for their technical support; and Silvia Papalini, Anraoi Rooney, Lieselotte Claes and Anamarija Banjac for their assistance during fMRI data collection.

## Additional information

### Funding

| Funder | Grant reference number | Author |
| --- | --- | --- |
| Fonds Wetenschappelijk Onderzoek | G078929N | Bram Vervliet |
| KU Leuven | C16/19/002 | Bram Vervliet |
| KU Leuven | | Anne L Willems<br>Lukas Van Oudenhove |

The funders had no role in study design, data collection and interpretation, or the decision to submit the work for publication.

## Author contributions
Anne L Willems, Conceptualization, Data curation, Formal analysis, Investigation, Visualization, Methodology, Writing – original draft; Lukas Van Oudenhove, Conceptualization, Formal analysis, Methodology, Writing – review and editing; Bram Vervliet, Conceptualization, Data curation, Supervision, Funding acquisition, Methodology, Writing – review and editing

## Author ORCIDs
Anne L Willems ⓘ https://orcid.org/0000-0003-2234-8916

## Ethics
The study was approved by the Ethical committee UZ/KU Leuven (S63852). All participants provided written informed consent.

Reviewer #1 (Public Review): https://doi.org/10.7554/eLife.91400.4.sa1
Reviewer #2 (Public Review): https://doi.org/10.7554/eLife.91400.4.sa2
Reviewer #3 (Public Review): https://doi.org/10.7554/eLife.91400.4.sa3
Author response https://doi.org/10.7554/eLife.91400.4.sa4

---

# Additional files

## Supplementary files
MDAR checklist

Supplementary file 1. Demographics of included participants.

Supplementary file 2. Descriptives of questionnaire scores.

Supplementary file 3. Trial types and numbers.

## Data availability
As per request of our ethics committee, participants indicated in the informed consent document if they agreed (or did not agree) that their coded (deidentified) research data would be made available on open source platforms (such as OSF) upon publication of the manuscript. One participant did not consent to this. Therefore, we cannot make the full data set (including single-trial data) publicly available. Interested researchers can contact ALW or BV directly by email to request access to these data. No project proposal needs to be submitted. Unthresholded T maps of all for all reported fMRI contrasts, including the neural signatures, are available on Neurovault (https://neurovault.org/collections/19366/). All analyses code and ROI masks have been made available on OSF (https://osf.io/ywpks/).

The following datasets were generated:

| Author(s) | Year | Dataset title | Dataset URL | Database and Identifier |
|---|---|---|---|---|
| Willems AL | 2025 | Omissions of Threat Trigger Subjective Relief and Prediction Error-Like Signaling in the Human Reward and Salience Systems | https://identifiers.org/neurovault.collection:19366 | NeuroVault, collection:19366 |
| Willems A, Van Oudenhove L, Vervliet B | 2023 | Omissions of Threat Trigger Subjective Relief and Reward Prediction Error-Like Signaling in the Human Reward System | https://osf.io/ywpks/ | Open Science Framework, ywpks |

---

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

# Appendix 1

## Deviations from pre-registration

Main analyses for this study were pre-registered on Open Science Framewwork (OSF) and made available online (https://osf.io/ugkzf). While the general analysis approach was maintained, a number of small deviations from the pre-registered plan were made in the paper. Here, we provide an overview of these deviations.

- In addition to the predictors-of-interest we specified in the pre-registration (Probability and Intensity), we entered Run (four levels: 1, 2, 3, 4) (only in rating and SCR models) and average stimulation-unpleasantness (centered) as predictors of no-interest, as these increased model fit (as assessed via AIC). Note that Run-specific intercepts were included in the first-level GLMs of the fMRI data, but the effects of Run were not explicitly tested.
- We specified in the pre-registration that we would smooth our preprocessed functional data for the univariate analysis, however we also used the smoothed data for the multivariate analyses, since previous research has shown that this does not have a detrimental effect and might even improve the results (*Hendriks et al., 2017*; *Op de Beeck, 2010*).
- We specified in the pre-registration that all regressors in the subject-level GLMs would be modelled as stick-functions (duration = 0). However, in the paper we only modeled the outcome (omission/shock) as stick regressors (similar to previous research), whereas the other regressors (instructions, ratings) were modeled as boxcar regressors to remove their variance from the implicit baseline. Furthermore, we did not include a separate regressor for the onset of the clock as it overlapped with the regressor of the instructions.
- We specified in the pre-registration that the effects of Intensity and Probability on the fMRI data would be analyzed via an full-factorial F-test in SPM12. However, to keep this analysis comparable to our rating and SCR analyses, we extracted parameter estimates from each ROI for each Probability x Intensity combination in the paper, and entered these estimates in a Linear Mixed Model including Probability, Intensity, and their interaction as regressors-of-interest, and average stimulation unpleasantness (centered) as regressor of no-interest.
- We specified in the pre-registration that relief-ratings would be entered as parametric modulator in order to assess trial-by-trial correlations between fMRI data and relief ratings. In the paper, the ratings were z-scored before entering them into the model.
- We specified in the pre-registration that we would use an 8-fold cross-validation for the LASSO-PCR. However, in the paper, we used a 5-fold cross-validation instead, in line with lab standards. Furthermore, in addition to using virtual lesions, we applied a bootstrapping approach to assess the contribution of individual regions to the model signature.
- We added the red nucleus and habenula to the secondary ROI mask, because of their close spatial proximity to the dopaminergic midbrain (red nucleus), their efferent connection to the dopaminergic midbrain (habenula) and their functional relation to omission processing in the past (*Linnman et al., 2011*; *Hikosaka, 2010*).
- We specified in the pre-registration that we would explore threat omission processing in an extended ROI mask using a voxel-wise, FDR small volume corrected approach, followed by a whole-brain parcel-wise approach. However, instead of the whole-brain parcel-wise approach, we used a voxel-wise approach within a grey matter mask, using FWE-correction. Furthermore, as for our main analyses we extracted averaged beta-estimates from each omission-processing cluster (identified with a cluster threshold of $p < 0.05$, FWE-corrected, following voxel threshold $p < 0.001$) and entered these in a Probability x Intensity LMM.

## Appendix 2

### Preprocessing of MRI data with fMRIPrep

Preprocessing of the MRI data was performed using fMRIPrep 20.2.3 (RRID:SCR_016216; *Esteban et al., 2019*; *Esteban, 2018*), which is based on Nipype 1.6.1 (RRID:SCR_002502; *Gorgolewski et al., 2011*; *Gorgolewski, 2018*).

### Anatomical data preprocessing

A total of 1 T1-weighted (T1w) images were found within the input BIDS dataset. The T1-weighted (T1w) image was corrected for intensity non-uniformity (INU) with N4BiasFieldCorrection (*Tustison et al., 2010*), distributed with ANTs 2.3.3 (RRID:SCR_004757; *Avants et al., 2008*), and used as T1w-reference throughout the workflow. The T1w-reference was then skull-stripped with a Nipype implementation of the antsBrainExtraction.sh workflow (from ANTs), using OASIS30ANTs as target template. Brain tissue segmentation of cerebrospinal fluid (CSF), white-matter (WM) and gray-matter (GM) was performed on the brain-extracted T1w using fast (FSL 5.0.9, RRID:SCR_002823; *Zhang et al., 2001*). Volume-based spatial normalization to one standard space (MNI152NLin2009cAsym) was performed through nonlinear registration with antsRegistration (ANTs 2.3.3), using brain-extracted versions of both T1w reference and the T1w template. The following template was selected for spatial normalization: ICBM 152 Nonlinear Asymmetrical template version 2009c [RRID:SCR_008796; TemplateFlow ID: MNI152NLin2009cAsym] (*Fonov et al., 2009*).

### Functional data preprocessing

For each of the 4 BOLD runs found per subject (across all tasks and sessions), the following preprocessing was performed. First, a reference volume and its skull-stripped version were generated using a custom methodology of fMRIPrep. A B0-nonuniformity map (or fieldmap) was estimated based on two (or more) echo-planar imaging (EPI) references with opposing phase-encoding directions, with 3dQwarp (AFNI 20160207; *Wenzel et al., 2018*). Based on the estimated susceptibility distortion, a corrected EPI (echo-planar imaging) reference was calculated for a more accurate co-registration with the anatomical reference. The BOLD reference was then co-registered to the T1w reference using flirt FSL 5.0.9 *Jenkinson and Smith, 2001* with the boundary-based registration (*Greve and Fischl, 2009*) cost-function. Co-registration was configured with nine degrees of freedom to account for distortions remaining in the BOLD reference. Head-motion parameters with respect to the BOLD reference (transformation matrices, and six corresponding rotation and translation parameters) are estimated before any spatiotemporal filtering using mcflirt (FSL 5.0.9 *Jenkinson et al., 2002*). BOLD runs were slice-time corrected using 3dTshift from AFNI 20160207 (RRID:SCR_005927; *Cox and Hyde, 1997*). The BOLD time-series (including slice-timing correction when applied) were resampled onto their original, native space by applying a single, composite transform to correct for head-motion and susceptibility distortions. These resampled BOLD time-series will be referred to as preprocessed BOLD in original space, or just preprocessed BOLD. The BOLD time-series were resampled into standard space, generating a preprocessed BOLD run in MNI152NLin2009cAsym space. First, a reference volume and its skull-stripped version were generated using a custom methodology of fMRIPrep. Several confounding time-series were calculated based on the preprocessed BOLD: framewise displacement (FD), DVARS and three region-wise global signals. FD was computed using two formulations following Power absolute sum of relative motions *Power et al., 2014* and Jenkinson (relative root mean square displacement between affines *Jenkinson and Smith, 2001*). FD and DVARS are calculated for each functional run, both using their implementations in Nipype following the definitions by *Power et al., 2014*. The three global signals are extracted within the CSF, the WM, and the whole-brain masks. Additionally, a set of physiological regressors were extracted to allow for component-based noise correction (CompCor *Behzadi et al., 2007*). Principal components are estimated after high-pass filtering the preprocessed BOLD time-series (using a discrete cosine filter with 128 s cut-off) for the two CompCor variants: temporal (tCompCor) and anatomical (aCompCor). tCompCor components are then calculated from the top 2% variable voxels within the brain mask. For aCompCor, three probabilistic masks (CSF, WM and combined CSF +WM) are generated in anatomical space. The implementation differs from that of *Behzadi et al., 2007* in that instead of eroding the masks by 2 pixels on BOLD space, the aCompCor masks are subtracted a mask of pixels that likely contain a volume fraction of GM. This mask is obtained by thresholding the corresponding partial volume map at 0.05, and it ensures components are not extracted from voxels containing a minimal fraction of GM. Finally, these masks are resampled into BOLD space and binarized by thresholding at 0.99

(as in the original implementation). Components are also calculated separately within the WM and CSF masks. For each CompCor decomposition, the k components with the largest singular values are retained, such that the retained components' time series are sufficient to explain 50% of variance across the nuisance mask (CSF, WM, combined, or temporal). The remaining components are dropped from consideration. The head-motion estimates calculated in the correction step were also placed within the corresponding confounds file. The confound time series derived from head motion estimates and global signals were expanded with the inclusion of temporal derivatives and quadratic terms for each (*Satterthwaite et al., 2013*). Frames that exceeded a threshold of 0.9 mm FD or 2.0 standardised DVARS were annotated as motion outliers. All resamplings can be performed with a single interpolation step by composing all the pertinent transformations (i.e. head-motion transform matrices, susceptibility distortion correction when available, and co-registrations to anatomical and output spaces). Gridded (volumetric) resamplings were performed using antsApplyTransforms (ANTs), configured with Lanczos interpolation to minimize the smoothing effects of other kernels (*Lanczos, 1964*). Non-gridded (surface) resamplings were performed using mri_vol2surf (FreeSurfer).

Many internal operations of fMRIPrep use Nilearn 0.6.2 (RRID:SCR_001362) *Abraham et al., 2014*, mostly within the functional processing workflow. For more details of the pipeline, see the section corresponding to workflows in fMRIPrep's documentation.

# Appendix 3

## Whole-brain threat omission responses

In addition to examining how unexpected omissions of threat are processed in the pre-specified ROIs, we explored whole-brain fMRI activations related to unexpected threat omissions (*Appendix 3—figure 1* and *Appendix 3—table 1*) and relief modulation (*Appendix 3—figure 4* and *Appendix 3—table 2*). Results of these analyses are presented below for exploratory purposes, but are not interpreted. For each contrast, group-level activity maps were masked with a grey matter mask and thresholded at p<0.001 (uncorrected). An overview of the (de)activations is shown for exploratory purposes in the figures and MNI coordinates of the peak activations within each cluster can be found in the tables.

**Appendix 3—table 1.** Whole-brain omission fMRI responses.

Contrast: non0%>0%

| L/R | Region | K | p (cluster) | MNI Coordinates (xyz) | | | T (peak) |
|---|---|---|---|---|---|---|---|
| L | Mid cingulate gyrus, extending to supplementary motor area and anterior cingulate gyrus | 1606 | <0.001 | -3 | 4 | 45 | 8.19 |
| R | Insula | 1287 | <0.001 | 44 | 12 | 1 | 8.05 |
| L | Insula | 1117 | <0.001 | –33 | 24 | 10 | 7.45 |
| L | Putamen | 65 | 0.006 | –25 | 6 | 1 | 6.83 |
| R | Cerebellum | 417 | <0.001 | 10 | –61 | –10 | 6.46 |
| R | Cerebellum | 37 | 0.093 | 38 | –63 | –26 | 5.70 |
| L | Supplementary motor area | 40 | 0.068 | –15 | 2 | 67 | 5.26 |
| L | Superior temporal gyrus | 148 | <0.001 | –59 | –33 | 21 | 5.26 |
| L | Inferior parietal gyrus | 170 | <0.001 | –43 | –53 | 54 | 5.03 |
| L | Cerebellum vermis | 55 | 0.016 | -3 | –75 | –15 | 4.84 |
| R | Inferior parietal gyrus | 37 | 0.093 | 50 | –47 | 54 | 4.74 |
| L | Supplementary motor area | 44 | 0.046 | -7 | -9 | 71 | 4.51 |

Contrast: non0%<0%

| L/R | Region | K | p (cluster) | MNI Coordinates (xyz) | | | T (peak) |
|---|---|---|---|---|---|---|---|
| L | Fusiform gyrus, extending to lingual gyrus, occipital gyrus and calcarine gyrus | 2408 | <0.001 | –31 | –59 | –15 | 8.86 |
| L | Posterior cingulate gyrus | 638 | <0.001 | -9 | –49 | 34 | 7.34 |
| L | vmPFC | 665 | <0.001 | -5 | 42 | –17 | 6.91 |
| R | Superior occipital gyrus, extending to calcarine gyrus | 1686 | <0.001 | 26 | –93 | 18 | 6.87 |
| L | Middle temporal gyrus | 176 | <0.001 | –63 | -9 | -8 | 6.73 |
| R | Fusiform gyrus | 294 | <0.001 | 32 | –37 | –15 | 6.73 |
| R | Precentral gyrus | 60 | 0.010 | 44 | –23 | 60 | 6.70 |
| R | Cerebellum | 87 | 0.001 | 16 | –87 | –41 | 6.33 |
| R | Middle temporal gyrus | 81 | 0.002 | 62 | -5 | –15 | 5.74 |
| R | Precuneus | 79 | 0.002 | 12 | –53 | 12 | 5.73 |
| L | Hippocampus | 64 | 0.007 | –25 | –11 | –23 | 5.17 |
| L | Posterior orbitofrontal gyrus | 37 | 0.093 | –43 | 28 | –15 | 5.12 |
| R | Middle frontal gyrus | 91 | 0.001 | 28 | 26 | 45 | 4.98 |
| L | Middle frontal gyrus | 230 | <0.001 | –27 | 22 | 54 | 4.93 |
| L | Middle temporal gyrus | 60 | 0.010 | –57 | –57 | -4 | 4.89 |
| L | Paracentral lobule | 46 | 0.037 | -5 | –31 | 67 | 4.68 |

*Appendix 3—table 1 Continued on next page*

*Appendix 3—table 1 Continued*

**Contrast: non0%>0%**

| | | | | | | | |
|---|---|---|---|---|---|---|---|
| R | Paracentral lobule | 39 | 0.076 | 6 | −37 | 62 | 4.59 |

Note. Regions are identified at voxel-level p<0.001, and with cluster correction p<0.05 (FWE-corrected); *L/R* indicates if the cluster (or peak of the cluster) is part of the left or right hemisphere; *Region* name is identified using the AAL atlas; *K* is the number of voxels in the cluster; *Coordinates* are the MNI coordinates of cluster peak; *T* is the value of the T-statistic of the cluster peak.

For each of the omission-processing clusters we extracted the beta-estimates per probability and intensity level. As in our main analyses, we entered them into a Probability x Intensity LMM in order to assess how omission-responses were modulated by the provided probability and intensity instructions. Results are plotted per cluster in *Appendix 3—figures 2 and 3*. Significance of the main effects are indicated by the reported p-values. Only the left putamen cluster did not show a difference between 100% and 0% trials (BF in favor of null-hypothesis: 3.06; axiom 3), illustrated by the teal frame. All other regions had a stronger (de)activation for 100% trials compared to 0% trials.

**Appendix 3—table 2.** Whole-brain relief modulation.

**Contrast: Positive modulation**

| L/R | Region | K | p (cluster) | MNI Coordinates (xyz) | | | T (peak) |
|---|---|---|---|---|---|---|---|
| R | Insula | 1212 | <0.001 | 42 | 16 | -4 | 8.91 |
| L | Insula | 1027 | <0.001 | −35 | 12 | 5 | 8.31 |
| R | Mid cingulate gyrus, extending to supplementary motor area and anterior cingulate gyrus | 1739 | <0.001 | 2 | 10 | 43 | 6.80 |
| R | Cerebellum | 191 | <0.001 | 38 | −51 | −30 | 6.74 |
| R | Superior temporal gyrus, extending to supramarginal gyrus | 174 | <0.001 | 66 | −39 | 23 | 6.71 |
| R | Cerebellum | 114 | <0.001 | 8 | −71 | −12 | 6.60 |
| L | Supramarginal gyrus | 155 | <0.001 | −63 | −47 | 29 | 5.84 |
| L | Cerebellum | 99 | <0.001 | −37 | −51 | −32 | 5.22 |
| R | Medial frontal gyrus | 44 | 0.042 | 6 | 34 | 38 | 5.00 |

**Contrast: Negative modulation**

| L/R | Region | K | p (cluster) | MNI Coordinates (xyz) | | | T (peak) |
|---|---|---|---|---|---|---|---|
| L | vmPFC | 381 | <0.001 | -7 | 32 | −17 | 9.12 |
| R | Angular gyrus, extending to occipital gyri | 523 | <0.001 | 48 | −73 | 32 | 7.59 |
| L | Angular gyrus, extending to occipital gyri | 535 | <0.001 | −45 | −73 | 36 | 6.62 |
| L | Fusiform gyrus | 40 | 0.063 | −27 | −39 | −19 | 5.98 |
| R | Middle frontal gyrus | 107 | <0.001 | 26 | 28 | 45 | 5.70 |
| R | Lingual gyrus | 167 | <0.001 | 14 | −79 | −12 | 5.68 |
| R | Superior occipital gyrus | 39 | 0.070 | 16 | −91 | 18 | 5.40 |
| L | Precuneus | 534 | <0.001 | -1 | −63 | 45 | 5.33 |
| L | Lingual gyrus | 191 | <0.001 | −29 | −81 | −17 | 5.22 |
| L | Middle frontal gyrus | 194 | <0.001 | −29 | 20 | 47 | 5.19 |
| L | Postcentral gyrus | 69 | 0.004 | −63 | -5 | 32 | 4.78 |
| L | Lingual gyrus | 48 | 0.028 | −27 | −53 | -6 | 4.46 |

Note. Regions are identified at voxel-level p<0.001, and with cluster correction p<0.05 (FWE-corrected); *L/R* indicates if the cluster (or peak of the cluster) is part of the left or right hemisphere; *Region* name is identified using the AAL atlas; *K* is the number of voxels in the cluster; *Coordinates* are the MNI coordinates of cluster peak; *T* is the value of the T-statistic of the cluster peak.

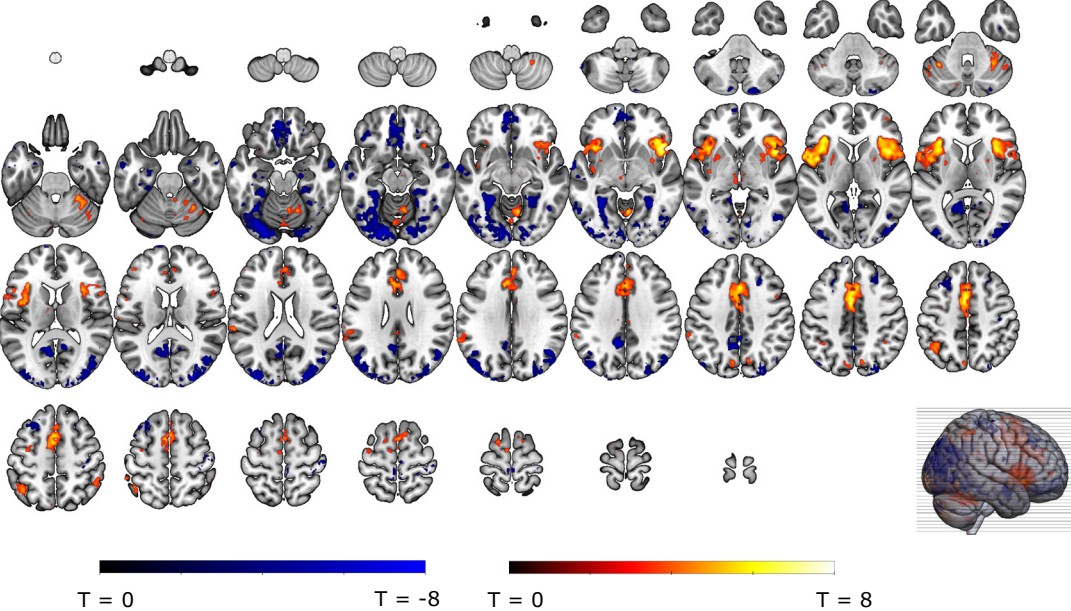

**Appendix 3—figure 1.** Whole-brain (grey-matter masked) omission responses identified via the non0 % > 0% omission contrast, thresholded at p < 0.001 (uncorrected) for display purposes.

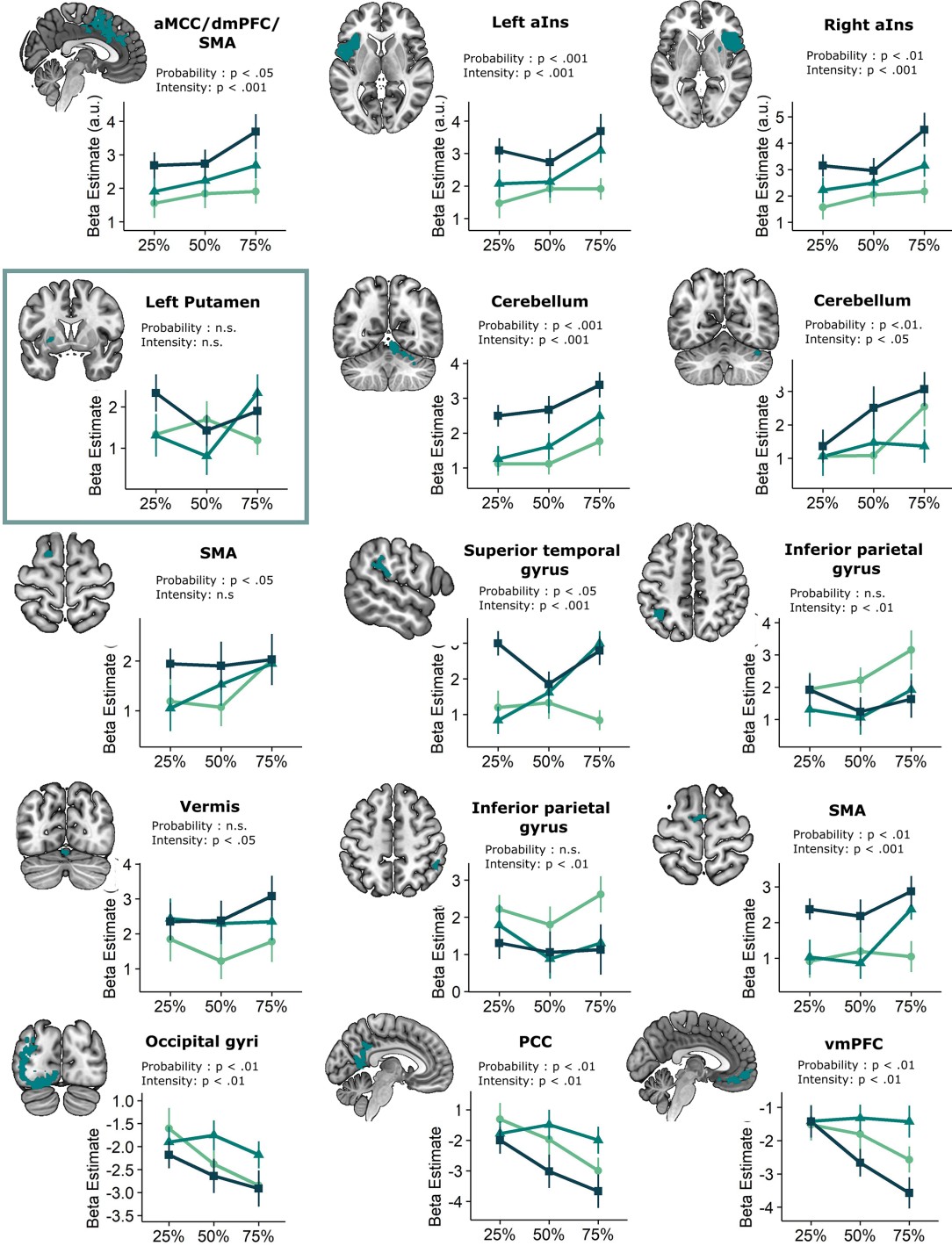

**Appendix 3—figure 2.** Probability- and intensity-related changes in threat omission-related fMRI responses per exploratory cluster. Significant effects are indicated by the p-value.

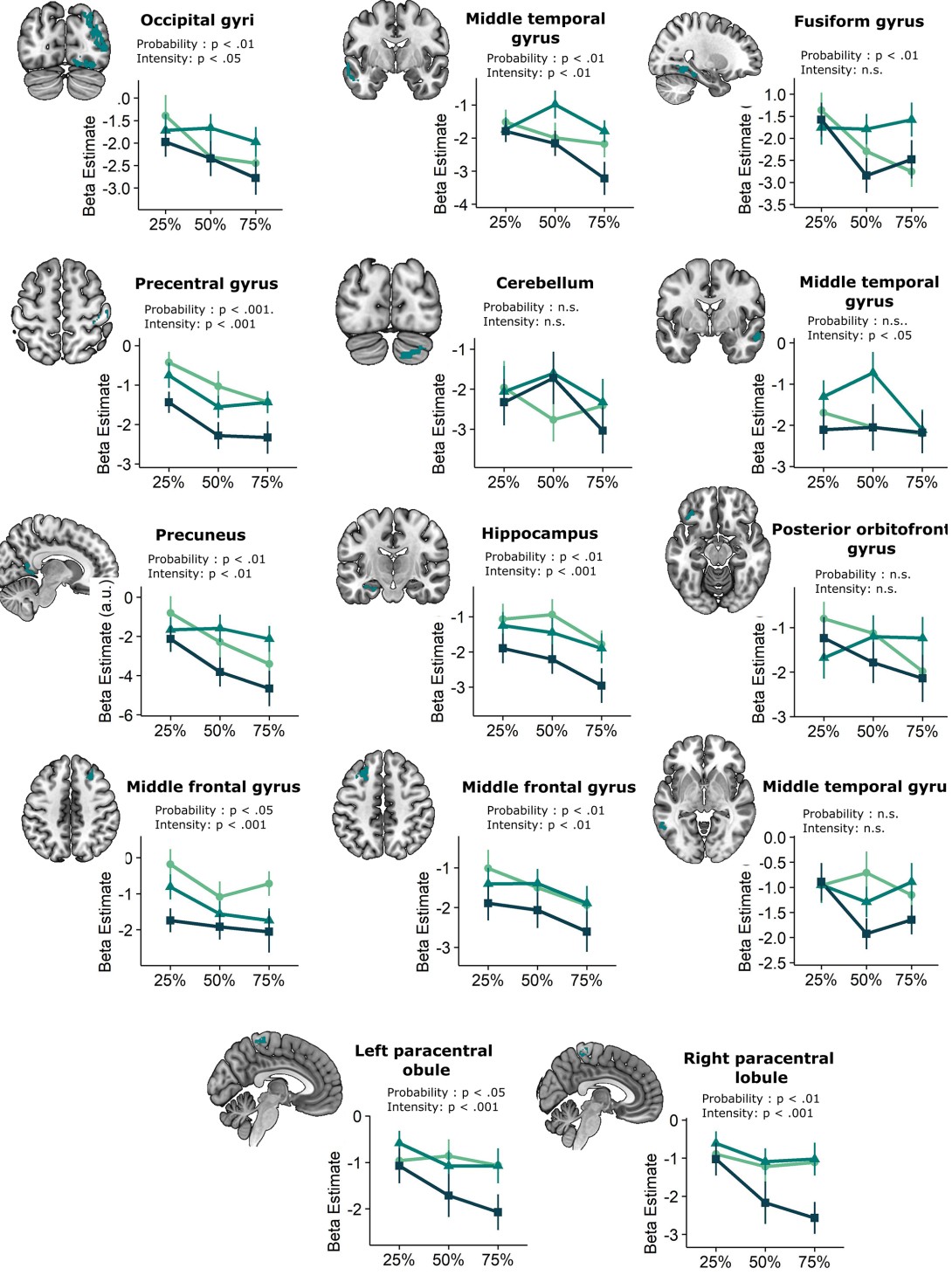

**Appendix 3—figure 3.** Probability- and intensity-related changes in threat omission-related fMRI responses per exploratory cluster. Significant effects are indicated by the p-value.

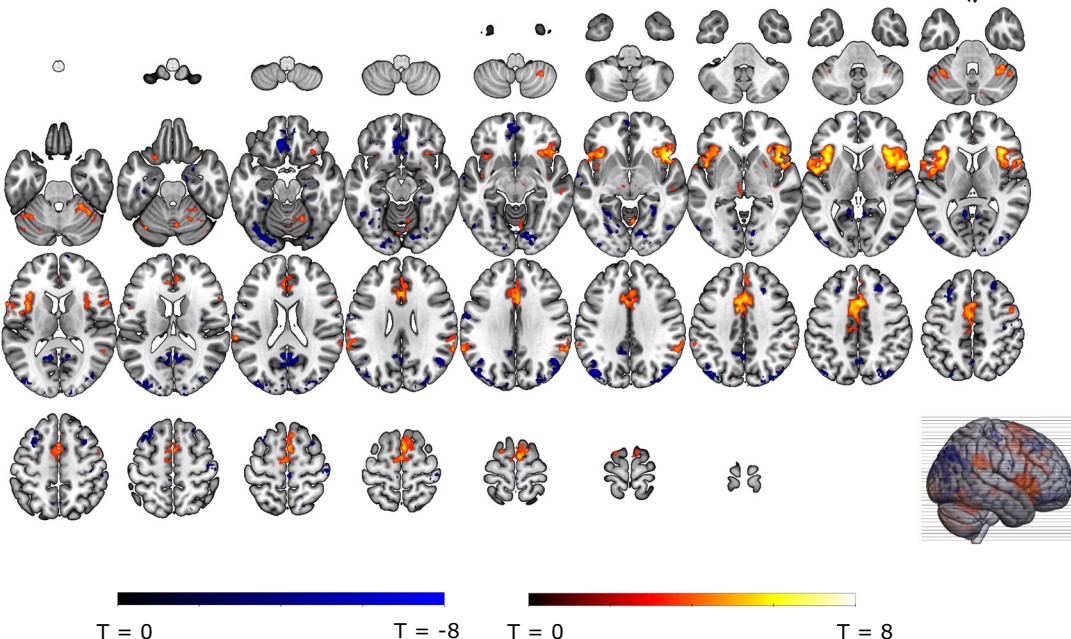

T = 0    T = -8    T = 0    T = 8

**Appendix 3—figure 4.** Whole-brain (grey-matter masked) relief modulation of omission-related fMRI responses identified via the relief modulation contrast, thresholded at p < 0.001 (uncorrected) for display purposes.

## Appendix 4

### Exploring subjective, physiological and neural responses to the anticipation and delivery of the stimulation

In addition to examining how unexpected omissions of threat are processed, we explored how Probability and Intensity instructions affected subjective, physiological and neural responses to the anticipation and delivery of the stimulation.

### Anticipatory SCR to the instructions

A 5 (Probability: 0%, 25%, 50%, 75%, 100%) x 4 (Run: 1, 2, 3, 4) LMM revealed that anticipatory SCR to the countdown clock increased with increasing probability instructions (main effect of Probability, $F(4, 1804)=49.30$, $p<0.001$, $\omega_p^2 = 0.10$). Bonferroni-Holm corrected follow-up pairwise contrasts confirmed that SCR were larger for the anticipation of completely predicted pain (100% instructions) than for unpredictable pain (25%, 50%, and 75% instructions; $p<0.001$), which were larger than SCR to the anticipation of completely predicted pain omission (0% instructions, $p<0.05$, except for 25% vs 0% where $p=0.11$). Additionally, anticipatory responses to unpredictable pain instructions increased as a function of the provided probability instructions (75% vs 25%, $p<0.05$).

A follow-up 4 (Probability: 25%, 50%, 75%, 100%) x 3 (Intensity: weak, moderate, strong) x 4 (Run: 1, 2, 3, 4) LMM examining the combined effect of Probability and Intensity in more detail showed that anticipatory SCR increased for both increasing probability ($F(3,1468)=54.36$, $p<0.001$, $\omega_p^2 = 0.10$) and increasing intensity instructions ($F(2,1468)=67.33$, $p<.001$, $\omega_p^2 = 0.08$). Responses were overall higher to the strongest intensity ($p<0.001$) compared to the weak and moderate intensities, which did not differ significantly ($p=0.15$). Finally a significant Probability x Intensity interaction revealed that the effect of Probability was most pronounced when anticipating a strong stimulus with 100% certainty, and not significant for any of the 25% to 75% comparisons ($p>0.07$) ($F(6,1468.06)=2.94$, $p<0.01$, $\omega_p^2 = 0.01$). Notably, the effect of Run was also significant ($F(3,1468.23)=30.96$, $p<0.001$, $\omega_p^2 = 0.06$), indicating that anticipatory SCR gradually declined over time (run 1 vs run 4, $p<0.001$). Nevertheless, there were no significant interactions of Run with either Intensity or Probability ($p$'s$>0.098$), suggesting that responses to the effects of interest did not alter over time.

The absence of a clear increase in anticipatory SCR over 25% to 75% probability instructions on a group level, could indicate that participants did not believe the instructed probabilities. In fact, the probability instructions did not map exactly onto the actual experienced probabilities of stimulation: all instructions were followed by a stimulation in 25% of the trials. It might therefore be that the small to absent probability effects for omission responses were the result of aberrant anticipation of stimulation. In order to control for this alternative explanation, we plotted for each participant their anticipatory SCR over increasing probability levels (*Appendix 4—figure 2*). These plots revealed that most participants did have larger anticipatory SCR responses for 75% trials compared to 25% trials, in line with our instructions, and only 5 participants had the opposite pattern (red frames). These participants were removed in a follow-up subgroup analysis.

### Anticipatory fMRI responses to the presentation of the instructions

We explored anticipation-related whole-brain fMRI activations. Results of these analyses are presented below for exploratory purposes, but are not interpreted. For each contrast, group-level activity maps were masked with a grey matter mask and thresholded at $p<0.001$ (uncorrected). An overview of the (de)activations is shown for exploratory purposes in the figures and MNI co-ordinates of the peak activations within each cluster can be found in the tables.

We reran our first level model, now including separate epoch-regressors for the onset of instructions of each probability x intensity combination (13 regressors), and separate general regressors for the omissions, stimulations and ratings. We contrasted all non0% trials (where there was in fact expectation of stimulation) with 0% trials in order to assess which regions show increased/ decreased activation in relation to expectation of stimulation. Note that 0% trials in our task are similar to CS- trials in standard conditioning/extinction procedures, as these trials indicated that no stimulation would follow.

**Appendix 4—table 1.** Whole-brain anticipatory fMRI responses to the presentation of the instructions.

**Contrast: non0%>0%**

| L/R | Region | K | p (cluster) | MNI Coordinates (xyz) | | | T (peak) |
|---|---|---|---|---|---|---|---|
| R | Cerebellum, vermis | 149 | <0.001 | 4 | –63 | -8 | 8.80 |
| L | Superior temporal gyrus | 362 | <0.001 | –59 | –33 | 21 | 8.13 |
| L | Mid cingulate gyrus | 2058 | <0.001 | -5 | -1 | 45 | 7.95 |
| L | Rolandic operculum | 919 | <0.001 | –59 | 4 | 5 | 7.49 |
| L | Thalamus | 175 | <0.001 | –5 | –21 | 1 | 7.26 |
| R | Insula | 852 | <0.001 | 42 | 18 | -6 | 7.18 |
| L | Inferior frontal gyrus | 76 | 0.003 | –39 | 38 | 12 | 6.08 |
| R | Brain stem | 128 | <0.001 | 6 | –29 | -8 | 6.03 |
| L | Insula | 50 | 0.033 | –41 | –13 | -4 | 5.66 |
| R | Cerebellum | 220 | <0.001 | 30 | –45 | –30 | 5.56 |
| L | Precuneus | 72 | 0.005 | -7 | –49 | 51 | 5.36 |
| R | Supplementary motor area | 54 | 0.023 | 12 | -9 | 67 | 5.26 |
| L | Inferior parietal gyrus | 78 | 0.003 | –45 | –55 | 49 | 4.94 |

**Contrast: non0%<0%**

| L/R | Region | K | p (cluster) | MNI Coordinates (xyz) | | | T (peak) |
|---|---|---|---|---|---|---|---|
| L | Fusiform gyrus | 369 | <0.001 | –31 | –45 | -6 | 8.08 |
| L | Middle temporal gyrus | 678 | <0.001 | –59 | -9 | -8 | 7.80 |
| R | Angular gyrus | 329 | <0.001 | 50 | –71 | 36 | 7.30 |
| R | Fusiform gyrus | 286 | <0.001 | 32 | –45 | -6 | 7.24 |
| L | Fusiform gyrus | 545 | <0.001 | –23 | –85 | –17 | 6.78 |
| R | Middle temporal gyrus | 109 | <0.001 | 58 | -1 | –21 | 6.66 |
| L | Superior frontal gyrus | 43 | 0.063 | -9 | 48 | 45 | 6.14 |
| L | Angular gyrus | 398 | <0.001 | –41 | –75 | 43 | 5.81 |
| R | Middle frontal gyrus | 118 | <0.001 | 28 | 26 | 45 | 5.60 |
| L | Precuneus | 83 | 0.002 | -5 | –49 | 36 | 5.12 |
| L | Middle frontal gyrus | 154 | <0.001 | –23 | 26 | 45 | 5.10 |
| R | Cerebellum | 62 | 0.011 | 16 | –85 | –41 | 4.35 |

Note. Regions are identified at voxel-level p<0.001, and with cluster correction p<0.05 (FWE-corrected); *L/R* indicates if the cluster (or peak of the cluster) is part of the left or right hemisphere; *Region* name is identified using the AAL atlas; *K* is the number of voxels in the cluster; *Coordinates* are the MNI coordinates of cluster peak; *T* is the value of the T-statistic of the cluster peak.

We next assessed how the provided probability and intensity instructions modulated the anticipatory fMRI activations. To this end, we specified a first-level linear contrasts for probability contrast weights: –4 (for 25%-regressors), –2 (for 50%-regressors), 2 (for 75%-regressors) and 4 (for 100% regressors), see *Appendix 4—figure 4* and *Appendix 4—table 2*. We assessed how the intensity instructions modulated the anticipatory fMRI activations, by specifying a first-level contrasts for intensity (Strong >Weak), see *Appendix 4—figure 5* and *Appendix 4—table 3*

**Appendix 4—table 2.** ity contrast in whole-brain anticipatory fWhole-brain anticipatory fMRI responses to increasing probability.

**Contrast: Increase with increasing probability**

| L/R | Region | K | p (cluster) | MNI Coordinates (xyz) | | | T (peak) |
|-----|--------|---|-------------|-----|-----|-----|----------|
| L | Mid cingulate gyrus | 1394 | <0.001 | -3 | 4 | 43 | 7.88 |
| L | Middle temporal gyrus | 63 | 0.004 | −57 | −65 | 7 | 6.74 |
| L | Precentral gyrus | 278 | <0.001 | −59 | 10 | 10 | 6.65 |
| L | Putamen | 125 | <0.001 | −23 | 4 | -4 | 6.30 |
| L | Postcentral gyrus | 98 | <0.001 | −33 | −43 | 62 | 6.07 |
| R | Cerebellum | 64 | 0.004 | 28 | −45 | −50 | 6.03 |
| L | Supramarginal gyrus | 201 | <0.001 | −65 | −33 | 27 | 6.02 |
| L | Middle frontal gyrus | 347 | <0.001 | −41 | 46 | 16 | 5.84 |
| R | Cerebellum | 39 | 0.050 | 52 | −55 | −34 | 5.83 |
| R | Insula | 71 | 0.002 | 40 | 20 | 5 | 5.81 |
| L | Precuneus | 266 | <0.001 | -9 | −55 | 54 | 5.77 |
| R | Cerebellum | 216 | <0.001 | 34 | −51 | −32 | 5.72 |
| R | Middle frontal gyrus | 62 | 0.004 | 32 | 42 | 27 | 5.43 |
| R | Supramarginal gyrus | 277 | <0.001 | 58 | −41 | 32 | 5.37 |
| L | Cerebellum | 122 | <0.001 | −45 | −53 | −37 | 5.17 |
| R | Insula | 37 | 0.063 | 40 | 6 | 5 | 5.06 |
| L | Precuneus | 95 | <0.001 | -7 | −77 | 45 | 5.06 |
| L | Cerebellum | 73 | 0.002 | −13 | −75 | −41 | 4.89 |
| L | Putamen | 35 | 0.079 | −25 | 6 | 7 | 4.65 |
| R | Cerebellum | 34 | 0.089 | 36 | −61 | −39 | 4.44 |

Contrast: Decrease with increasing probability

*No significant clusters of activation*

Note. Regions are identified at voxel-level p<0.001, and with cluster correction p<0.05 (FWE-corrected); *L/R* indicates if the cluster (or peak of the cluster) is part of the left or right hemisphere; *Region* name is identified using the AAL atlas; *K* is the number of voxels in the cluster; *Coordinates* are the MNI coordinates of cluster peak; *T* is the value of the T-statistic of the cluster peak.

**Appendix 4—table 3.** Intensity contrast in whole-brain anticipatory fMRI activations.

**Contrast: Strong >Weak**

| L/R | Region | K | p (cluster) | MNI Coordinates (xyz) | | | T (peak) |
|-----|--------|---|-------------|-----|-----|-----|----------|
| R | Supplementary motor area | 2942 | <0.001 | 6 | 6 | 67 | 9.92 |
| R | Insula | 1028 | <0.001 | 30 | 22 | -8 | 8.90 |
| R | Cerebellum | 1194 | <0.001 | 22 | −57 | −21 | 8.37 |
| L | Cerebellum | 478 | <0.001 | −35 | −57 | −26 | 7.71 |
| L | Superior temporal gyrus | 814 | <0.001 | −51 | 2 | 1 | 7.66 |
| R | Supramarginal gyrus | 674 | <0.001 | 64 | −45 | 32 | 6.91 |
| L | Cerebellum | 147 | <0.001 | 36 | −51 | −52 | 6.90 |
| L | Thalamus | 184 | <0.001 | 6 | −17 | 1 | 6.78 |
| L | Supramarginal gyrus | 568 | <0.001 | −59 | −25 | 29 | 6.48 |

*Appendix 4—table 3 Continued on next page*

*Appendix 4—table 3 Continued*

**Contrast: Strong >Weak**

| | | | | | | | |
|---|---|---|---|---|---|---|---|
| L | Postcentral gyrus | 111 | <0.001 | −35 | −19 | 45 | 6.47 |
| L | Postcentral gyrus | 100 | <0.001 | −31 | −43 | 62 | 6.21 |
| R | Precentral gyrus | 166 | <0.001 | 52 | -7 | 47 | 6.19 |
| L | Postcentral gyrus | 75 | 0.002 | −31 | −31 | 54 | 5.47 |
| R | Cerebellum | 40 | 0.064 | 16 | −67 | −48 | 5.45 |
| L | Insula | 47 | 0.031 | −35 | -9 | 1 | 5.05 |
| R | Amygdala | 63 | 0.007 | 22 | -1 | −12 | 5.04 |

**Contrast: Strong <Weak**

| | | | | | | | |
|---|---|---|---|---|---|---|---|
| R | Superior frontal gyrus | 170 | <0.001 | 24 | 36 | 47 | 6.09 |
| R | Angular gyrus | 115 | <0.001 | 40 | −73 | 45 | 5.24 |
| L | Inferior parietal gyrus | 72 | 0.003 | −33 | −71 | 43 | 4.80 |
| R | Middle frontal gyrus | 63 | 0.007 | 48 | 42 | 14 | 4.71 |

Note. Regions are identified at voxel-level p<0.001, and with cluster correction p<0.05 (FWE-corrected); *L/R* indicates if the cluster (or peak of the cluster) is part of the left or right hemisphere; *Region* name is identified using the AAL atlas; *K* is the number of voxels in the cluster; *Coordinates* are the MNI coordinates of cluster peak; *T* is the value of the T-statistic of the cluster peak.

## Subjective and physiological responses to the stimulation

Besides focusing on the omission and anticipation window, we explored how Probability and Intensity instructions affected unconditioned responding (self-reported unpleasantness and stimulation SCR) to the delivery of the stimulation. We conducted two additional linear mixed models with self-reported US unpleasantness and stimulation SCR as dependent variables and Probability (2 levels: non100%, 100%) and Intensity (weak, moderate, strong), and their interaction as fixed effects, in addition to a subject-specific intercept as random effect. Note that we pooled all non100% probability levels so that the probability levels were balanced.

The 2 (probability: non-100%, 100%) x 3 (Intensity: weak, moderate, strong) x 4 (Run: 1, 2, 3, 4) LMM showed that unexpected stimulations (following 25%, 50%, 75% instructions) (*Appendix 4— figure 6A*) were rated as significantly more unpleasant and (*Appendix 4—figure 6B*) elicited higher SCR than fully expected stimulations (following 100% instructions), evidenced by main effects of Probability for self-reported unpleasantness ($F(1,678) = 16.15$, p<0.001, $\omega_p^2 = 0.02$) and stimulation SCR ($F(1,569) = 4.07$, p<0.05, $\omega_p^2 = 0.01$). Likewise, (*Appendix 4—figure 6A*) stronger stimulations were experienced as more unpleasant and (*Appendix 4—figure 6B*) elicited higher SCR, evidenced by main effects of Intensity for self-reported unpleasantness ($F(2,678) = 1015.37$, p<0.001, $\omega_p2 = 0.75$) and stimulation SCR ($F(2,569) = 156.33$, p<0.001, $\omega_p^2 = 0.35$) and confirmed by follow-up Bonferroni-Holm corrected pairwise comparisons (p's<0.001).

## fMRI responses to the stimulation

Results of the stimulation-related fMRI analyses are presented below for exploratory purposes, but are not interpreted. For each contrast, group-level activity maps were masked with a grey matter mask and thresholded at p<0.001 (uncorrected). An overview of the (de)activations is shown for exploratory purposes in the figures and MNI co-ordinates of the peak activations within each cluster can be found in the tables.

We assessed whole-brain (grey-matter masked) stimulation-related fMRI responses based on the stimulation >baseline contrast (see *Appendix 4—figure 7* and *Appendix 4—table 4*). Furthermore, we explored whole-brain (grey-matter masked) whether unexpected stimulations elicited stronger stimulation-related (de)activation compared to expected stimulations based on the non-100% stimulation >100% stimulation contrast (*Appendix 4—figure 8* and *Appendix 4— table 5*).

**Appendix 4—table 4.** Whole-brain stimulation-induced activations.

**Contrast: Stimulation >baseline**

| L/R | Region | K | p (cluster) | MNI Coordinates (xyz) | | | T (peak) |
|-----|--------|---|-------------|------|------|------|----------|
| L | Insula | 5078 | <0.001 | −39 | -1 | 16 | 15.62 |
| R | Insula | 3774 | <0.001 | 44 | 10 | -4 | 11.71 |
| R | Cerebellum | 2577 | <0.001 | 14 | −55 | −17 | 11.32 |
| L | Cerebellum | 684 | <0.001 | −35 | −63 | −26 | 9.92 |
| L | Mid cingulate gyrus | 2048 | <0.001 | -3 | 8 | 43 | 9.44 |
| L | Middle frontal gyrus | 271 | <0.001 | −41 | 42 | 23 | 7.12 |
| R | Middle temporal gyrus | 119 | <0.001 | 54 | −31 | 1 | 5.11 |
| L | Cerebellum | 78 | 0.007 | −17 | −77 | −37 | 4.85 |

**Contrast: Stimulation <baseline**

| L/R | Region | K | p (cluster) | MNI Coordinates (xyz) | | | T (peak) |
|-----|--------|---|-------------|------|------|------|----------|
| R | Occipital gyri, extending to fusiform, lingual gyri | 4348 | <0.001 | 32 | −91 | −10 | 14.53 |
| L | Occipital gyri, extending to fusiform, lingual gyri | 4411 | <0.001 | −35 | −91 | −15 | 13.70 |
| L | vmPFC | 1883 | <0.001 | -5 | 42 | −15 | 11.66 |
| R | Precentral gyrus | 271 | <0.001 | 40 | −21 | 51 | 10.21 |
| R | Postcentral gyrus | 85 | 0.004 | 36 | −31 | 56 | 8.25 |
| R | Middle temporal gyrus | 188 | <0.001 | 60 | -3 | −19 | 7.95 |
| L | Middle temporal gyrus | 582 | <0.001 | −61 | −13 | −15 | 6.69 |
| R | Superior frontal gyrus | 136 | <0.001 | 26 | 32 | 49 | 6.22 |
| L | Precentral gyrus | 143 | <0.001 | −57 | -3 | 29 | 6.20 |
| L | Posterior cingulate gyrus | 887 | <0.001 | -3 | −55 | 29 | 6.11 |
| L | Superior frontal gyrus | 136 | <0.001 | −27 | 26 | 56 | 5.94 |
| R | Cerebellum | 46 | 0.083 | 20 | −83 | −41 | 5.31 |

Note. Regions are identified at voxel-level p<0.001, and with cluster correction p<0.05 (FWE-corrected); *L/R* indicates if the cluster (or peak of the cluster) is part of the left or right hemisphere; *Region* name is identified using the AAL atlas; *K* is the number of voxels in the cluster; *Coordinates* are the MNI coordinates of cluster peak; *T* is the value of the T-statistic of the cluster peak.

**Appendix 4—table 5.** Whole-brain unexpected stimulation-induced activations.

**Contrast: non100%>100% stimulations**

| L/R | Region | K | p (cluster) | MNI Coordinates (xyz) | | | T (peak) |
|-----|--------|---|-------------|------|------|------|----------|
| R | Middle temporal gyrus | 121 | <0.001 | 54 | −21 | -6 | 6.52 |
| R | Rolandic operculum | 50 | 0.014 | 42 | −11 | 18 | 5.57 |
| R | Frontal superior gyrus | 44 | 0.027 | 24 | 20 | 47 | 5.17 |
| L | Middle temporal gyrus | 42 | 0.034 | −59 | −25 | -4 | 5.10 |
| L | Precentral gyrus | 72 | 0.002 | −57 | -7 | 32 | 4.75 |
| R | Middle temporal gyrus | 36 | 0.067 | 50 | −55 | 21 | 4.67 |
| R | Medial superior frontal gyrus | 34 | 0.085 | 8 | 54 | 23 | 4.49 |
| R | Mid cingulate gyrus | 34 | 0.085 | 4 | −47 | 34 | 4.32 |

*Appendix 4—table 5 Continued on next page*

*Appendix 4—table 5 Continued*

**Contrast: non100%>100% stimulations**

| | | | | | | | |
|---|---|---|---|---|---|---|---|
| L | Middle temporal gyrus | 33 | 0.096 | −53 | −55 | 23 | 4.32 |

**Contrast: non100%<100% stimulations**

*No significant clusters of activation*

Note. Regions are identified at voxel-level p<0.001, and with cluster correction p<0.05 (FWE-corrected); *L/R* indicates if the cluster (or peak of the cluster) is part of the left or right hemisphere; *Region* name is identified using the AAL atlas; *K* is the number of voxels in the cluster; *Coordinates* are the MNI coordinates of cluster peak; *T* is the value of the T-statistic of the cluster peak.

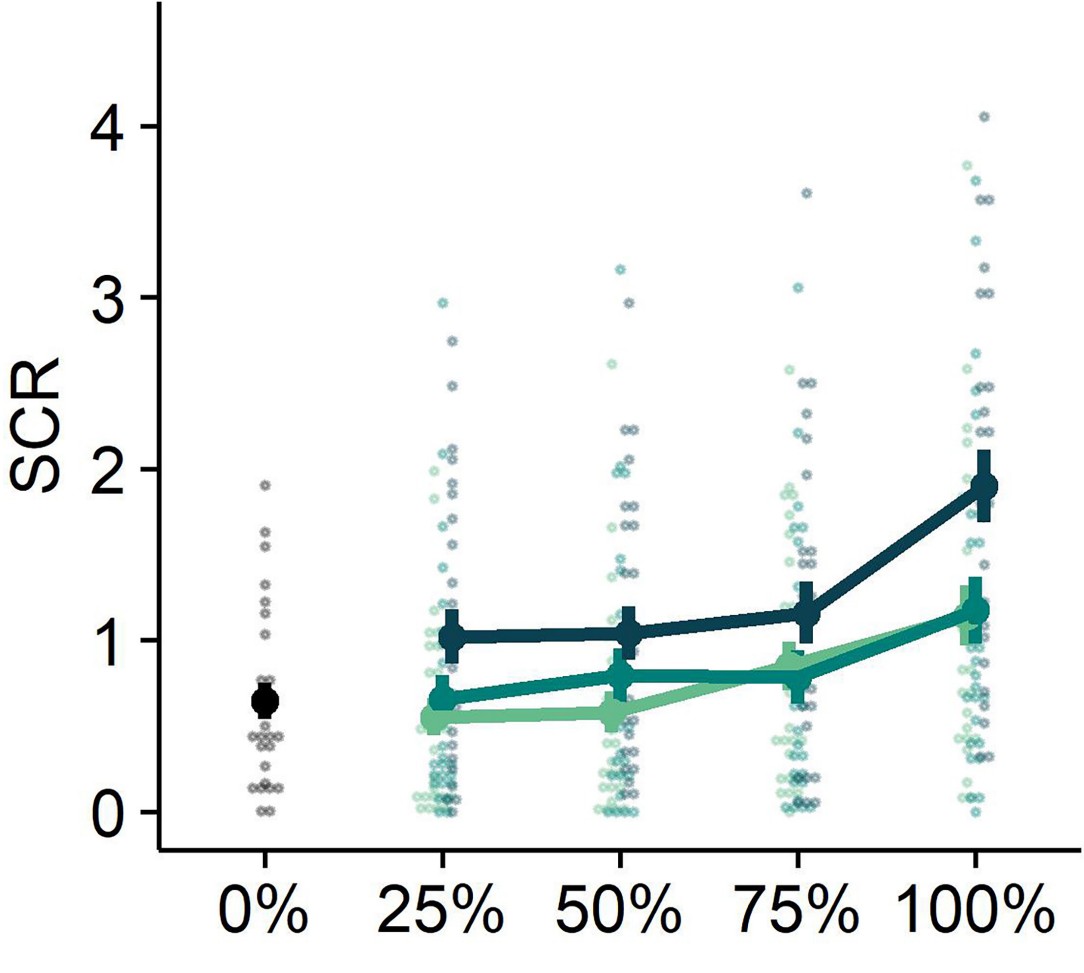

**Appendix 4—figure 1.** Anticipatory SCR to the instructions were larger for stimulations of a higher instructed probability and intensity. Individual data points are presented, with the group averages plotted on top. The error bars represent standard error of the mean.

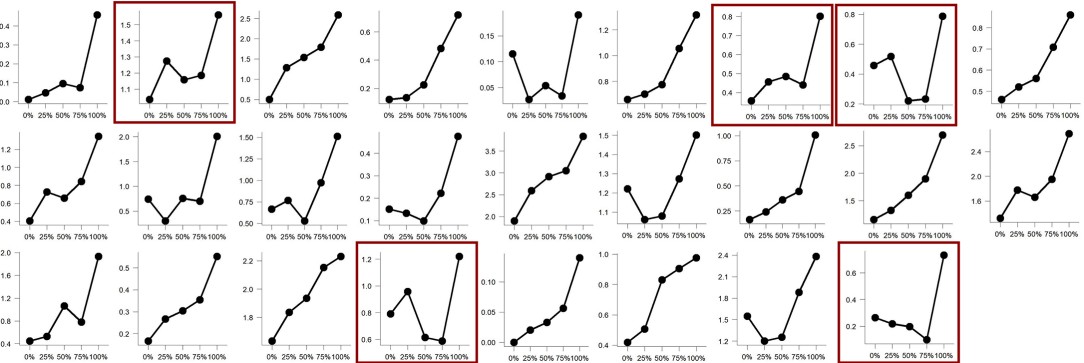

**Appendix 4—figure 2.** Individual anticipatory SCR followed probability instructions.

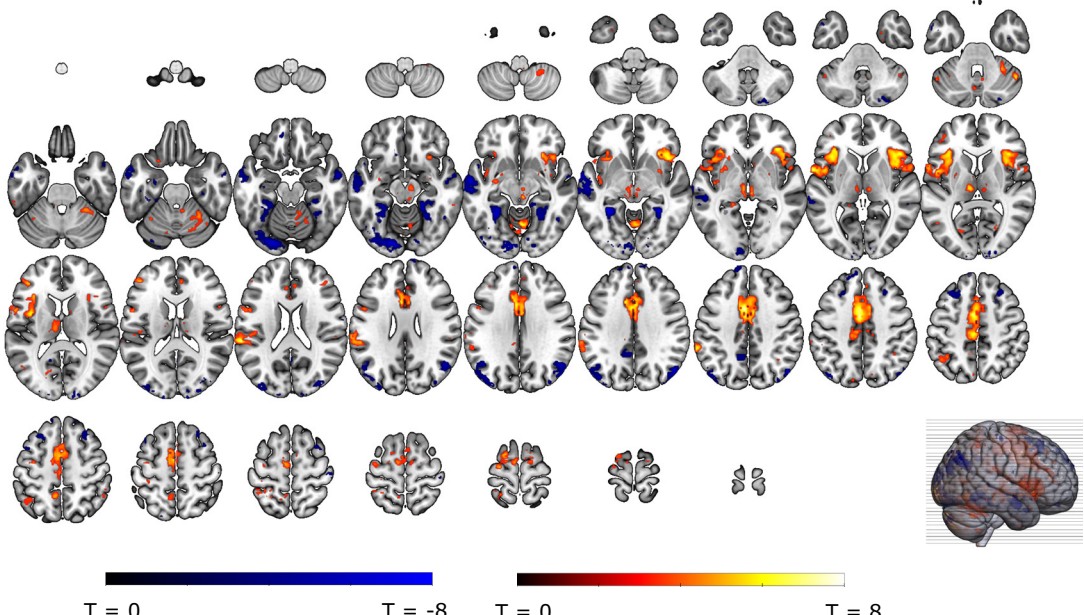

**Appendix 4—figure 3.** Whole-brain (grey-matter masked) anticipatory fMRI responses to the presentation of the instructions identified via the non0 % > 0% anticipation contrast, thresholded at p < 0.001 (uncorrected) for display purposes.

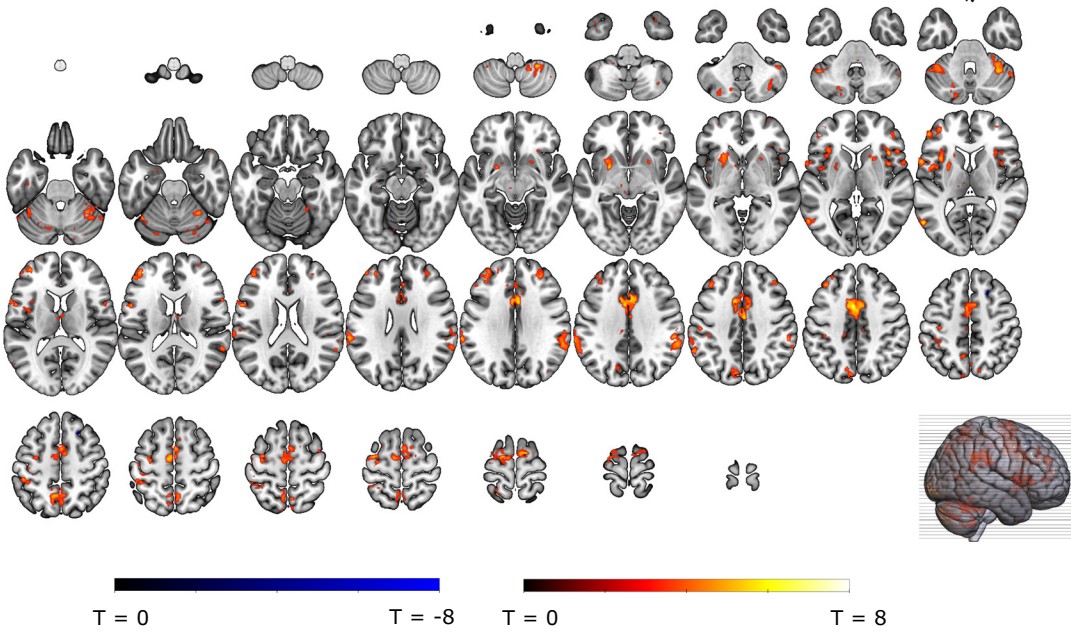

**Appendix 4—figure 4.** A linear increase in whole-brain (grey-matter masked) anticipatory fMRI activations for increasing probability instructions, identified via the probability contrast, thresholded at p < 0.001 (uncorrected) for display purposes.

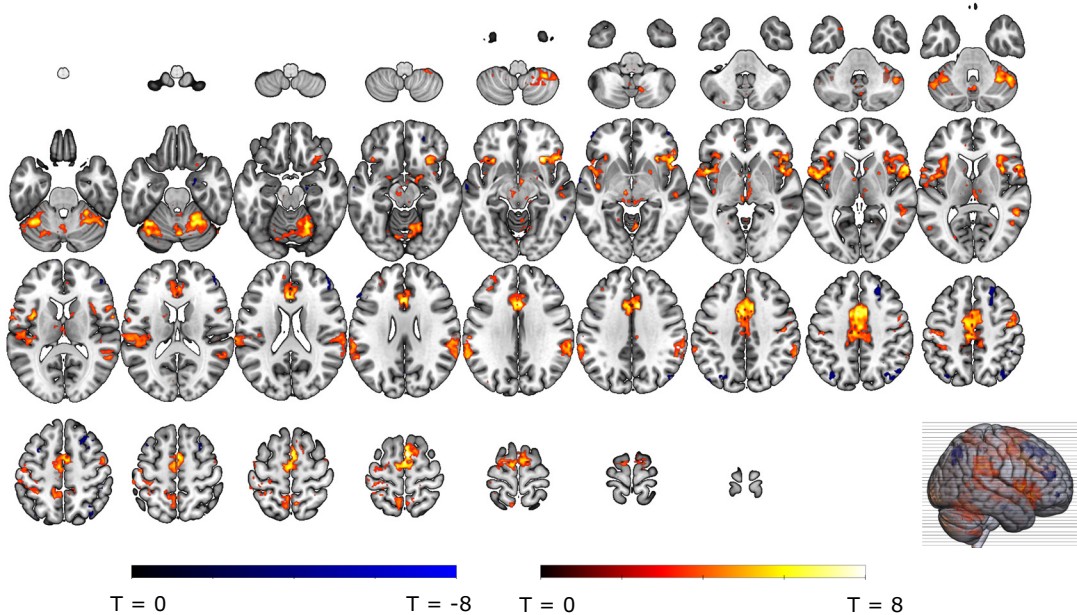

**Appendix 4—figure 5.** Intensity contrast in whole-brain (grey-matter masked) anticipatory fMRI activations, identified via the Intensity contrast, and thresholded at p < 0.001 (uncorrected) for display purposes.

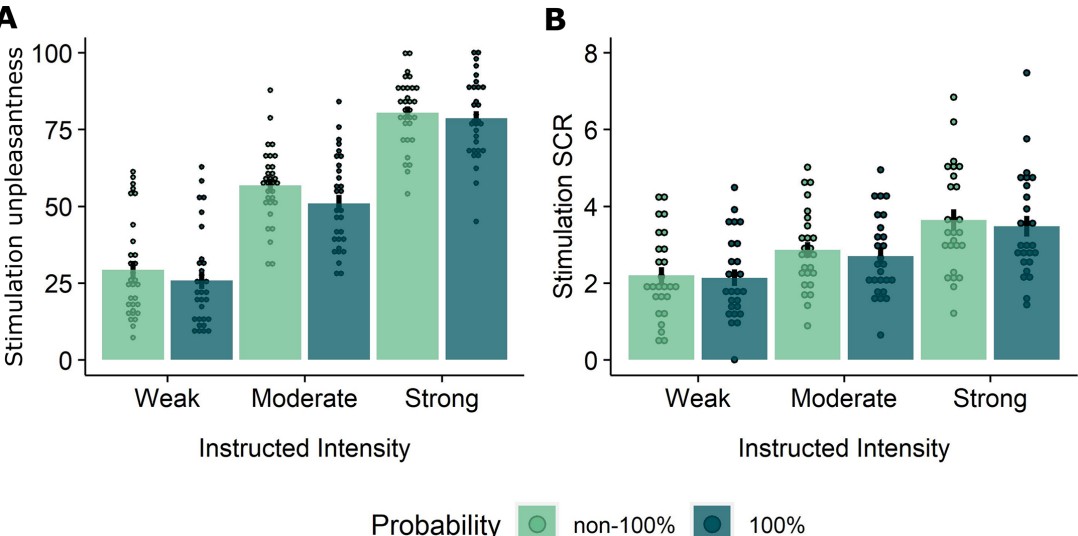

**Appendix 4—figure 6.** Probability and intensity effects for self-reported unpleasantness (A) and stimulation SCR (B).

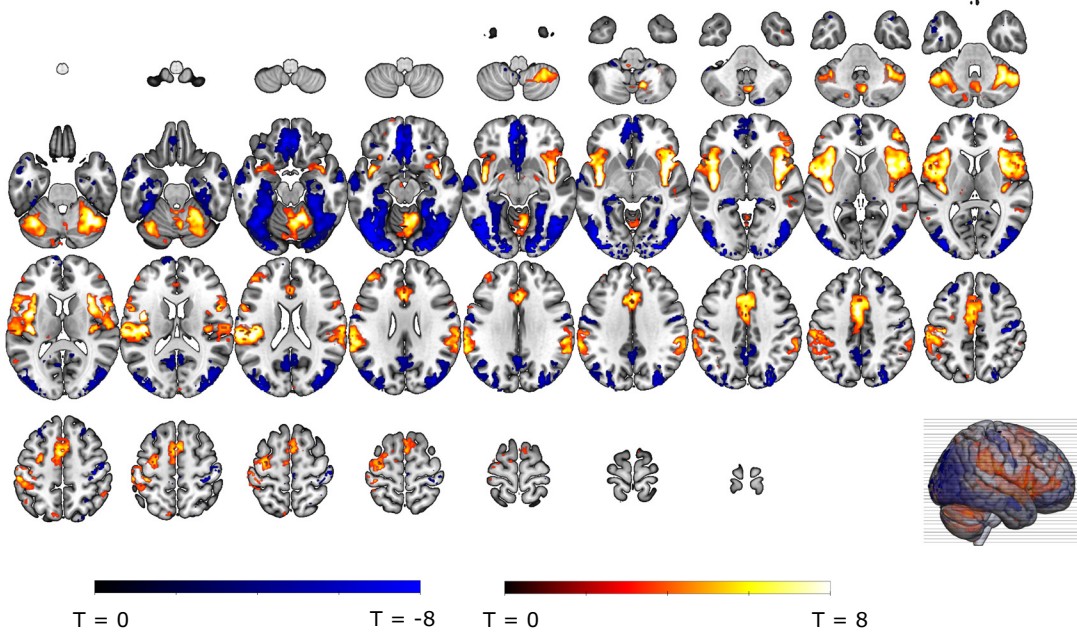

**Appendix 4—figure 7.** Whole-brain (grey-matter masked) stimulation-related fMRI responses based on the stimulation > baseline contrast, thresholded at p < 0.001 (uncorrected) for display purposes.

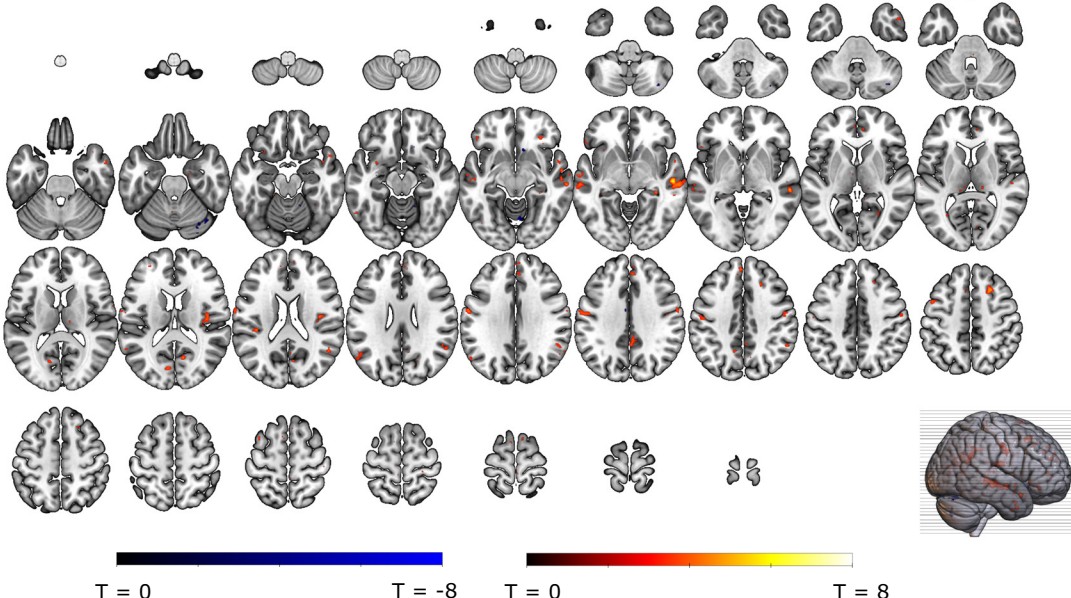

**Appendix 4—figure 8.** Whole-brain (grey-matter masked) unexpected stimulation-related fMRI responses based on the non-100% stimulation > 100% stimulation contrast, and thresholded at p < 0.001 (uncorrected) for display purposes.

## Appendix 5

### Reliability of the instructions and learning related effects

#### Post-experimental manipulation checks

As post-experimental manipulation checks for the probability and intensity instructions, participants were asked at the end of the task how many stimuli they thought they received following instructions of each probability level (probability manipulation check), and how much effort they would exert to prevent future weak/moderate/strong stimulation (from 0 'no effort' to 100 'a lot of effort'). A Friedman test revealed that participants recollected the number of stimuli they received in line with the provided instructions ($\chi^2$(4)=69.6, p<0.001, W=.58; *Appendix 5—figure 1*). Follow-up Wilcoxon signed-rank tests indicated that participants recollected having experienced the least number of stimulations following 0% instructions (all p's<0.001). Furthermore, they recollected having experienced some stimulations after the 25% instructions, but less than following 50% to 100% instructions (p<0.05). The differences between 50%, 75% and 100% did not reach significance following Bonferroni-Holm correction (p>0.13). In line with the intensity instructions, an Intensity (weak, moderate, strong) LMM indicated that participants were willing to exert more effort to prevent stronger stimulations (*F*(2, 58)=97.22, p<0.001, $\omega_p^2$ = 0.76, all Bonferroni-Holm corrected pairwise comparisons, p<0.001; *Appendix 5—figure 1*). Overall, these checks support that the EVA task was successful at inducing expectations of threat.

### 'Accurate' probability instructions do not alter the Probability-effect

A question that was raised by the reviewers was whether the inconsistency between the probability instruction and the experienced reinforcement rate could have detrimental effects on the Probability-related results; especially because the effect of Probability was smaller when only including non-0% trials.

However, there are good reasons to believe that the relatively smaller difference between 25% to 75% trials was not caused by the 'inaccurate' nature of our instructions, but that they are inherent to 'uncertain' probabilities.

First, in a previously unpublished pilot study, we provided participants with 'accurate' probability instructions, meaning that the instruction corresponded to the actual reinforcement rate (e.g., 75% instructions were followed by a stimulation in 75% of the trials etc.). In line with the present results and our previous behavioral study (*Willems and Vervliet, 2021*), the results of this pilot (N=20) showed that the difference in the reported relief between the different probability levels was largest when comparing 0% and the rest (25%, 50%, and 75%). Furthermore the overall effect size of Probability (excluding 0%) matched the one of our previous behavioral study (*Willems and Vervliet, 2021*): $\eta_p^2$ = +/-0.50 (see *Appendix 5—figure 2*).

Second, also in other published studies that used CSs with varying reinforcement rates (which either included explicit written instructions of the reinforcement rates or not) showed that the difference in expectations, anticipatory SCR or omission SCR was largest when comparing the CS0% to the other CSs of varying reinforcement rates (*Ojala et al., 2022*; *Grings and Sukoneck, 1971*; *Ohman et al., 1973*).

Together, this suggests that when there is a possibility of stimulation, any additional difference in probability will have a smaller effect on the omission responses, irrespective of whether the underlying reinforcement rate is accurate or not.

Do the Probability and Intensity effects change over time?In addition to the Intensity and Probability predictors of interest, run number and average US unpleasantness (mean-centered) were entered in the LMM examining relief-pleasantness and omission SCR as predictors of no-interest. For both outcome variables we found significant main effects of Run, indicating that reported relief (*Appendix 5—figure 3A F*(3,1031.73)=9.56, p<0.001, $\omega_p^2$ = 0.02) and omission SCR (*Appendix 5—figure 3B F*(3,862.21)=15.51, p<0.001, $\omega_p^2$ = 0.05) decreased over runs (for all outcome variables, Run 1>Run 4 contrast, p<0.001). The overall absence of significant interactions with Probability and Intensity suggests that our effects of interest did not significantly change over time. Note, there was a trend-level Intensity x Run interaction for relief-pleasantness (*F*(6,1031)=1.98, p=0.065). However, post-hoc contrasts confirmed that the intensity effect did not disappear over blocks. We found that the average unpleasantness of the stimulation had a significant positive effect on the self-reported relief ($\beta$=0.96, p<0.001), suggesting that the more unpleasant the stimulation was perceived, the

more pleasant the relief participants reported whenever the stimulation was omitted (*Appendix 5—figure 3C*).

The effect of Run and the Gambler's Fallacy

A question that was raised by the reviewers was whether omission-related responses could be influenced by dynamical learning or the Gambler's Fallacy, which might have affected the effectiveness of the Probability manipulation.

Inspired by this question, we exploratorily assessed the role of the Gambler's Fallacy and the effects of Run in a separate set of analyses. Indeed, it is possible that participants start to expect a stimulation more when more time has passed since the last stimulation was experienced. To test this alternative hypothesis, we specified two new regressors that calculated for each trial of each participant how many trials had passed since the last stimulation (or since the beginning of the experiment) either overall (across all trials of all probability types; hence called the overall-lag regressor) or per probability level (across trials of each probability type separately; hence called the lag-per-probability regressor). For both regressors a value of 0 indicates that the previous trial was either a stimulation trial or the start of experiment, a value of 1 means that the last stimulation trial was 2 trials ago, etc.

The new models including these regressors for each omission response type (i.e., omission-related activations for each ROI, relief, and omission-SCR) were specified as follows:

1. <u>For the overall lag</u>
   Omission response ~Probability * Intensity * Run +US-unpleasantness+Overall lag + (1|Subject).
2. <u>For the lag per probability level</u>

   Omission response ~Probability * Intensity * Run + US-unpleasantness+Lag-per-probability: Probability + (1|Subject).

Where US-unpleasantness scores were mean-centered across participants; "*" represents main effects and interactions, and ":" represents an interaction (without main effect). Note that we only included an interaction for the lag-per-probability model to estimate separate lag-parameters for each probability level.

The results of these analyses are presented in the tables below. Overall, we found that adding these lag-regressors to the model did not alter our main results. That is: for the VTA/SN, relief and omission-SCR, the main effects of Probability and Intensity remained. Interestingly, the overall-lag-effect itself was significant for VTA/SN activations and omission SCR, indicating that VTA/SN activations were larger when more time had passed since the last stimulation (beta = 0.19), whereas SCR were smaller when more time had passed (beta = –0.03). This pattern is reminiscent of the Perruchet effect, namely that the explicit expectancy of a US increases over a run of non-reinforced trials (in line with the gambler's fallacy effect), whereas the conditioned physiological response to the conditional stimulus declines (in line with the extinction effect) (*Perruchet, 1985*; *McAndrew et al., 2012*). Thus, the observed dissociation between the VTA/SN activations and omission SCR might similarly point to two distinctive processes where VTA/SN activations are more dependent on a consciously controlled process that is subjected to the gambler's fallacy, whereas the strength of the SCR responses is more dependent on an automatic associative process that is subjected to extinction. Importantly, however, even though the temporal distance to the last stimulation had these opposing effects on VTA/SN activations and omission SCRs, the main effects of the probability manipulation remained significant for both outcome variables. This means that the core results of our study still hold.

Next to the overall-lag effect, the lag-per-probability regressor was only significant for the vmPFC. A follow-up of the beta estimates of the lag-per-probability regressors for each probability level revealed that vmPFC activations increased with temporal distance from shock, but only for the 50% trials (beta = 0.47, t=2.75, p<0.01), and not the 25% (beta = 0.25, t=1.49, p=0.14) or the 75% trials (beta = 0.28, t=1.62, p=0.10).

*Appendix 5—table 1 Continued on next page*

**Appendix 5—table 1.** F-statistics and corresponding p-values from the overall lag model.

| Regressor | Relief F | P | SCR F | P | VTA/SN (*) F | P | Left vPut F | P | NAC F | P | vmPFC F | P |
|---|---|---|---|---|---|---|---|---|---|---|---|---|
| Probability | **30.04** | **<0.001** | **4.90** | **<0.01** | **3.59** | **<0.05** | 0.18 | n.s. | 0.88 | n.s. | 1.73 | n.s. |
| Intensity | **620.62** | **<0.001** | **106.65** | **<0.001** | **7.81** | **<0.001** | **3.88** | **<0.05** | 0.70 | n.s. | **4.90** | **<0.01** |
| Run | **9.71** | **<0.001** | **15.56** | **<0.001** | 1.13 | n.s. | 0.76 | n.s. | 0.62 | n.s. | 0.44 | n.s. |
| Probability x Intensity | **3.69** | **<0.01** | 1.54 | n.s. | 1.15 | n.s. | 1.39 | n.s. | 1.76 | n.s. | 0.70 | n.s. |
| Probability x Run | 1.13 | n.s. | 1.24 | n.s. | 1.02 | n.s. | 1.22 | n.s. | 1.74 | n.s. | 1.26 | n.s. |
| Intensity x Run | 1.94 | 0.07 | 1.30 | n.s. | 1.94 | 0.07 | 1.59 | n.s. | 1.01 | n.s. | **2.41** | **<0.05** |
| Probability x Intensity x Run | 0.56 | n.s. | 0.87 | n.s. | 0.76 | n.s. | 0.71 | n.s. | 0.76 | n.s. | 0.83 | n.s. |
| Overall-lag | 2.56 | n.s. | **4.68** | **<0.05** | **11.30** | **<0.001** | <0.01 | n.s. | 0.16 | n.s. | <0.01 | n.s. |
| US-unpleasantness | **29.60** | **<0.001** | 3.00 | 0.096 | 1.84 | n.s. | 0.06 | n.s. | 3.44 | 0.07 | 0.26 | n.s. |

*(*) F-test and p-values were based on the model where outliers were rescored to 2SD from the mean. Note that when retaining the influential outliers for this model, the p-value of the probability effect was p=0.06. For all other outcome variables, rescoring the outliers did not change the results. Significant effects are indicated in bold.

**Appendix 5—table 2.** F-statistics and corresponding p-values from the lag per probability level model.

| Regressor | Relief F | P | SCR F | P | VTA/SN (*) F | P | Left vPut F | P | NAC F | P | vmPFC F | P |
|---|---|---|---|---|---|---|---|---|---|---|---|---|
| Probability | **23.07** | **<0.001** | **4.44** | **<0.05** | **3.28** | **<0.05** | 0.22 | n.s. | 0.18 | n.s. | 0.31 | n.s. |
| Intensity | **625.52** | **<0.001** | **107.49** | **<0.001** | **7.33** | **<0.001** | **3.88** | **<0.05** | 0.66 | n.s. | **4.85** | **<0.01** |
| Run | **8.63** | **<0.001** | **13.91** | **<0.001** | 1.09 | n.s. | 0.72 | n.s. | 0.60 | n.s. | 1.09 | n.s. |
| Probability x Intensity | **3.87** | **<0.01** | 1.33 | n.s. | 1.01 | n.s. | 1.40 | n.s. | 1.81 | n.s. | 0.63 | n.s. |
| Probability x Run | 1.63 | n.s. | 1.10 | n.s. | 1.25 | n.s. | 1.16 | n.s. | 1.60 | n.s. | 1.17 | n.s. |
| Intensity x Run | 2.08 | 0.053 | 1.25 | n.s. | 1.84 | 0.09 | 1.61 | n.s. | 1.02 | n.s. | **2.55** | **<0.05** |
| Probability x Intensity x Run | 0.55 | n.s. | 0.90 | n.s. | 0.72 | n.s. | 0.71 | n.s. | 0.73 | n.s. | 0.84 | n.s. |
| Lag per probability: Probability | 1.51 | n.s. | 1.32 | n.s. | **1.12** | **n.s.** | 0.10 | n.s. | 0.96 | n.s. | **4.14** | **<0.01** |
| US-unpleasantness | **29.69** | **<0.001** | 2.99 | 0.097 | 1.63 | n.s. | 0.06 | n.s. | 3.34 | 0.08 | 0.27 | n.s. |

*(*) F-test and p-values were based on the model where outliers were rescored to 2SD from the mean. Note that when retaining the influential outliers for this model, the p-value of the Intensity x Run interaction was p=0.05. For all other outcome variables, rescoring the outliers did not change the results. Significant effects are indicated in bold.

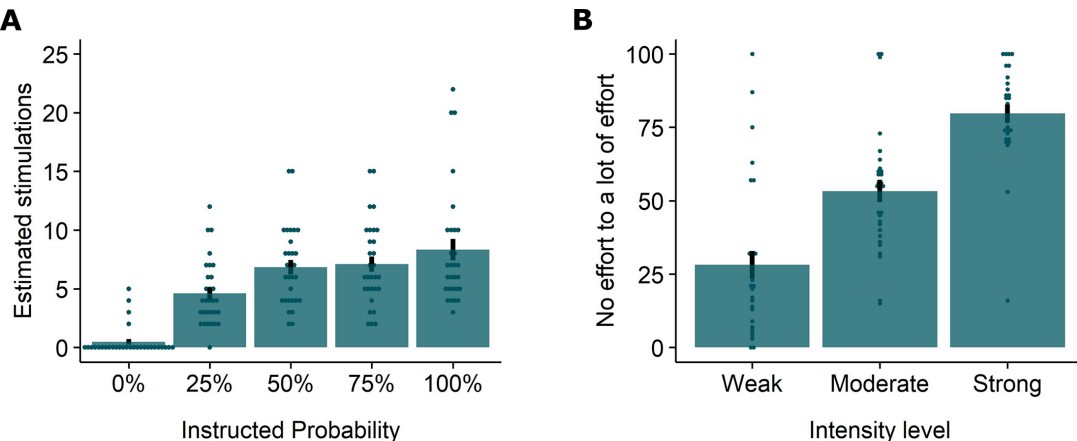

**Appendix 5—figure 1.** Post experimental recollections of stimulation and effort to avoid future stimulations. A. Participants recollected having received more stimulations following instructions of a higher probability. B. Participants were willing to exert more effort to prevent stronger stimulations. In both graphs, individual data points are presented, with the group averages plotted on top. The error bars represent standard error of the mean.

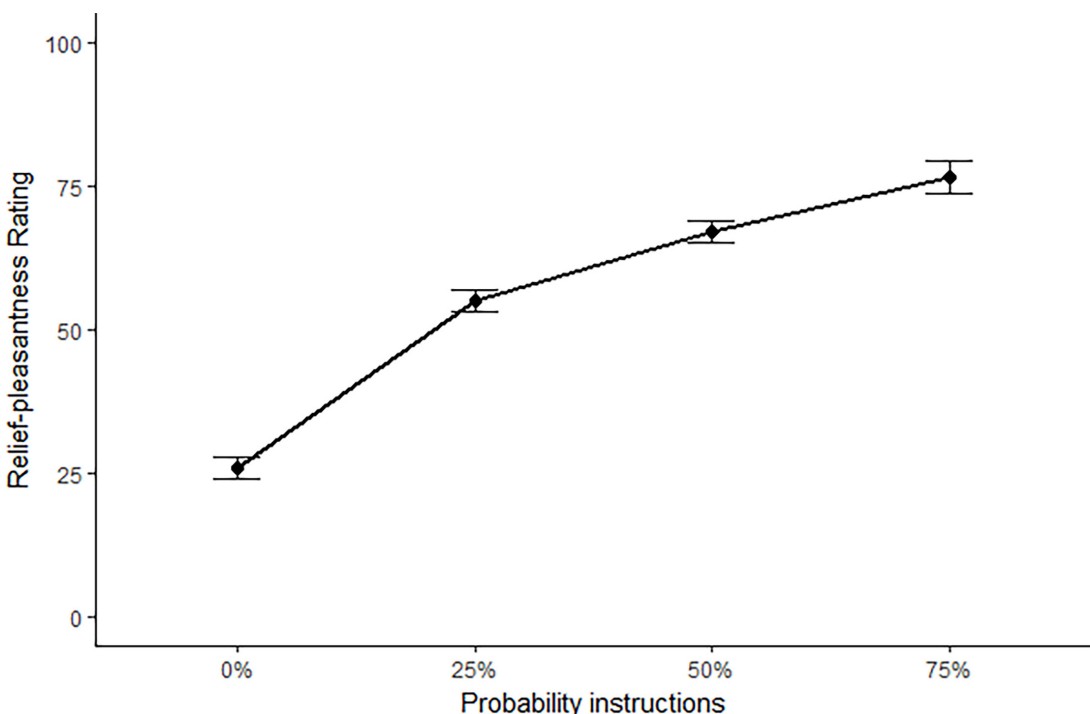

**Appendix 5—figure 2.** Results from an unpublished pilot study where the reinforcement rate matched the instructions. Main effect of Probability including 0% : $F_{(1.74, 31.23)} = 53.94$, $p < 0.001$, $\eta p 2 = 0.75$; Main effect of Probability excluding 0%: $F_{(1.50, 28.43)} = 21.03$, $p < 0.001$, $\eta p 2 = 0.53$.

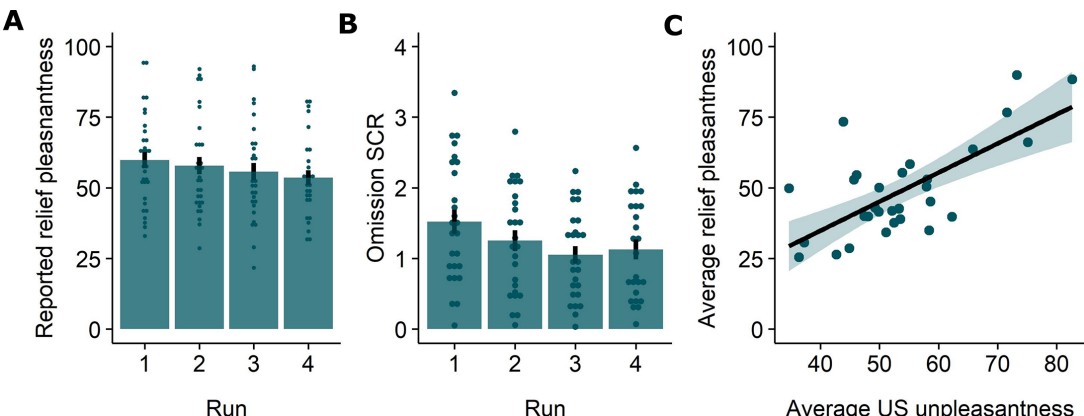

**Appendix 5—figure 3.** Run and US unpleasantness effects for subjective and physiological omission responses. Reported relief-pleasantness (A) and omission SCR (B) decreased over runs. C. Participants who perceived the stimulation as more unpleasant (on average) reported higher levels of relief (on average).

# Appendix 6

## The neural signature of omission SCR

To explore the relationship between omission-related SCR responses and omission-related fMRI activations, we assessed whole-brain (grey-matter masked) SCR modulation of omission responses by adding trial-by-trial SCR responses to the omission of the stimulation as parametric modulator to the omission regressor (similar to the parametric modulation analyses of relief). Results of this analysis are presented below for exploratory purposes, but are not interpreted. An overview of the (de)activations is shown for exploratory purposes in the *Appendix 6—figure 1* and MNI co-ordinates of the peak activations within each cluster can be found in *Appendix 6—table 1*.

Furthermore, similar to the LASSO-PCR model of relief, we trained a LASSO-PCR model for omission SCR in order to identify the pattern of omission-related brain responses that can predict the magnitude of the omission SCR response. The model was trained based on the data of SCR-responders (N=25), using five-fold cross validation. (*Appendix 6—figure 2A*) This yielded a map of positive and negative regression weights (signature response).(*Appendix 6—figure 2C*) Predicted and observed SCR correlated significantly (r=0.29, p<0.001).(*Appendix 6—figure 2B*) Bootstrap tests (5000 samples) identified the features that contributed most the prediction.

**Appendix 6—table 1.** Whole-brain SCR modulation.

**Contrast: Positive modulation**

| L/R | Region | K | p (cluster) | MNI Coordinates (xyz) | | | T (peak) |
|-----|--------|---|-------------|------|------|------|----------|
| L | Insula | 426 | <0.001 | –35 | 8 | 10 | 6.86 |
| R | Mid cingulate gyrus | 410 | <0.001 | 2 | 12 | 38 | 6.03 |
| L | Superior temporal gyrus | 87 | 0.001 | –63 | –33 | 21 | 6.03 |
| R | Supplementary motor gyrus | 235 | <0.001 | 4 | 4 | 65 | 6.01 |
| R | Insula | 554 | <0.001 | 46 | 20 | -4 | 5.91 |
| L | Cerebellum | 163 | <0.001 | –35 | –53 | –30 | 5.76 |
| L | Thalamus | 45 | 0.029 | -1 | –11 | 12 | 5.70 |
| R | Cerebellum | 50 | 0.017 | 30 | –49 | –50 | 5.59 |
| R | Cerebellum | 156 | <0.001 | 24 | –63 | –19 | 5.56 |
| R | Cerebellum, vermis | 45 | 0.029 | 6 | –61 | -6 | 5.16 |
| R | Supramarginal gyrus | 35 | 0.087 | 64 | –33 | 29 | 5.07 |
| L | Cerebellum | 42 | 0.040 | -9 | –81 | –26 | 5.01 |
| L | Cerebellum | 47 | 0.023 | –45 | –71 | –26 | 4.90 |

| Contrast: Negative modulation | | | | | | | |
|-----|--------|---|-------------|------|------|------|----------|
| L | Hippocampus | 110 | <0.001 | –23 | –11 | –23 | 7.02 |
| R | Hippocampus | 153 | <0.001 | 26 | –17 | –19 | 7.00 |
| L | Paracentral lobule | 123 | <0.001 | -3 | –31 | 69 | 6.65 |
| R | Paracentral lobule | 128 | <0.001 | 4 | –31 | 67 | 6.49 |
| L | vmPFC | 317 | <0.001 | -3 | 42 | –15 | 6.38 |
| R | Angular gyrus | 332 | <0.001 | 44 | –59 | 29 | 6.31 |
| L | Precuneus | 64 | 0.004 | -1 | –61 | 36 | 5.96 |
| L | Precuneus | 37 | 0.070 | -9 | –63 | 16 | 5.74 |
| R | Middle frontal gyrus | 50 | 0.017 | 30 | 36 | –12 | 5.67 |
| R | Posterior cingulate gyrus | 102 | <0.001 | 10 | –57 | 21 | 5.61 |
| L | Fusiform gyrus | 146 | <0.001 | –29 | –41 | –21 | 5.56 |

*Appendix 6—table 1 Continued on next page*

*Appendix 6—table 1 Continued*

**Contrast: Positive modulation**

| L/R | Region | K | p | x | y | z | T |
|-----|--------|---|---|---|---|---|---|
| R | Precentral gyrus | 109 | <0.001 | 46 | –21 | 60 | 5.54 |
| L | Middle occipital gyrus | 274 | <0.001 | –37 | –75 | 34 | 5.36 |
| R | Inferior temporal gyrus | 71 | 0.002 | 42 | –67 | -4 | 5.17 |
| L | Superior temporal gyrus | 52 | 0.014 | –63 | -5 | -6 | 5.11 |
| L | Middle temporal gyrus | 40 | 0.050 | –59 | –63 | -1 | 5.02 |
| L | Middle temporal gyrus | 39 | 0.056 | –61 | –41 | 3 | 4.91 |
| L | Middle cingulate gyrus | 89 | <0.001 | -5 | –41 | 36 | 4.88 |
| R | Middle frontal gyrus | 58 | 0.008 | 26 | 18 | 47 | 4.79 |
| L | Inferior temporal gyrus | 56 | 0.009 | –47 | 2 | –41 | 4.54 |

Note. Regions are identified at voxel-level p<0.001, and with cluster correction p<.05 (FWE-corrected); *L/R* indicates if the cluster (or peak of the cluster) is part of the left or right hemisphere; *Region* name is identified using the AAL atlas; *K* is the number of voxels in the cluster; *Coordinates* are the MNI coordinates of cluster peak; *T* is the value of the T-statistic of the cluster peak.

**Appendix 6—table 2.** Main omission SCR signature clusters identified via bootstrapping.

**Positive weight clusters**

| L/R | Region | K | MNI Coordinates (xyz) | | | Z (peak) |
|-----|--------|---|---|---|---|---|
| R | Cerebellum Crus 2 | 11 | 8 | –87 | –30 | 0.00019 |
| L | Cerebellum Crus 2 | 12 | –19 | –87 | –28 | 0.00016 |
| L | Cerebellum Crus 1 | 16 | –41 | –71 | –26 | 0.00015 |
| R | Calcarine gyrus | 12 | 2 | –91 | –12 | 0.00023 |
| R | Fusiform gyrus | 10 | 30 | –57 | -8 | 0.00020 |
| R | Inferior occipital gyrus | 15 | 36 | –89 | -6 | 0.00023 |
| L | Anterior cingulate cortex | 12 | -1 | 26 | 18 | 0.00026 |
| L | Supramarginal gyrus | 21 | –55 | –45 | 27 | 0.00020 |
| L | Cuneus | 12 | -9 | –81 | 29 | 0.00018 |
| R | Superior occipital gyrus | 70 | 30 | –71 | 43 | 0.00018 |
| L | Postcentral gyrus | 12 | –61 | –19 | 36 | 0.00013 |
| L | Precuneus | 21 | -3 | –71 | 51 | 0.00023 |

**Negative weight clusters**

| L/R | Region | K | MNI Coordinates (xyz) | | | Z (peak) |
|-----|--------|---|---|---|---|---|
| L | Cerebellum | 18 | –33 | –45 | –26 | –0.00025 |
| L | Cerebellum Crus 1 | 11 | –35 | –83 | –21 | –0.00031 |
| R | Lingual gyrus | 12 | 30 | –85 | –17 | –0.00026 |
| L | Fusiform gyrus | 33 | –23 | –69 | –15 | –0.00025 |
| R | Superior frontal gyrus | 16 | 40 | 42 | –15 | –0.00020 |
| L | Lingual gyrus | 15 | –39 | –85 | –12 | –0.00014 |
| R | Inferior occipital gyrus | 29 | 42 | –79 | –12 | –0.00014 |
| R | Inferior temporal gyrus | 10 | 60 | –41 | –12 | –0.00015 |
| R | Inferior occipital gyrus | 13 | 30 | –89 | -6 | –0.00015 |
| R | Middle temporal gyrus | 11 | 52 | –55 | 1 | –0.00017 |
| L | Middle occipital gyrus | 24 | –39 | –87 | 5 | –0.00016 |

*Appendix 6—table 2 Continued on next page*

*Appendix 6—table 2 Continued*

**Positive weight clusters**

| | | | | | | |
|---|---|---|---|---|---|---|
| R | Middle occipital gyrus | 17 | 40 | –85 | 5 | –0.00019 |
| R | Middle frontal gyrus | 17 | 42 | 42 | 7 | –0.00014 |
| L | Middle occipital gyrus | 17 | –51 | –73 | 7 | –0.00020 |
| L | Caudate | 11 | –17 | 20 | 7 | –0.00019 |
| L | Thalamus | 16 | –13 | –31 | 10 | –0.00026 |
| R | Calcarine gyrus | 10 | 4 | –75 | 12 | –0.00019 |
| R | Caudate | 14 | 14 | 6 | 16 | –0.00016 |
| R | Postcentral gyrus | 10 | 62 | -3 | 21 | –0.00017 |
| R | Frontal middle gyrus | 17 | 44 | 40 | 23 | –0.00015 |
| R | Superior frontal gyrus | 11 | 20 | 62 | 23 | –0.00015 |
| L | Precentral gyrus | 11 | –55 | 6 | 29 | –0.00013 |
| R | Precuneus | 10 | 2 | –77 | 45 | –0.00019 |
| L | Postcentral gyrus | 11 | –51 | –15 | 54 | –0.00016 |
| R | Precuneus | 11 | 10 | –69 | 60 | –0.00021 |
| R | Supplementary motor area | 15 | 4 | –25 | 60 | –0.00015 |

Note. Clusters (FDR-corrected, k>10, following bootstrapping). *L/R* indicates if the cluster (or peak of the cluster) is part of the left or right hemisphere; *Region* name is identified using the AAL atlas. *K* is the number of voxels in the cluster; *coordinates* are the MNI coordinates of cluster peak, Z is the signature weight of the cluster peak.

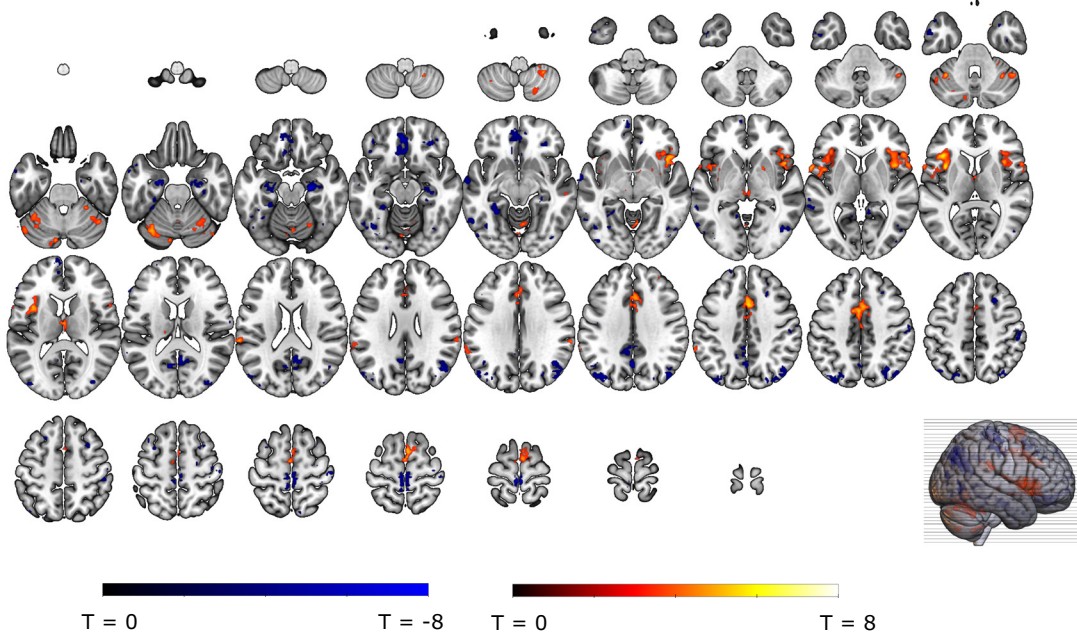

T = 0    T = -8    T = 0    T = 8

**Appendix 6—figure 1.** Whole-brain (grey-matter masked) SCR modulation of omission-related fMRI responses identified via the SCR modulation contrast, thresholded at p < 0.001 (uncorrected) for display purposes.

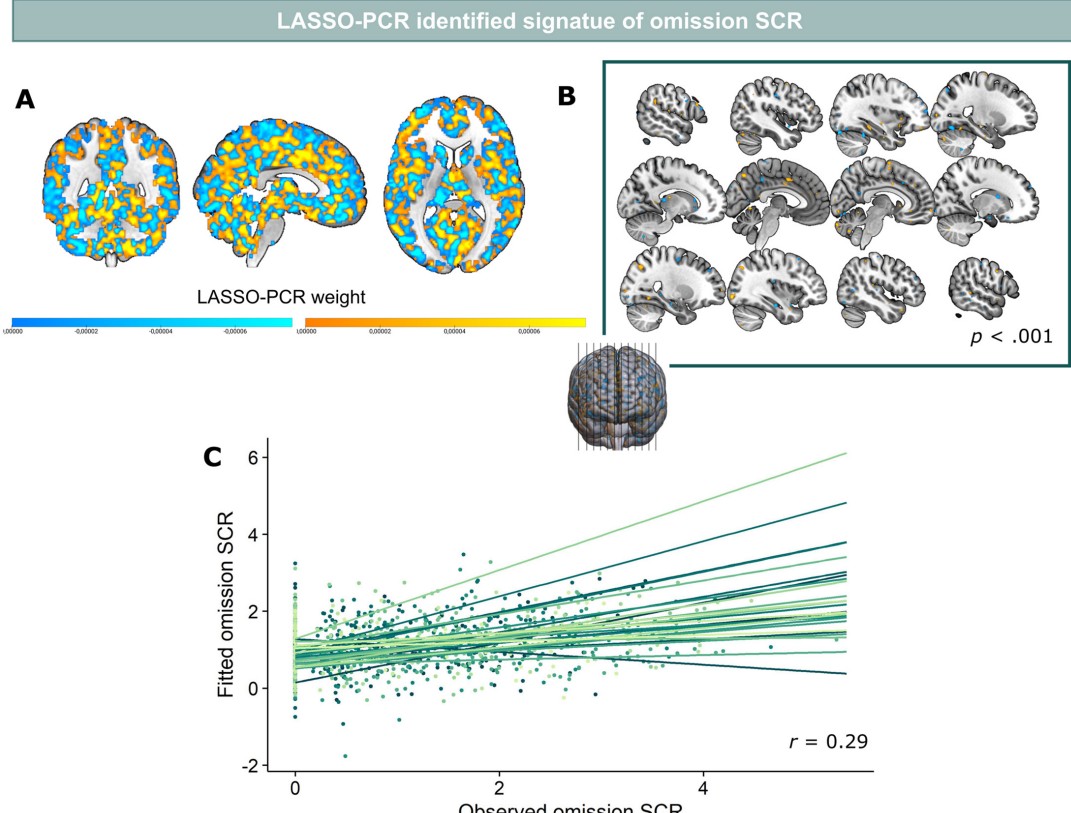

**Appendix 6—figure 2.** The signature of omission SCR. A. Signature weights. Positive weights are presented in orange-yellow; negative weights are presented in blue. B. Regions that contributed most to the signature response based on bootstrap tests (5000 samples). Clusters represent voxels with a bootstrapped p-value of <0.001 (uncorrected). C. Model predicted and observed SCR responses correlated significantly (r = 0.29).

