## [Editor Report · eLife assessment]

This study presents **valuable** findings on the relationship between prediction errors and brain activation in response to unexpected omissions of painful electric shocks. The strengths are the research question posed, as it has remained unresolved if prediction errors in the context of biologically aversive outcomes resemble reward-based prediction errors. The evidence is **solid** but there are weaknesses in the experimental design, where verbal instructions do not align with experienced outcome probabilities. It is further unclear how to interpret neural prediction error signaling in the assumed absence of learning. The work will be of interest to cognitive neuroscientists and psychologists studying appetitive and aversive learning.

---

## [Referee Report · Reviewer #1 (Public Review)]

Summary:

Willems and colleagues test whether unexpected shock omissions are associated with reward-related prediction errors by using an axiomatic approach to investigate brain activation in response to unexpected shock omission. Using an elegant design that parametrically varies shock expectancy through verbal instructions, they see a variety of responses in reward-related networks, only some of which adhere to the axioms necessary for prediction error. In addition, there were associations between omission-related responses and subjective relief. They also use machine learning to predict relief-related pleasantness and find that none of the a priori "reward" regions were predictive of relief, which is an interesting finding that can be validated and pursued in future work.

Strengths:

The authors pre-registered their approach and the analyses are sound. In particular, the axiomatic approach tests whether a given region can truly be called a reward prediction error. Although several a priori regions of interest satisfied a subset of axioms, no ROI satisfied all three axioms, and the authors were candid about this. A second strength was their use of machine learning to identify a relief-related classifier. Interestingly, none of the ROIs that have been traditionally implicated in reward prediction error reliably predicted relief, which opens important questions for future research.

Weaknesses:

The authors have done many analyses to address weaknesses in response to reviews. I will still note that given that one third of participants (n=10) did not show parametric SCR in response to instructions, it seems like some learning did occur. As prediction error is so important to such learning, a weakness of the paper is that conclusions about prediction error might differ if dynamic learning were taken into account using quantitative models.

---

## [Referee Report · Reviewer #2 (Public Review)]

The paper by Willems aimed to uncover the neural implementations of threat omissions and showed that the VTA/SN activity was stronger following the omissions of more probable and intense shock stimulation, mimicking a reward PE signal.

My main concern remains regarding the interpretation of the task as a learning paradigm (extinction) or simply an expectation violation task (difference between instructed and experienced probability), though I appreciate some of the extra analyses in the responses to the reviewers. Looking at both the behavioral and neural data, a clear difference emerges among different US intensities and non-0% vs. 0% contrasts, however, the difference across probabilities was not clear in the figures, potentially partly due to the false instructions subjects received about the shock probabilities.

The lack of probability related PE demonstration, both in behavior and to a less extent in imaging data, does not fully support the PE axioms (0% and 100% are by themselves interesting categories since the instruction and experience matched well and therefore might need to be interpreted differently from other probabilities).

As the other reviewers pointed out, the application of instruction together with extinction paradigm complicates the interpretation of results. Also, the trial-by-trial analysis suggestion was responded by the probability x run interaction analysis, which still averaged over trials within each run to estimate a beta coefficient. So my evaluation remains that this is a valuable study to test PE axioms in the human reward and salience systems but authors need to be extremely careful with their wordings as to why this task is not a particularly learning paradigm (or the learning component did not affect their results, which was in conflict with the probability related SCR, pleasantness ratings as well as BOLD signals).

---

## [Referee Report · Reviewer #3 (Public Review)]

Summary:

The authors conducted a human fMRI study investigating the omission of expected electrical shocks with varying probabilities. Participants were informed of the probability of shock and shock intensity trial-by-trial. The time point corresponding to the absence of the expected shock (with varying probability) was framed as a prediction error producing the cognitive state of relief/pleasure for the participant. fMRI activity in the VTA/SN and ventral putamen corresponded to the surprising omission of a high probability shock. Participants' subjective relief at having not been shocked correlated with activity in brain regions typically associated with reward-prediction errors. The overall conclusion of the manuscript was that the absence of an expected aversive outcome in human fMRI looks like a reward-prediction error seen in other studies that use positive outcomes.

Strengths:

Overall, I found this to be a well-written human neuroimaging study investigating an often overlooked question on the role of aversive prediction errors, and how they may differ from reward-related prediction errors. The paper is well-written and the fMRI methods seem mostly rigorous and solid.

Once again, the authors were very responsive to feedback. I have no further comments.

---

## [Author Response]

The following is the authors’ response to the previous reviews.

**Reviewer #1 (Public Review):**

The reviewer retained most of their comments from the previous reviewing round. In order to meet these comments and to further examine the dynamic nature of threat omission-related fMRI responses, we now re-analyzed our fMRI results using the single trial estimates. The results of these additional analyses are added below in our response to the recommendations for the authors of reviewer 1. However, we do want to reiterate that there was a factually incorrect statement concerning our design in the reviewer’s initial comments. Specifically, the reviewer wrote that “25% of shocks are omitted, regardless of whether subjects are told that the probability is 100%, 75%, 50%, 25%, or 0%.” We want to repeat that this is not what we did. 100% trials were always reinforced (100% reinforcement rate); 0% trials were never reinforced (0% reinforcement rate). For all other instructed probability levels (25%, 50%, 75%), the stimulation was delivered in 25% of the trials (25% reinforcement rate). We have elaborated on this misconception in our previous letter and have added this information more explicitly in the previous revision of the manuscript (e.g., lines 125-129; 223-224; 486-492).

**Reviewer #1 (Recommendations For The Authors):**
I do not have any further recommendations, although I believe an analysis of learning-related changes is still possible with the trial-wise estimates from unreinforced trials. The authors' response does not clarify whether they tested for interactions with run, and thus the fact that there are main effects does not preclude learning. I kept my original comments regarding limitations, with the exception of the suggestion to modify the title.

We thank the reviewer for this recommendation. In line with their suggestion, we have now reanalyzed our main ROI results using the trial-by-trial estimates we obtained from the firstlevel omission>baseline contrasts. Specifically, we extracted beta-estimates from each ROI and entered them into the same Probability x Intensity x Run LMM we used for the relief and SCR analyses. Results from these analyses (in the full sample) were similar to our main results. For the VTA/SN model, we found main effects of Probability (F = 3.12, p = .04), and Intensity (F = 7.15, p < .001) (in the model where influential outliers were rescored to 2SD from mean). There was no main effect of Run (F = 0.92, p = .43) and no Probability x Run interaction (F = 1.24, p = .28). If the experienced contingency would have interfered with the instructions, there should have been a Probability x Run interaction (with the effect of Probability only being present in the first runs). Since we did not observe such an interaction, our results indicate that even though some learning might still have taken place, the main effect of Probability remained present throughout the task.

There is an important side note regarding these analyses: For the first level GLM estimation, we concatenated the functional runs and accounted for baseline differences between runs by adding run-specific intercepts as regressors of no-interest. Hence, any potential main effect of run was likely modeled out at first level. This might explain why, in contrast to the rating and SCR results (see Supplemental Figure 5), we found no main effect of Run. Nevertheless, interaction effects should not be affected by including these run-specific intercepts.

Note that when we ran the single-trial analysis for the ventral putamen ROI, the effect of intensity became significant (F = 3.89, p = .02). Results neither changed for the NAc, nor the vmPFC ROIs.

**Reviewer #2 (Public Review):**
Comments on revised version:I want to thank the authors for their thorough and comprehensive work in revising this manuscript. I agree with the authors that learning paradigms might not be a necessity when it comes to study the PE signals, but I don't particularly agree with some of the responses in the rebuttal letter ("Furthermore, conditioning paradigms generally only include one level of aversive outcome: the electrical stimulation is either delivered or omitted."). This is of course correct description for the conditioning paradigm, but the same can be said for an instructed design: the aversive outcome was either delivered or not. That being said, adopting the instructed design itself is legitimate in my opinion.

We thank the reviewer for this comment. We have now modified the phrasing of this argument to clarify our reasoning (see lines 102-104: “First, these only included one level of aversive outcome: the electrical stimulation was either delivered at a fixed intensity, or omitted; but the intensity of the stimulation was never experimentally manipulated within the same task.”).

The reason why we mentioned that “the aversive outcome is either delivered or omitted” is because in most contemporary conditioning paradigms only one level of aversive US is used. In these cases, it is therefore not possible to investigate the effect of US Intensity. In our paradigm, we included multiple levels of aversive US, allowing us to assess how the level of aversiveness influences threat omission responding. It is indeed true that each level was delivered or not. However, our data clearly (and robustly across experiments, see Willems & Vervliet, 2021) demonstrate that the effects of the instructed and perceived unpleasantness of the US (as operationalized by the mean reported US unpleasantness during the task) on the reported relief and the omission fMRI responses are stronger than the effect of instructed probability.

My main concern, which the authors spent quite some length in the rebuttal letter to address, still remains about the validity for different instructed probabilities. Although subjects were told that the trials were independent, the big difference between 75% and 25% would more than likely confuse the subjects, especially given that most of us would fall prey to the Gambler's fallacy (or the law of small numbers) to some degree. When the instruction and subjective experience collides, some form of inference or learning must have occurred, making the otherwise straightforward analysis more complex. Therefore, I believe that a more rigorous/quantitative learning modeling work can dramatically improve the validity of the results. Of course, I also realize how much extra work is needed to append the computational part but without it there is always a theoretical loophole in the current experimental design.

We agree with the reviewer that some learning may have occurred in our task. However, we believe the most important question in relation to our study is: to what extent did this learning influence our manipulations of interest?

In our reply to reviewer 1, we already showed that a re-analysis of the fMRI results using the trial-by-trial estimates of the omission contrasts revealed no Probability x Run interaction, suggesting that – overall – the probability effect remained stable over the course of the experiment. However, inspired by the alternative explanation that was proposed by this reviewer, we now also assessed the role of the Gambler’s fallacy in a separate set of analyses. Indeed, it is possible that participants start to expect a stimulation more after more time has passed since the last stimulation was experienced. To test this alternative hypothesis, we specified two new regressors that calculated for each trial of each participant how many trials had passed since the last stimulation (or since the beginning of the experiment) either overall (across all trials of all probability types; hence called the overall-lag regressor) or per probability level (across trials of each probability type separately; hence called the lag-per-probability regressor). For both regressors a value of 0 indicates that the previous trial was either a stimulation trial or the start of experiment, a value of 1 means that the last stimulation trial was 2 trials ago, etc.

The results of these additional analyses are added in a supplemental note (see supplemental note 6), and referred to in the main text see lines 231-236: “Likewise, a post-hoc trial-by-trial analysis of the omission-related fMRI activations confirmed that the Probability effect for the VTA/SN activations was stable over the course of the experiment (no Probability x Run interaction) and remained present when accounting for the Gambler’s fallacy (i.e., the possibility that participants start to expect a stimulation more when more time has passed since the last stimulation was experienced) (see supplemental note 6). Overall, these post-hoc analyses further confirm the PE-profile of omission-related VTA/SN responses”.

Addition to supplemental material (pages 16-18)

Supplemental Note 6: The effect of Run and the Gambler’s Fallacy

A question that was raised by the reviewers was whether omission-related responses could be influenced by dynamical learning or the Gambler’s Fallacy, which might have affected the effectiveness of the Probability manipulation.

Inspired by this question, we exploratorily assessed the role of the Gambler’s Fallacy and the effects of Run in a separate set of analyses. Indeed, it is possible that participants start to expect a stimulation more when more time has passed since the last stimulation was experienced. To test this alternative hypothesis, we specified two new regressors that calculated for each trial of each participant how many trials had passed since the last stimulation (or since the beginning of the experiment) either overall (across all trials of all probability types; hence called the overall-lag regressor) or per probability level (across trials of each probability type separately; hence called the lag-per-probability regressor). For both regressors a value of 0 indicates that the previous trial was either a stimulation trial or the start of experiment, a value of 1 means that the last stimulation trial was 2 trials ago, etc.

The new models including these regressors for each omission response type (i.e., omission-related activations for each ROI, relief, and omission-SCR) were specified as follows:

(1) For the overall lag:

Omission response ~ Probability * Intensity * Run + US-unpleasantness + Overall-lag + (1|Subject).

(2) For the lag per probability level:

Omission response ~ Probability * Intensity * Run + US-unpleasantness + Lag-perprobability : Probability + (1|Subject).

Where US-unpleasantness scores were mean-centered across participants; “*” represents main effects and interactions, and “:” represents an interaction (without main effect). Note that we only included an interaction for the lag-per-probability model to estimate separate lag-parameters for each probability level.

The results of these analyses are presented in the tables below. Overall, we found that adding these lag-regressors to the model did not alter our main results. That is: for the VTA/SN, relief and omission-SCR, the main effects of Probability and Intensity remained. Interestingly, the overall-lag-effect itself was significant for VTA/SN activations and omission SCR, indicating that VTA/SN activations were larger when more time had passed since the last stimulation (beta = 0.19), whereas SCR were smaller when more time had passed (beta = -0.03). This pattern is reminiscent of the Perruchet effect, namely that the explicit expectancy of a US increases over a run of non-reinforced trials (in line with the gambler’s fallacy effect) whereas the conditioned physiological response to the conditional stimulus declines (in line with an extinction effect, Perruchet, 1985; McAndrew, Jones, McLaren, & McLaren, 2012). Thus, the observed dissociation between the VTA/SN activations and omission SCR might similarly point to two distinctive processes where VTA/SN activations are more dependent on a consciously controlled process that is subjected to the gambler’s fallacy, whereas the strength of the omission SCR responses is more dependent on an automatic associative process that is subjected to extinction. Importantly, however, even though the temporal distance to the last stimulation had these opposing effects on VTA/SN activations and omission SCRs, the main effects of the probability manipulation remained significant for both outcome variables. This means that the core results of our study still hold.

Next to the overall-lag effect, the lag-per-probability regressor was only significant for the vmPFC. A follow-up of the beta estimates of the lag-per-probability regressors for each probability level revealed that vmPFC activations increased with increasing temporal distance from the stimulation, but only for the 50% trials (beta = 0.47, t = 2.75, p < .01), and not the 25% (beta = 0.25, t = 1.49, p = .14) or the 75% trials (beta = 0.28, t = 1.62, p = .10).

Author response table 1.

**Author response table 1. sa4table1:** 

	Omission responses											
Regressor	Relief		SCR		VTA/SN (*)		Left vPut		NAC		vmPFC	
	F	P		P	F	P						
Probability	30.04	<.001	4.9	<.01	3.59	<.05	0.18	n.s.	0.88	n.s.	1.73	n.s.
Intensity	620.62	<.001	106.65	<.001	7.81	<.001	3.88	<.05	0.7	n.s.	4.9	<.01
Run	9.71	<.001	15.56	<.001	1.13	n.s.	0.76	n.s.	0.62	n.s.	0.44	n.s.
Probability x Intensity	3.69	<.01	1.54	n.s.	1.15	n.s.	1.39	n.s.	1.76	n.s.	0.7	n.s.
Probability x Run	1.13	n.s.	1.24	n.s.	1.02	n.s.	1.22	n.s.	1.74	n.s.	1.26	n.s.
Intensity x Run	1.94	0.07	1.3	n.s.	1.94	0.07	1.59	n.s.	1.01	n.s.	2.41	<.05
Probability x Intensity x Run	0.56	n.s.	0.87	n.s.	0.76	n.s.	0.71	n.s.	0.76	n.s.	0.83	n.s.
Overall-lag	2.56	n.s.	4.68	<.05	11.3	<.001	<0.01	n.s.	0.16	n.s.	<0.01	n.s.
US-unpleasantness	29.6	<.001	3	0.096	1.84	n.s.	0.06	n.s.	3.44	0.07	0.26	n.s.

F-statistics and corresponding p-values from the overall lag model

(*) F-test and p-values were based on the model where outliers were rescored to 2SD from the mean. Note that when retaining the influential outliers for this model, the p-value of the probability effect was p = .06. For all other outcome variables, rescoring the outliers did not change the results. Significant effects are indicated in bold.

Author response table 2.

**Author response table 2. sa4table2:** 

	Omission responses
Regressor	Relief	SCR	VTA/SN (*)	Left vPut	NAC	vmPFC
FF	P	F	P	F	P	F	P	F	P	F	P
											
Probability	23.07	<.001	4.44	<.05	3.28	<.05	0.22	n.s.	0.18	n.s.	0.31	n.s.
Intensity	625.52	<.001	107.49	<0.01	7.33	<0.01	3.88	<.05	0.66	n.s.	4.85	<.01
Run	8.63	<.001	13.91	<0.01	1.09	n.s.	0.72	n.s.	0.60	n.s.	1.09	n.s.
Probability x Intensity	3.87	<.01	1.33	n.s.	1.01	n.s.	1.40	n.s.	1.81	n.s.	0.63	n.s.
Probability x Run	1.63	n.s.	1.10	n.s.	1.25	n.s.	1.16	n.s.	1.60	n.s.	1.17	n.s.
Intensity x Run	2.08	.053	1.25	n.s.	1.84	.09	1.61	n.s.	1.02	n.s.	2.55	<.05
Probability x Intensity x Run	0.55	n.s.	0.90	n.s.	0.72	n.s.	0.71	n.s.	0.73	n.s.	0.84	n.s.
Lag per probability: Probability	1.51	n.s.	1.32	n.s.	1.12	n.s.	0.10	n.s.	0.96	n.s.	4.14	<.01
US- unpleasantness	26.69	<.001	2.99	.097	1.63	n.s.	0.06	n.s.	3.34	.08	0.27	n.s.

Table 2 F-statistics and corresponding p-values from the lag per probability level model

(*) F-test and p-values were based on the model where outliers were rescored to 2SD from the mean. Note that when retaining the influential outliers for this model, the p-value of the Intensity x Run interaction was p = .05. For all other outcome variables, rescoring the outliers did not change the results. Significant effects are indicated in bold.

As the authors mentioned in the rebuttal letter, "selecting participants only if their anticipatory SCR monotonically increased with each increase in instructed probability 0% < 25% < 50% < 75% < 100%, N = 11 participants", only ~1/3 of the subjects actually showed strong evidence for the validity of the instructions. This further raises the question of whether the instructed design, due to the interference of false instruction and the dynamic learning among trials, is solid enough to test the hypothesis .

We agree with the reviewer that a monotonic increase in anticipatory SCR with increasing probability instructions would provide the strongest evidence that the manipulation worked. However, it is well known that SCR is a noisy measure, and so the chances to see this monotonic increase are rather small, even if the underlying threat anticipation increases monotonically. Furthermore, between-subject variation is substantial in physiological measures, and it is not uncommon to observe, e.g., differential fear conditioning in one measure, but not in another (Lonsdorf & Merz, 2017). It is therefore not so surprising that ‘only’ 1/3 of our participants showed the perfect pattern of monotonically increasing SCR with increasing probability instructions. That being said, it is also important to note that not all participants were considered for these follow-up analyses because valid SCR data was not always available.

Specifically, N = 4 participants were identified as anticipation non-responders (i.e. participant with smaller average SCR to the clock on 100% than on 0% trials; pre-registered criterium) and were excluded from the SCR-related analyses, and N = 1 participant had missing data due to technical difficulties. This means that only 26 (and not 31) participants were considered for the post hoc analyses. Taking this information into account, this means that 21 out of 26 participants (approximately 80%) showed stronger anticipatory SCR following 75% instructions compared to 25% instructions and that 11 out of 26 participants (approximately 40%) even showed the monotonical increase in their anticipatory SCR (see supplemental figure 4). Furthermore, although anticipatory SCR gradually decreased over the course of the experiment, there was no Run x Probability interaction, indicating that the instructions remained stable throughout the task (see supplemental figure 3).

**Reviewer #2 (Recommendations For The Authors):**
A more operational approach might be to break the trials into different sections along the timeline and examine how much the results might have been affected across time. I expect the manipulation checks would hold for the first one or two runs and the authors then would have good reasons to focus on the behavioral and imaging results for those runs.

This recommendation resembles the recommendation by reviewer 1. In our reply to reviewer 1, we showed the results of a re-analysis of the fMRI data using the trial-by-trial estimates of the omission contrasts, which revealed no Probability x Run interaction, suggesting that – overall - the probability effect remained (more or less) stable over the course of the experiment. For a more in depth discussion of the results of this additional analysis, we refer to our answer to reviewer 1.

**Reviewer #3 (Public Review):**
Comments on revised version:The authors were extremely responsive to the comments and provided a comprehensive rebuttal letter with a lot of detail to address the comments. The authors clarified their methodology, and rationale for their task design, which required some more explanation (at least for me) to understand. Some of the design elements were not clear to me in the original paper.The initial framing for their study is still in the domain of learning. The paper starts off with a description of extinction as the prime example of when threat is omitted. This could lead a reader to think the paper would speak to the role of prediction errors in extinction learning processes. But this is not their goal, as they emphasize repeatedly in their rebuttal letter. The revision also now details how using a conditioning/extinction framework doesn't suit their experimental needs.

We thank the reviewer for pointing out this potential cause of confusion. We have now rewritten the starting paragraph of the introduction to more closely focus on prediction errors, and only discuss fear extinction as a potential paradigm that has been used to study the role of threat omission PE for fear extinction learning (see lines 40-55). We hope that these adaptations are sufficient to prevent any false expectations. However, as we have mentioned in our previous response letter, not talking about fear extinction at all would also not make sense in our opinion, since most of the knowledge we have gained about threat omission prediction errors to date is based on studies that employed these paradigms.

Adaptation in the revised manuscript (lines 40-55):

“We experience pleasurable relief when an expected threat stays away1. This relief indicates that the outcome we experienced (“nothing”) was better than we expected it to be (“threat”). Such a mismatch between expectation and outcome is generally regarded as the trigger for new learning, and is typically formalized as the prediction error (PE) that determines how much there can be learned in any given situation2. Over the last two decades, the PE elicited by the absence of expected threat (threat omission PE) has received increasing scientific interest, because it is thought to play a central role in learning of safety. Impaired safety learning is one of the core features of clinical anxiety4. A better understanding of how the threat omission PE is processed in the brain may therefore be key to optimizing therapeutic efforts to boost safety learning. Yet, despite its theoretical and clinical importance, research on how the threat omission PE is computed in the brain is only emerging.

To date, the threat omission PE has mainly been studied using fear extinction paradigms that mimic safety learning by repeatedly confronting a human or animal with a threat predicting cue (conditional stimulus, CS; e.g. a tone) in the absence of a previously associated aversive event (unconditional stimulus, US; e.g., an electrical stimulation). These (primarily non-human) studies have revealed that there are striking similarities between the PE elicited by unexpected threat omission and the PE elicited by unexpected reward.”

It is reasonable to develop a new task to answer their experimental questions. By no means is there a requirement to use a conditioning/extinction paradigm to address their questions. As they say, "it is not necessary to adopt a learning paradigm to study omission responses", which I agree with. But the authors seem to want to have it both ways: they frame their paper around how important prediction errors are to extinction processes, but then go out of their way to say how they can't test their hypotheses with a learning paradigm.Part of their argument that they needed to develop their own task "outside of a learning context" goes as follows:(1) "...conditioning paradigms generally only include one level of aversive outcome: the electrical stimulation is either delivered or omitted. As a result, the magnitude-related axiom cannot be tested."(2) "....in conditioning tasks people generally learn fast, rendering relatively few trials on which the prediction is violated. As a result, there is generally little intra-individual variability in the PE responses"(3) "...because of the relatively low signal to noise ratio in fMRI measures, fear extinction studies often pool across trials to compare omission-related activity between early and late extinction, which further reduces the necessary variability to properly evaluate the probability axiom"These points seem to hinge on how tasks are "generally" constructed. However, there are many adaptations to learning tasks:(1) There is no rule that conditioning can't include different levels of aversive outcomes following different cues. In fact, their own design uses multiple cues that signal different intensities and probabilities. Saying that conditioning "generally only include one level of aversive outcome" is not an explanation for why "these paradigms are not tailored" for their research purposes. There are also several conditioning studies that have used different cues to signal different outcome probabilities. This is not uncommon, and in fact is what they use in their study, only with an instruction rather than through learning through experience, per se.(2) Conditioning/extinction doesn't have to occur fast. Just because people "generally learn fast" doesn't mean this has to be the case. Experiments can be designed to make learning more challenging or take longer (e.g., partial reinforcement). And there can be intra-individual differences in conditioning and extinction, especially if some cues have a lower probability of predicting the US than others. Again, because most conditioning tasks are usually constructed in a fairly simplistic manner doesn't negate the utility of learning paradigms to address PEaxioms.(3) Many studies have tracked trial-by-trial BOLD signal in learning studies (e.g., using parametric modulation). Again, just because other studies "often pool across trials" is not an explanation for these paradigms being ill-suited to study prediction errors. Indeed, most computational models used in fMRI are predicated on analyzing data at the trial level.

We thank the reviewer for these remarks. The “fear conditioning and extinction paradigms” that we were referring to in this paragraph were the ones that have been used to study threat omission PE responses in previous research (e.g., Raczka et al., 2011; Thiele et al. 2021; Lange et al. 2020; Esser et al., 2021; Papalini et al., 2021; Vervliet et al. 2017). These studies have mainly used differential/multiple-cue protocols where either one (or two) CS+ and one CS- are trained in an acquisition phase and extinguished in the next phase. Thus, in these paradigms: (1) only one level of aversive US is used; and (2) as safety learning develops over the course of extinction, there are relatively few omission trials during which “large” threat omission PEs can be observed (e.g. from the 24 CS+ trials that were used during extinction in Esser et al., the steepest decreases in expectancy – and thus the largest PE – were found in first 6 trials); and (3) there was never absolute certainty that the stimulation will no longer follow. Some of these studies have indeed estimated the threat omission PE during the extinction phase based on learning models, and have entered these estimates as parametric modulators to CS-offset regressors. This is very informative. However, the exact model that was used differed per study (e.g. Rescorla-Wagner in Raczka et al. and Thiele et al.; or a Rescorla- Wagner–Pearce- Hall hybrid model in Esser et al.). We wanted to analyze threat omission-responses without commitment to a particular learning model. Thus, in order to examine how threat omissionresponses vary as a function of probability-related expectations, a paradigm that has multiple probability levels is recommended (e.g. Rutledge et al., 2010; Ojala et al., 2022)

The reviewer rightfully pointed out that conditioning paradigms (more generally) can be tailored to fit our purposes as well. Still, when doing so, the same adaptations as we outlined above need to be considered: i.e. include different levels of US intensity; different levels of probability; and conditions with full certainty about the US (non)occurrence. In our attempt to keep the experimental design as simple and straightforward as possible, we decided to rely on instructions for this purpose, rather than to train 3 (US levels) x 5 (reinforcement levels) = 15 different CSs. It is certainly possible to train multiple CSs of varying reinforcement rates (e.g. Grings et al. 1971, Ojala et al., 2022). However, given that US-expectation on each trial would primarily depend on the individual learning processes of the participants, using a conditioning task would make it more difficult to maintain experimental control over the level of USexpectation elicited by each CS. As a result, this would likely require more extensive training, and thus prolong the study procedure considerably. Furthermore, even though previous studies have trained different CSs for different reinforcement rates, most of these studies have only used one level of US. Thus, in order to not complexify our task to much, we decided to rely on instructions rather than to train CSs for multiple US levels (in addition to multiple reinforcement rates).

We have tried to clarify our reasoning in the revised version of the manuscript (see introduction, lines 100-113):

“The previously discussed fear conditioning and extinction studies have been invaluable for clarifying the role of the threat omission PE within a learning context. However, these studies were not tailored to create the varying intensity and probability-related conditions that are required to systematically evaluate the threat omission PE in the light of the PE axioms. First, these only included one level of aversive outcome: the electrical stimulation was either delivered or omitted; but the intensity of the stimulation was never experimentally manipulated within the same task. As a result, the magnitude-related axiom could not be tested. Second, as safety learning progressively developed over the course of extinction learning, the most informative trials to evaluate the probability axiom (i.e. the trials with the largest PE) were restricted to the first few CS+ offsets of the extinction phase, and the exact number of these informative trials likely differed across participants as a result of individually varying learning rates. This limited the experimental control and necessary variability to systematically evaluate the probability axiom. Third, because CS-US contingencies changed over the course of the task (e.g. from acquisition to extinction), there was never complete certainty about whether the US would (not) follow. This precluded a direct comparison of fully predicted outcomes. Finally, within a learning context, it remains unclear whether brain responses to the threat omission are in fact responses to the violation of expectancy itself, or whether they are the result of subsequent expectancy updating.”

Again, the authors are free to develop their own task design that they think is best suited to address their experimental questions. For instance, if they truly believe that omission-related responses should be studied independent of updating. The question I'm still left puzzling is why the paper is so strongly framed around extinction (the word appears several times in the main body of the paper), which is a learning process, and yet the authors go out of their way to say that they can only test their hypotheses outside of a learning paradigm.

As we have mentioned before, the reason why we refer to extinction studies is because most evidence on threat omission PE to date comes from fear extinction paradigms.

The authors did address other areas of concern, to varying extents. Some of these issues were somewhat glossed over in the rebuttal letter by noting them as limitations. For example, the issue with comparing 100% stimulation to 0% stimulation, when the shock contaminates the fMRI signal. This was noted as a limitation that should be addressed in future studies, bypassing the critical point.

It is unclear to us what the reviewer means with “bypassing the critical point”. We argued in the manuscript that the contrast we initially specified and preregistered to study axiom 3 (fully predicted outcomes elicit equivalent activation) could not be used for this purpose, as it was confounded by the delivery of the stimulation. Because 100% trials aways included the stimulation and 0% trials never included stimulation, there was no way to disentangle activations related to full predictability from activations related to the stimulation as such.

**Reviewer #3 (Recommendations For The Authors):**
I'm not sure the new paragraph explaining why they can't use a learning task to test their hypotheses is very convincing, as I noted in my review. Again, it is not a problem to develop a new task to address their questions. They can justify why they want to use their task without describing (incorrectly in my opinion) that other tasks "generally" are constructed in a way that doesn't suit their needs.

For an overview of the changes we made in response to this recommendation, we refer to our reply to the public review.

We look forward to your reply and are happy to provide answers to any further questions or comments you may have.